# Quantitative imaging of lipid transport in mammalian cells

Juan M. Iglesias-Artola[1], Kristin Böhlig[1], Kai Schuhmann[1], Katelyn C. Cook[1], H. Mathilda Lennartz[1], Milena Schuhmacher[1,2], Pavel Barahtjan[1,2], Cristina Jiménez López[1], Radek Šachl[3], Vannuruswamy Garikapati[1], Karina Pombo-Garcia[1], Annett Lohmann[1], Petra Riegerová[3], Martin Hof[3], Björn Drobot[4], Andrej Shevchenko[1], Alf Honigmann[1,5,6 ✉] & André Nadler[1 ✉]

Eukaryotic cells produce over 1,000 different lipid species that tune organelle membrane properties, control signalling and store energy[1,2]. How lipid species are selectively sorted between organelles to maintain specific membrane identities is largely unclear, owing to the difficulty of imaging lipid transport in cells[3]. Here we measured the retrograde transport and metabolism of individual lipid species in mammalian cells using time-resolved fluorescence imaging of bifunctional lipid probes in combination with ultra-high-resolution mass spectrometry and mathematical modelling. Quantification of lipid flux between organelles revealed that directional, non-vesicular lipid transport is responsible for fast, species-selective lipid sorting, in contrast to the slow, unspecific vesicular membrane trafficking. Using genetic perturbations, we found that coupling between energy-dependent lipid flipping and non-vesicular transport is a mechanism for directional lipid transport. Comparison of metabolic conversion and transport rates showed that non-vesicular transport dominates the organelle distribution of lipids, while species-specific phospholipid metabolism controls neutral lipid accumulation. Our results provide the first quantitative map of retrograde lipid flux in cells[4]. We anticipate that our pipeline for mapping of lipid flux through physical and chemical space in cells will boost our understanding of lipids in cell biology and disease.

Eukaryotic cells produce thousands of chemically distinct lipid species with varying side-chain unsaturation, length and regiochemistry that belong to dozens of lipid classes defined on the basis of the lipid headgroup and backbone[1]. Lipid species are differentially distributed across organelle membranes, which is important to establish organelle identities and functions[2,3,5,6]. How the organelle-specific distribution of lipids is established and maintained is incompletely understood[3].

Lipid biosynthesis occurs mostly in the endoplasmic reticulum (ER) and lipids are subsequently distributed through vesicular trafficking and membrane-contact sites to other organelles[2,7,8]. During antero-grade transport towards the plasma membrane (PM) lipids are modified, before they are either recycled through retrograde transport to the ER or catabolized in lysosomes, peroxisomes and mitochondria. Understanding how organelle membrane lipid identities arise requires quantitative measurements of intracellular transport kinetics and local metabolism of individual, molecular distinct lipid species.

While anterograde lipid flux from the ER to the PM has been characterized for some lipid classes using metabolic labelling and organelle fractionation[3], the trafficking of individual lipid species in particular in the retrograde lipid transport pathway is not well understood, with the exception of sphingomyelin (SM), of which endocytic trafficking

has been studied in some detail[9]. Thus far, one of the key limitations has been that distinct lipid species could not be faithfully imaged using fluorescence microscopy, hindering the analysis of transport dynamics. Here we used minimally modified lipid probes, ultra-high-resolution Fourier-transform (FT) mass spectrometry (MS), fluorescence imaging and mathematical modelling to quantitatively map the kinetics of species-specific lipid transport and metabolism, identify the primary mechanism of lipid sorting into organelle membranes and build a publicly accessible lipid flux atlas (http://doi.org/21.11101/0000-0007-FCE5-B).

## Fluorescence imaging of lipid transport

To quantify the kinetics of transport and metabolism of individual lipid species in mammalian cells, we made use of photoactivatable and clickable (bifunctional) lipids[10–15] and a combination of pulse–chase fluorescence imaging with ultra-high-resolution MS and mathematical modelling. In contrast to other lipid probes that are optimized to either modulate lipid levels (photocaged lipids[16–18], photoswitchable lipids[19,20]), to visualize lipid localization (lipid–fluorophore conjugates[21,22]) or to monitor lipid metabolism (isotope-labelled lipids[23,24],

[1]Max Planck Institute of Molecular Cell Biology and Genetics, Dresden, Germany. [2]École polytechnique fédérale de Lausanne, Lausanne, Switzerland. [3]J. Heyrovský Institute of Physical Chemistry, Academy of Sciences of the Czech Republic v.v.i., Prague, Czech Republic. [4]Helmholtz Zentrum Dresden Rossendorf, Institute of Resource Ecology, Dresden, Germany. [5]Technische Universität Dresden, Biotechnologisches Zentrum, Center for Molecular and Cellular Bioengineering (CMCB), Dresden, Germany. [6]Cluster of Excellence Physics of Life, TU Dresden, Dresden, Germany. ✉e-mail: alf.honigmann@tu-dresden.de; nadler@mpi-cbg.de

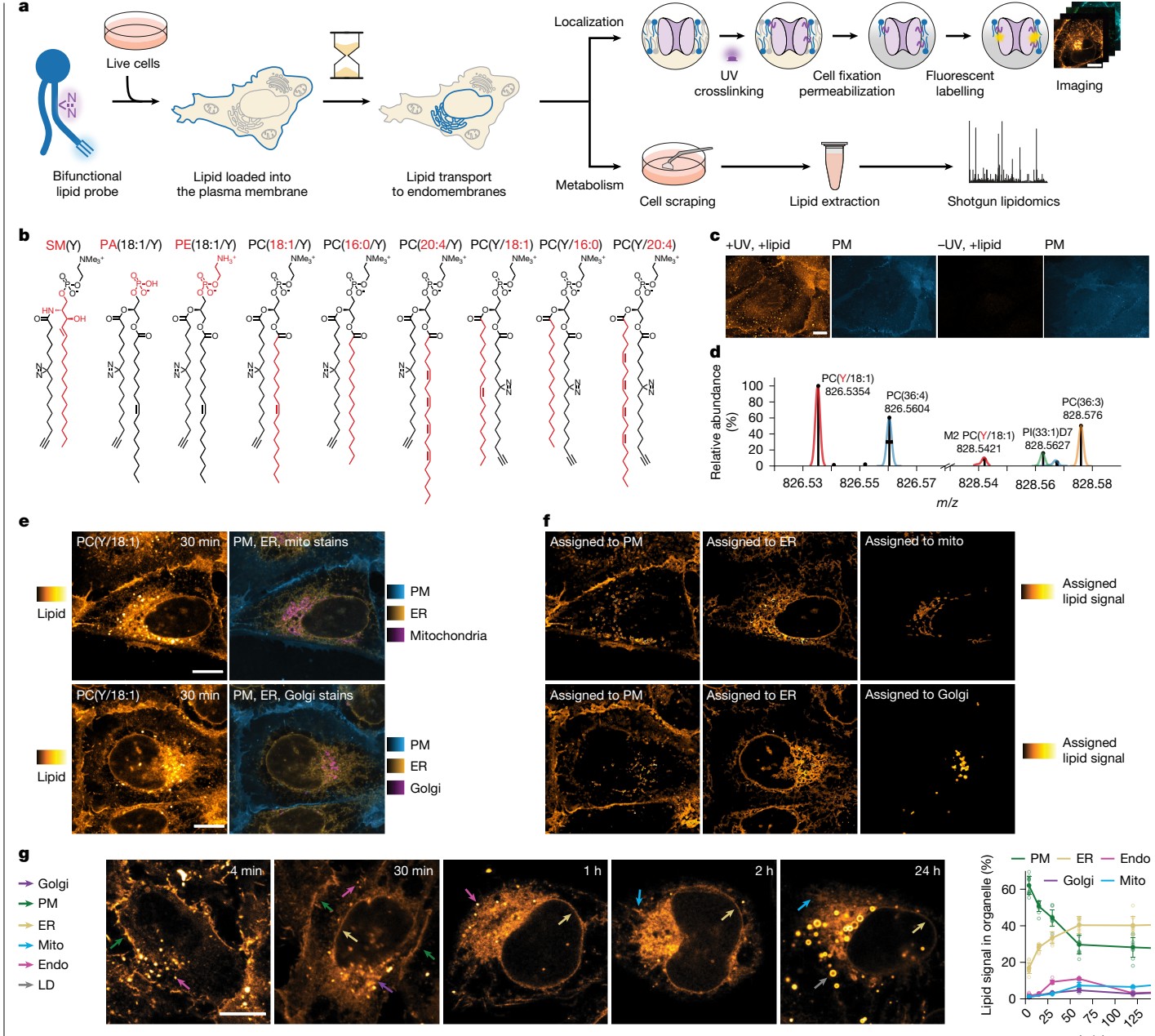

**Fig. 1 | Lipid probe library, imaging and MS workflows, and lipid transport time-course experiments. a**, Schematic of the combined analysis of lipid transport and metabolism. Lipid probes were loaded into the PM using α-methyl-cyclodextrin-mediated exchange reactions, crosslinked and fluorescently labelled for imaging or extracted and analysed by MS to monitor metabolism. **b**, The bifunctional lipid probes synthesized for this study. Unique structural elements are highlighted in red. **c**, Lipid delivery to the PM and the selectivity of lipid labelling for PC(Y/18:1). Scale bar, 10 μm. **d**, Ultra-high mass resolution (resolution$_{m/z=800}$ = 420,000) enables baseline separation of peaks spaced by a few millidaltons and their unequivocal assignment to molecular ions of lipids (as annotated; [M-H]⁻/[M+HCO₂]⁻) in total lipid extract. M2: second isotopic peak; PI(33:1)D7: deuterated internal standard. **e**, Representative images (PC(Y/18:1), 30 min timepoint) showing lipid signal (left) and individual organelles markers (right) by four-colour fluorescence imaging. Scale bars, 10 μm. **f**, Lipid signal assignment for cells shown in **e** based on automated image segmentation. Lipid signal images are shown at identical settings and scale in **e** and **f**. **g**, Representative images from time-course experiments show the temporal development of the lipid signal distribution for PC(Y/18:1) (left). The coloured arrows indicate lipid localization in different organelles (green, PM; yellow, ER; cyan, mitochondria (Mito); violet, Golgi apparatus; magenta, endosomes (Endo); grey, lipid droplets). Scale bar, 10 μm. Images were brightness–contrast adjusted to enable comparison of lipid distributions at different timepoints. Right, quantification of temporal development of intracellular lipid distribution for PC(Y/18:1). Kinetics were constructed from five independent timepoints. Data are mean ± s.d. Individual *n* values are provided as source data.

clickable lipids[25–27]), bifunctional lipids enable the monitoring of both lipid localization and metabolism using the same probe.

To make distinct lipid species accessible for high-resolution fluorescence imaging and MS, we relied on two minimal modifications (diazirine and alkyne) within the lipid alkyl chain (Fig. 1a). We generated a library of bifunctional lipid probes covering four different lipid classes: phosphatidylcholine (PC), phosphatidic acid (PA), phosphatidylethanolamine (PE) and SM (Fig. 1b). PC and PE probes were chosen, as these two lipid classes constitute the majority of all phospholipids in mammalian cells (PC, ~33%; PE, ~20%; Extended Data Fig. 1).

PA was included as it is a key intermediate in lipid metabolic pathways, whereas SM is the primary sphingolipid. To study the influence of acyl chain composition on lipid transport within the main membrane lipid class, we generated PC probes of varying acyl chain length and unsaturation degree as well as *sn*-1/*sn*-2 acyl regioisomers (Fig. 1b). We confirmed for all probes that incorporation of the bifunctional acyl chain did not alter lipid-specific membrane properties such as phase behaviour and membrane order in model membranes using spectroscopy[28,29] (Extended Data Fig. 1 and Supplementary Fig. 1). Moreover, we compared the metabolism of bifunctional probes in cells with an isotope-labelled native PC species (Extended Data Fig. 5). These data indicate that the bifunctional probes closely resemble endogenous lipids with the bifunctional chain most closely matching the properties of a palmitoleic acid (16:1) side chain.

Studying retrograde lipid transport requires a well-defined initial localization at the PM. We therefore loaded lipid probes into U2OS cells through a 0.5–4 min pulse of α-methyl-cyclodextrin-mediated lipid exchange from donor liposomes, which incorporates individual lipid molecules into the outer leaflet of the PM[30] (Fig. 1a and Extended Data Fig. 2). The PM integrity was not affected by the loading process (Extended Data Fig. 2) and quantification using MS showed that 1–3% of the total cellular lipidome was exchanged with bifunctional lipid probes, while the overall lipidome composition, including cholesterol content, remained essentially unaffected (Extended Data Figs. 1 and 5).

After removal of the liposome-containing loading solution, cells were kept at 37 °C for 0 min to 24 h before lipid photo-crosslinking, cell fixation, removal of non-crosslinked lipids and fluorescence labelling through click chemistry (Fig. 1a and Supplementary Information). The transport of bifunctional lipids was analysed by confocal imaging of the photo-crosslinked and fluorescently labelled lipids at all timepoints (Fig. 1a). Control samples without lipid probes or ultraviolet (UV) irradiation showed very low unspecific background labelling (Fig. 1c and Extended Data Figs. 1 and 2). Lipid imaging was complemented with quantitative shotgun lipidomics by ultra-high-resolution FT MS for each timepoint to quantify the metabolic conversions during the transport. To this end, we used the mass difference between the two nitrogen atoms of the diazirine functional group (28.0061 Da) and two $CH_2$ (28.0313 Da) groups to distinguish between bifunctional lipids and native lipids (Fig. 1d and Extended Data Fig. 5).

To quantify lipid transport, we assigned the lipid fluorescence signal to distinct organelle membranes by determining the colocalization of lipids with organelle markers for the PM, Golgi apparatus, ER, endosomes and mitochondria (Fig. 1e–g and Extended Data Fig. 4). Segmented probability maps were generated for every organelle marker using the pixel classifier approach of the Ilastik software package[31]. We then retrieved the organelle-specific lipid signal-intensity distributions from pixels that were unambiguously assigned to one organelle. On the basis of these distributions, the lipid signal was partitioned between organelles in regions where organelle masks overlapped (Fig. 1f,g, Extended Data Fig. 4 and Supplementary Information). Taken together, we developed a lipid-imaging pipeline that enables quantification of the interorganelle transport of distinct lipid species starting from the PM and correlation of lipid transport with time-dependent metabolic conversion of lipids observed by MS.

## Specific lipid transport is non-vesicular

Visual inspection of the lipid localization in confocal images revealed clear differences in transport kinetics between the lipid classes, and between individual species within the same lipid class (Fig. 2c and Extended Data Figs. 3 and 4). Overall, poly-unsaturated PC species, PA and PE exhibited a pronounced early localization in the ER, whereas saturated PC species and SM were retained much longer in the PM and subsequently showed persistent localization in endosomes (Fig. 2c, Extended Data Fig. 3 and Supplementary Fig. 2). These observations indicated that the kinetics of intracellular lipid transport differ both on the level of lipid classes and individual lipid species.

To understand whether the observed transport selectivity arises from differential sorting of lipid species during vesicular or non-vesicular transport (Fig. 2a), we fitted a kinetic model describing the main lipid transport routes to the lipid transport data (Fig. 2b). The model included retrograde vesicular transport through endocytosis from the PM through endosomes and the Golgi apparatus to the ER and the competing, non-vesicular route from the PM to the ER, as well as lipid exchange between the ER and mitochondria. Anterograde transport from the ER to the PM was captured by a summary rate encompassing both vesicular and non-vesicular modes. Two a priori assumptions were made: (1) retrograde transport of lipids from the PM directly to the ER is exclusively non-vesicular; and (2) retrograde lipid transport along the secretory pathway (PM to endosomes to Golgi to ER) is exclusively vesicular (Fig. 2b,c and Extended Data Figs. 6 and 7; details on model design and performance are provided in the 'Mathematical modelling' section of the Supplementary Information).

Kinetic models were fitted globally for each lipid species, except for PA(18:1/Y), which was transported too fast for the time resolution of the time-course experiments, to obtain interorganelle transport rate constants (Fig. 2d and Extended Data Fig. 4a). To assess the robustness of derived kinetic parameters, we compared different model versions featuring lipid-transport networks of increasing complexity and accounting for the change of absolute bifunctional lipid content, which was quantified by MS (Fig. 2b, Extended Data Fig. 6 and Supplementary Information (Mathematical modelling)). The obtained rate constants between different models were similar, indicating robustness of the results (Fig. 2d–f, Extended Data Fig. 7 and Supplementary Tables 1–4).

Comparison of the lipid-transport rate constants for retrograde transport from the PM to endosomes with retrograde transport directly from the PM to the ER revealed that non-vesicular trafficking was up to 11-fold (effect sizes up to 10 pooled s.d.) faster for all lipids compared with vesicular transport (Fig. 2f). Furthermore, the rate constants of non-vesicular trafficking showed significant variation between lipid classes and species (Fig. 2e,f and Extended Data Fig. 7). The fastest non-vesicular retrograde transport was found for PE, followed by polyunsaturated PC species and SM, while transport of saturated PC species was comparatively slow.

To determine the structural determinants of lipid species selective non-vesicular transport, we compared the rate constants obtained for six different PC species. Polyunsaturated PC species were transported up to sevenfold (effect sizes up to 8 pooled s.d.) faster through the non-vesicular route than saturated PC species, whereas PC species bearing the bifunctional fatty acid at the *sn*-2 position were transported up to twofold (3 pooled s.d.) faster than the corresponding regioisomers featuring the bifunctional fatty acid at the *sn*-1 position (Fig. 2f). These findings imply that while both the unsaturation degree and acyl chain positioning influenced the rates of non-vesicular lipid transport, unsaturation degree appeared to be the primary discriminating structural feature. In contrast to the high selectivity observed during non-vesicular transport, differences between the transport rate constants of the same PC species were smaller in the vesicular endosomal transport pathway and followed no obvious trends (Fig. 2d and Extended Data Fig. 7). Rate constants describing individual steps of vesicular transport in the anterograde direction (from the ER through the Golgi and endosomes to the PM, models 1b–3b) were less well identified (rate constants are shown in Supplementary Tables 1–4). To cross-check the lipid-imaging results, we conducted organelle fractionation experiments at two timepoints using PC(Y/16:0) and PC(Y/20:4) (Fig. 1b) as exemplary lipids with slow and fast retrograde transport kinetics. Consistent with the lipid-imaging results, we found that PC(Y/16:0) was enriched in the PM compared with in the ER–Golgi fraction at both the 4 min timepoint and 30 min timepoint, while PC(Y/20:4) was already equilibrated between PM and ER after 4 min (Supplementary Fig. 7).

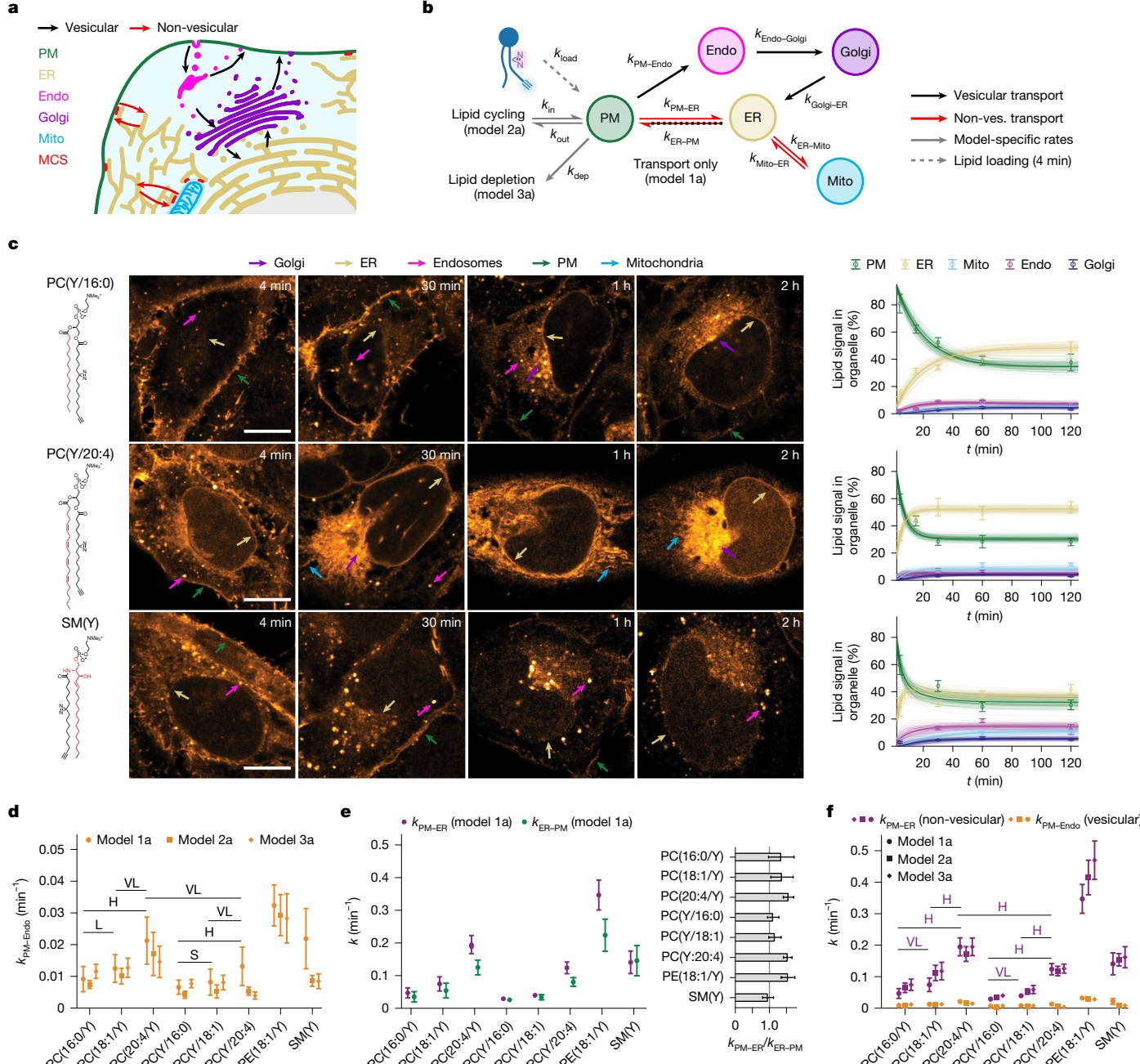

**Fig. 2 | Retrograde lipid transport occurs mainly through non-vesicular routes. a**, Schematic of the analysed cellular lipid-transport pipelines. MCS, membrane-contact site. **b**, Kinetic models for quantifying lipid transport from fluorescence microscopy and MS data. Non-ves., non-vesicular. **c**, Kinetics of lipid transport for PC(Y/16:0), PC(Y/18:1), PC(Y/20:4), SM(Y) and the corresponding model 1a fits. Unique structural elements of individual lipids are highlighted in red. Scale bars, 10 μm. Images were brightness–contrast adjusted to enable comparison of lipid distributions at different timepoints. Kinetics were constructed from five independent timepoints. Data are mean ± s.d. Individual *n* values are provided as source data. **d**, Comparison of rate constants describing retrograde vesicular transport from the PM to endosomes (models 1a–3a shown). **e**, Comparison of rate constants describing retrograde non-vesicular transport from the PM to the ER and total transport in the anterograde direction. **f**, Comparison of rate constants describing retrograde vesicular transport from the PM to endosomes and retrograde non-vesicular transport from the PM to the ER for all analysed lipid probes. The mean ± s.d. of the fitted rate constants was calculated from 100 MC model runs. Pairwise effect-size values are shown for PC species comparison of model 1a rates. Values for all lipids pairs are given in Extended Data Fig. 7. For **d** and **f**, Cohen's *d* values are indicated; very small (VS), >0.01; small (S), >0.20; medium (M), >0.50; large (L), >0.80; very large (VL), >1.20; huge (H), >2.00.

To test the predominant role of the non-vesicular route in retrograde transport, we inhibited vesicular trafficking using brefeldin A, which blocks COPI coat formation at Golgi membranes, and wortmannin, which broadly affects endosome formation and trafficking, and quantified the distribution of SM(Y), PE(18:1/Y), PC(Y:16:0) and PC(Y/20:4) between endolysosomes and the ER. Control measurements of transferrin uptake confirmed that endocytosis of protein cargoes was strongly reduced in cells treated with wortmannin or brefeldin A (Fig. 3b and Supplementary Fig. 3). We found that lipid transport to the ER was unaffected by blocking vesicular transport for all lipids after 4 min. After 30 min, only SM(Y) accumulated in perinuclear endosomes in wortmannin-treated cells, which is consistent with

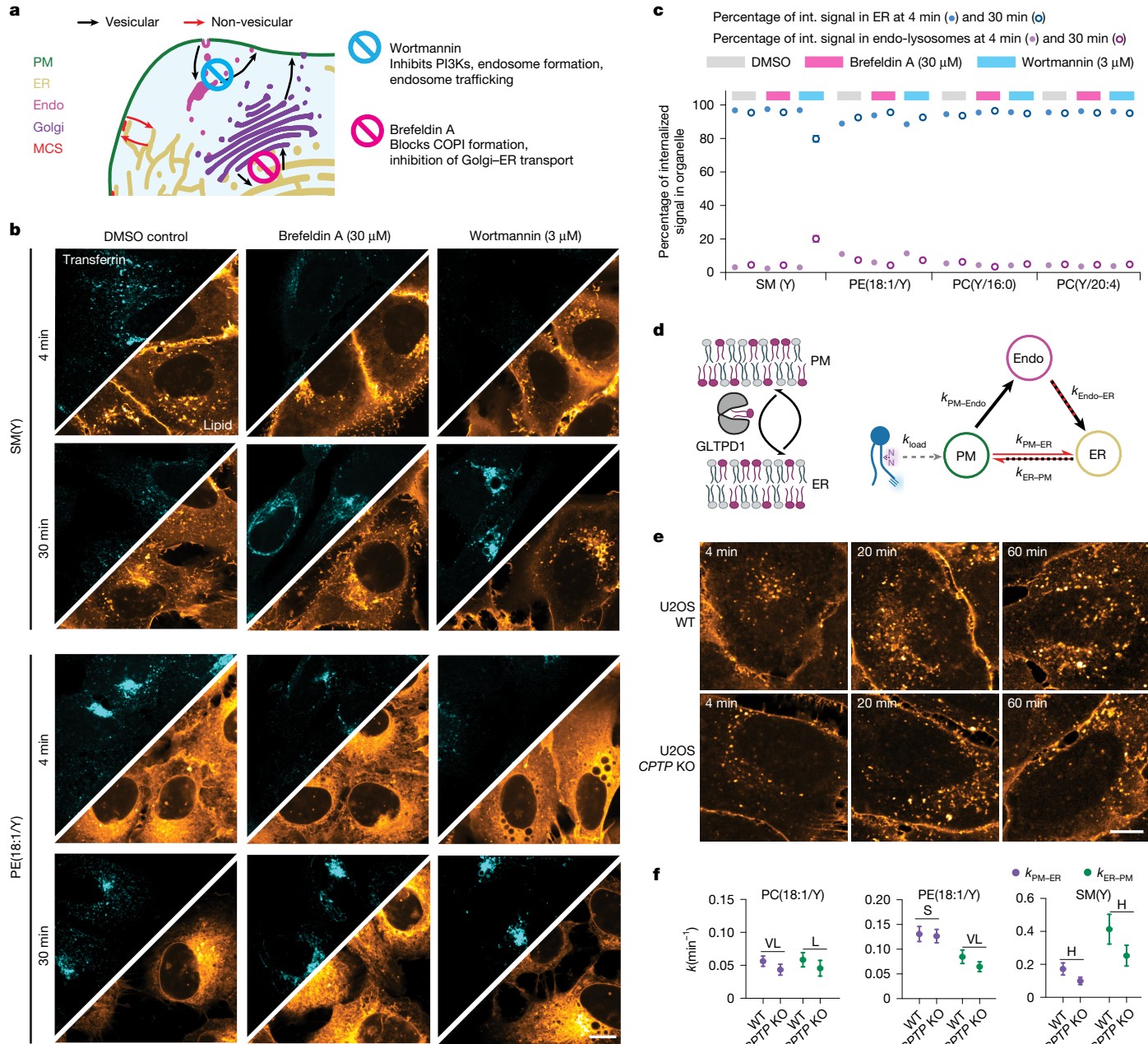

**Fig. 3 | Pharmacological and genetic perturbations confirm non-vesicular transport as the primary retrograde lipid-transport route. a**, Inhibition of vesicular trafficking between the Golgi and the ER (brefeldin A) and of endosome formation and trafficking (wortmannin). **b**, Transferrin uptake and localization and lipid localization in control cells and drug-treated cells at 4 min and 30 min after lipid loading. Inhibition of vesicular trafficking does not affect retrograde lipid transport. Scale bar, 10 μm. Transferrin images are shown at identical settings; lipid images were brightness–contrast adjusted to enable comparison of lipid distributions at different timepoints. **c**, Quantification of the lipid distribution between the ER and endolysosomes after treatment with DMSO (grey), brefeldin A (pink) or wortmannin (light blue). Data are mean ± s.d. Individual *n* values are provided as source data. Int., internalized. **d**, Schematic of GLTPD1-mediated lipid transfer and kinetic model used to assess the effects of *CPTP* KO on non-vesicular transport of SM(Y). **e**, Comparison of SM(Y) time-course experiments in *CPTP*-KO and U2OS WT cells. Scale bar, 10 μm. **f**, Rate constants for retrograde and anterograde PM–ER lipid transport for PC(18:1/Y), PE(18:1/Y) and SM(Y). Kinetics were constructed from six independent timepoints (mean ± s.d.) containing 5 field of views each with 5–10 cells. The mean ± s.d. was calculated from 100 MC model runs. Statistical analysis was performed using pairwise effect-size tests; Cohen's *d* values are indicated.

previous observations that anterograde transport of SM involves vesicular steps[32], whereas all other lipids were unaffected.

To elucidate the non-vesicular retrograde transport mechanism of SM(Y), we generated a cell line with knockout (KO) of *CPTP*, which encodes GLTPD1, a cytoplasmic cup-shaped lipid-transfer protein suggested to be specific for SM, with ceramide phosphate as a less preferred cargo[33]. Quantification of transport between the PM and the ER of PC(18:1/Y), PE(18:1/Y) and SM(Y) revealed that the *CPTP* KO had

the strongest effect on retrograde, non-vesicular transport of SM(Y), which was significantly reduced (0.57-fold of WT, 2.3 pooled s.d.). Smaller changes were observed for PC(18:1/Y) (0.77-fold of WT, 1.6 pooled s.d.) and PE(18:1/Y) (0.96 fold of WT, 0.27 pooled s.d.) (Fig. 3e,f and Supplementary Fig. 5). These results confirm the modelling result that retrograde transport of structural lipids is mostly non-vesicular. Taken together, we find that retrograde non-vesicular lipid transport is both faster and more selective than vesicular transport.

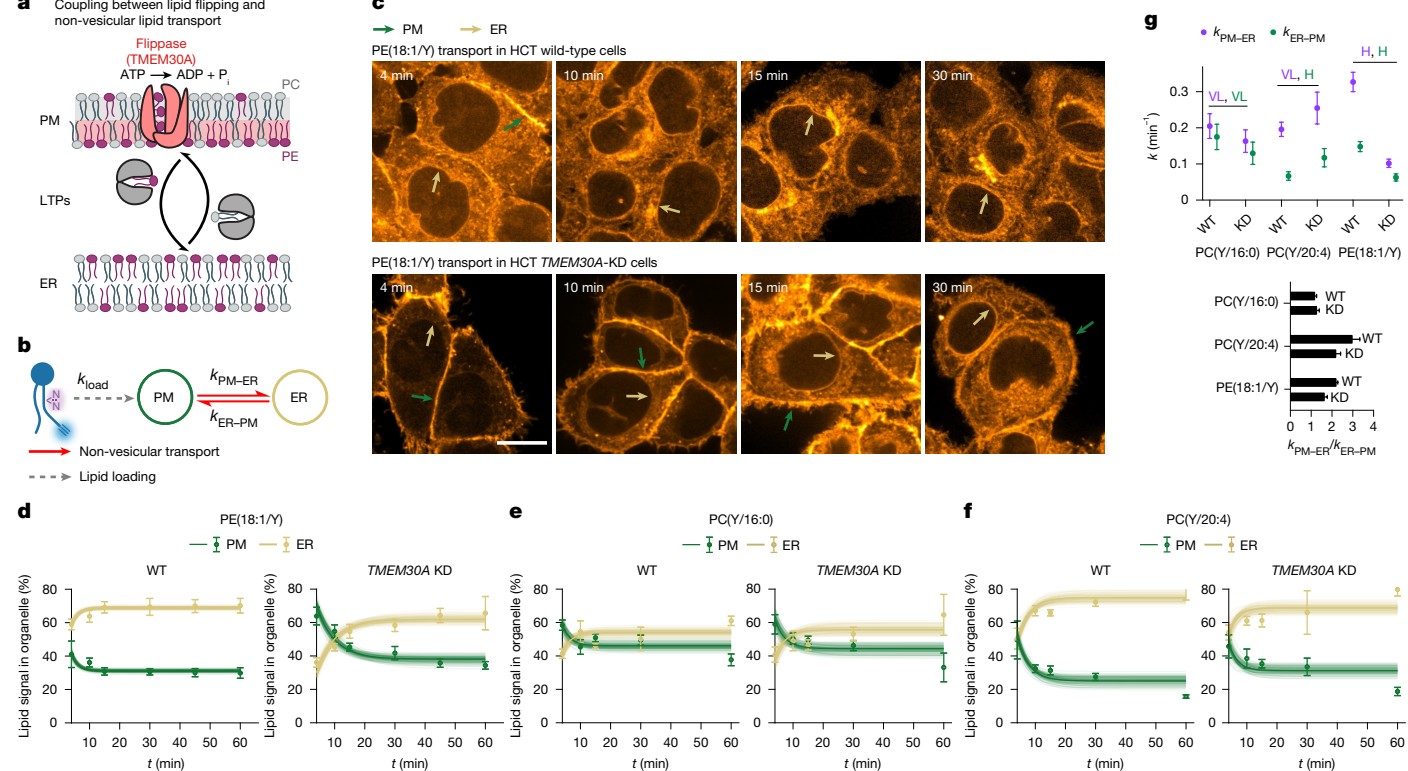

**Fig. 4 | Genetic perturbation experiments confirm the involvement of flippases in species-specific directional lipid transport. a**, Schematic of lipid trans-bilayer movement (lipid flipping) and non-vesicular lipid transport by lipid transfer proteins. $P_i$, inorganic phosphate. **b**, Kinetic model for the exchange of lipids between the PM and the ER. **c**, Comparison of time-course experiments for PE(18:1/Y), showing that lipid internalization dynamics are slower in HCT116 *TMEM30A*-KD cells than in HCT116 wild-type cells. The coloured arrows indicate lipid localization in different membrane types (green, PM; yellow, ER). Scale bar, 10 μm. Images were brightness–contrast adjusted to facilitate comparison of intracellular lipid localizations. **d**–**f**, Quantification of PE(18:1/Y) (**d**), PC(Y/16:0) (**e**) and PC(Y/20:4) (**f**) internalization kinetics and model fits. Kinetics were constructed from five independent timepoints. Data are mean ± s.d. Individual *n* values are provided as source data. **g**, Rate constants and quasi-equilibrium constants for retrograde and anterograde PM–ER lipid transport for PC(Y/16:0), PC(Y/20:4) and PE(18:1/Y). The mean ± s.d. was calculated from 100 MC model runs. Statistical analysis was performed using pairwise effect-size tests; Cohen's *d* values are indicated.

## Flippases drive transport selectivity

Next, we assessed the implications of the predominant non-vesicular lipid transport for the steady-state lipid distributions between organelles. The highest fraction of lipid signal in the PM at steady state was found for SM, followed by the saturated PC(Y/16:0) species, whereas polyunsaturated PC and PE localized preferentially to the ER (Fig. 2c and Extended Data Fig. 6c–e (top)). An analysis of quasi-equilibrium constants for lipid exchange between the PM and ER gave a very similar result (Fig. 2e (left)). These findings are consistent with the known lipid concentration gradients between organelles[3] and imply directional, non-vesicular lipid transport in cells. How different PC species, SM and PE can be directionally transported through non-vesicular pathways is not well understood. While directional transport of cholesterol, PA and phosphatidylserine (PS) against concentration gradients may be driven by phosphatidylinositol 4-phosphate counter-transport[8,34], it is unclear which process provides the energy for directional transport of other lipids, in particular through bridge-like lipid transfer proteins[35,36].

One attractive mechanism could be the coupling of passive non-vesicular transport to active trans-bilayer flipping of lipids between membrane leaflets catalysed by P4-ATPases, either directly[8] or indirectly through the use of the transmembrane lipid concentration gradient by scramblases[37–39]. To test this hypothesis, we investigated the role of lipid flippases in PE transport from the PM to the ER, a lipid that is known to be enriched in the inner leaflet of the PM through flippase activity[40] (Fig. 4a). We genetically knocked down (KD) *TMEM30A*

(Extended Data Fig. 8k,l and Supplementary Fig. 4), which encodes the common subunit of PM flippases that move aminophospholipids to the inner PM leaflet[41,42] (Fig. 4a) in HCT116 cells. We found that PE(18:1/Y) was transported threefold (11 pooled s.d.) more slowly in *TMEM30A*-KD cells than in WT cells (Fig. 4c,d,g). *TMEM30A*-KD cells had a significantly lower $k_{PM-ER}/k_{ER-PM}$ ratio (1.6 ± 0.1 versus 2.2 ± 0.1), indicating an altered steady-state distribution, with PE being more strongly enriched in the PM when lipid flipping is perturbed. These results provide direct evidence that ATP-dependent lipid flipping and non-vesicular transport of PE from the PM to the ER are coupled.

To assess whether the observed effect was specific for PE, we also quantified the transport of two PC species (PC(Y/16:0) and PC(Y/20:4)) in *TMEM30A*-KD cells. We found that transport of PC(Y/20:4) was slightly faster (1.3 fold, 1.2 pooled s.d.) and transport of PC(Y/16:0) was slightly slower in *TMEM30A*-KD cells (0.8 fold, 1.7 pooled s.d.) than in wild-type cells (Fig. 4e,f and Supplementary Fig. 6), and that the steady-state distribution of PC(Y/20:4) was shifted towards the PM, whereas the steady-state distribution of PC(Y/16:0) did not change. While these changes indicate a differential role of TMEM30A in the transport of distinct PC species, the overall effect on PC transport was minor compared with the impact on PE(18:1/Y) transport.

Taken together, the combination of genetic perturbations with lipid imaging indicates that active, species-selective flipping of lipids between PM leaflets in conjunction with selective transport by lipid-transfer proteins contributes to the establishment and maintenance of differential organelle membrane compositions despite continuous lipid exchange.

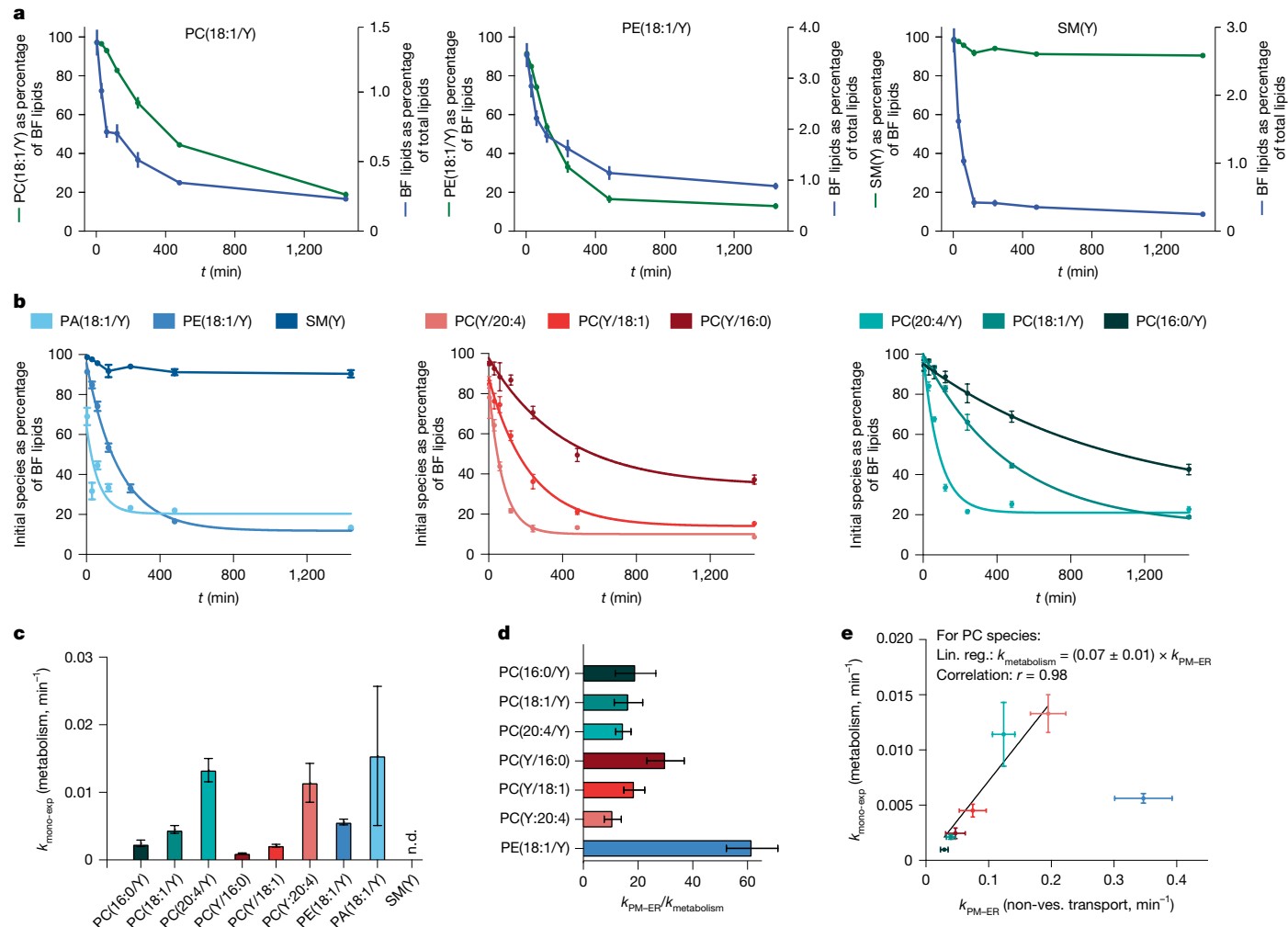

**Fig. 5 | Lipid metabolism is approximately one order of magnitude slower than lipid transport. a**, Bifunctional lipid retention and turnover. The fraction of initially supplied species as the percentage of all bifunctional (BF) lipids and the fraction of bifunctional lipids of the total lipidome for PC(18:1/Y), PE(18:1/Y) and SM(Y). **b**, The fraction of initially supplied lipid probe as the percentage of the bifunctional lipidome as a proxy for the speed of lipid metabolism. The solid lines indicate mono-exponential fits. SM(Y) data were not fitted as very little interconversion was observed; instead, a linear interpolation is shown. For **a** and **b**, data are mean ± 95% confidence intervals of three biological repeats containing two technical replicates each. **c**, Comparison of the determined mono-exponential rate constants for the metabolism of

individual lipid species. The error bars show the s.e. of the mono-exponential fit. n.d., not determined. **d**, Comparison of transport and metabolic rate constants shows that lipid transport is at least one order of magnitude faster. Error bars were obtained by error propagation. **e**, Lipid transport and metabolism rate constants are highly correlated for PC species despite a clear time-scale separation. The plot shows the $k_{\text{mono-exponential}}$ calculated from fitting the data to a single exponential decay versus the mean of $k_{\text{PM–ER}}$ from 100 MC runs. The error bars show the s.e. (metabolic rate constants) and s.d. (transport rate constants), calculated from 100 MC model runs. Lin. reg., linear regression.

## Lipid-transport speed exceeds metabolism

To assess the relative importance of lipid metabolism and lipid-transport processes for the maintenance of organelle lipid compositions, we next compared transport kinetics to lipid conversion kinetics. We monitored turnover of bifunctional lipid probes using ultra-high-resolution FT lipid MS (Extended Data Figs. 8–10 and Supplementary Information (Lipid metabolism)). To obtain a measure of global lipid metabolism, we determined how fast the bifunctional acyl chain of a respective lipid species is redistributed to other lipids by calculating the fraction of the initially supplied bifunctional species with respect to the total abundance of bifunctional lipids (Fig. 5a and Supplementary Fig. 8). Moreover, we determined the combined abundance of all bifunctional lipids in the overall lipidome as a measure of probe retention. As expected, both the abundance of the initially supplied species and the total bifunctional lipid content declined over time due to ongoing metabolism, lipid secretion and cell growth. For all

bifunctional glycerophospholipid probes, the kinetics of lipid conversion and overall bifunctional lipid depletion were similar, pointing to lipid metabolism as the primary mechanism for falling bifunctional lipid content (Fig. 5a and Supplementary Fig. 8). By contrast, bifunctional SM(Y) was quickly depleted from the overall lipidome while constituting over 90% of the bifunctional lipidome at all timepoints (Fig. 5a), which can either be explained by a rate-limiting first step of SM(Y) metabolism or alternative mechanisms of probe depletion, such as secretion.

Among glycerophospholipids, we found that the primary metabolic process for all bifunctional PC species was fatty acid cycling within the cellular PC pool. Moreover, neutral lipids and PE were generated to a certain amount, reaching a combined abundance of 20–40% of the bifunctional lipid pool, whereas incorporation into other glycerophospholipids, such as PS and PI, was negligible (Extended Data Figs. 8–10). PE(18:1/Y) and PA(18:1) probes were converted mainly to PC and neutral lipids (Extended Data Figs. 8–10). Furthermore, we observed that the

initially loaded glycerophospholipid species constituted between 10 and 40% of the bifunctional lipidome at later timepoints, indicating that a steady-state distribution resembling the overall glycerophospholipid composition had not been reached after 24 h.

To test whether the observed trends were affected by the chemical modifications of the probes or are generally indicative of the metabolic fate of PM lipids, we compared the metabolism of bifunctional PCs containing oleic and palmitic acid side chains to an isotope-labelled PC(18:1/16:0[13C]) probe, which was delivered to the PM using the same protocol that was used for the bifunctional lipids. The overall metabolism of the isotope-labelled PC closely resembled the palmitate-containing bifunctional probes (Extended Data Fig. 5). We therefore conclude that (1) the observed metabolic trends are due to metabolic bias of PM-resident lipids and (2) that lipids bearing the bifunctional fatty acid are most closely resembling native counterparts bearing a monounsaturated fatty acid in its place.

To compare the kinetics of lipid metabolism and transport, we obtained apparent rates of bulk metabolism by fitting a monoexponential model. We found an order of magnitude difference between the apparent conversion rate constants of lipid species with values ranging from $k_{met} = 0.001$ min$^{-1}$ to $k_{met} = 0.015$ min$^{-1}$ (Fig. 5b,c). Polyunsaturated PC species were metabolized faster than monounsaturated and saturated PC species; PA and PE were converted faster than the corresponding PC species with the same fatty acid composition, whereas SM was largely stable. By comparing the apparent rate constants of bifunctional lipid probe conversion to the non-vesicular transport rate constants from the PM to the ER, we found that metabolism is slower than transport by a factor of 10–60 for all investigated probes (Fig. 5c,d).

Notably, we found that transport and metabolism rate constants are highly correlated for the PC species despite a pronounced time-scale separation (Fig. 5e). As most PC lipid molecules are metabolically converted after the steady-state distributions are reached, this cannot be explained by delayed access of enzymes to bulk lipids within organelle membranes. The biophysical properties of the respective lipids could be directly responsible, for example, it could be caused by highly correlated activation energies for the transfer of lipids from the bulk membrane into the binding pockets of enzymes and lipid-transfer proteins, respectively. Alternatively, metabolic conversion could be directly coupled to transport, for example, by lipid substrate handover from a lipid-transfer protein to a lipid-metabolizing enzyme.

Taken together, we find that lipid transport is much faster than lipid metabolism. This finding suggests that the differential steady-state distribution of lipid species in the organelles of the secretory pathway mainly results from selective non-vesicular transport rather than local metabolic conversion.

## Metabolic channelling of acyl chains

On short time scales, lipid sorting was found to be dominated by non-vesicular lipid transport. To assess whether cases exist in which lipid metabolism controls cellular lipid distribution, we analysed the later timepoints of the time-course experiments. Quantification of lipid-imaging data revealed that bifunctional lipids derived from PC regioisomers bearing the same fatty acids with the bifunctional fatty acid either at the sn-1 or sn-2 position differently accumulated in lipid droplets (Extended Data Figs. 3 and 11a,b). Supplying sn-1-modified PCs resulted in approximately twofold higher accumulation of bifunctional lipids in lipid droplets after 24 h compared to sn-2 modified PCs, suggesting different conversion pathways for the regioisomers (Extended Data Fig. 11b). As the entire lipid droplet intensity distribution was shifted to higher intensities after supplying sn-1-modified PCs, this is highly likely due to differential metabolism at all lipid droplets as opposed to specialized subpopulations (Extended Data Figs. 10 and 11b).

The complementary MS data showed that the production of bifunctional cholesterol ester (CE) was up to sevenfold increased starting from the sn-1-modified PCs compared with the respective sn-2 regioisomers (Extended Data Figs. 10 and 11c). By contrast, conversion into a wide range of triacylglycerols (TAGs) occurred with similar kinetics and abundance for all lipid probes (Extended Data Figs. 10 and 11c), while native TAGs remained unchanged (Extended Data Fig. 5). Together, this suggests that the observed difference in bifunctional lipid accumulation resulted from differential CE formation rates.

The observed TAG patterns suggest a reaction sequence of bifunctional fatty acid cleavage, generation of bifunctional acyl-CoA and subsequent incorporation into TAGs by DGAT2 on lipid droplets (Extended Data Fig. 11f). The alternative route through PA and DAG is incompatible with the obtained data, as supplying PA(18:1/Y) resulted in the rapid formation of a single TAG species, while other species were generated much later (Extended Data Fig. 11d). As canonical CE and TAG biosynthesis routes both involve the same precursor, bifunctional acyl-CoA, the differential PC regioisomer rates can only occur if the bifunctional fatty acid of the sn-1 bifunctional PC isomer is preferentially channelled towards CE, for example, through a spatially coupled enzyme cascade comprising a sn-1-specific phospholipase, an acyl-CoA synthetase and a sterol O-acyl transferase (Extended Data Fig. 11e,f).

Taken together, our data indicate that cellular PC metabolism generates spatially separated pools of identical lipid metabolites for the biosynthesis of TAG and CE, respectively. Thus, our data provide evidence for metabolic bias[43] of specific lipid species within the lipid-storage pathway and demonstrate that subcellular accumulation of neutral lipids can be regulated by species-specific metabolism despite slower kinetics than lipid-transport processes.

## Discussion

Here we introduce a pipeline to profile the transport and metabolism of individual lipid species in the organelle system of eukaryotic cells with high spatiotemporal resolution. The kinetic comparison of transport and metabolism between different lipid species enabled us to address the fundamental question how cells maintain organelle-specific lipid compositions. We found that small variations in lipid chemical structure strongly influenced the kinetics of non-vesicular lipid transport and metabolism, implying a high degree of selectivity at the level of individual lipid species. This is illustrated by up to sevenfold differences in non-vesicular transport speed between individual PC species that differ in unsaturation degree and acyl chain positioning. Conversely, a similar degree of selectivity was not observed during retrograde vesicular lipid transport of lipids mediated by endocytosis. On the basis of our quantification, we estimate that between 85 and 95% of PM lipids are transported through non-vesicular routes rather than through endocytosis in the retrograde direction (Extended Data Fig. 7), as non-vesicular lipid transport was found to be significantly faster than membrane trafficking.

Together with earlier data indicating that trafficking of the bulk phospholipids PC and PE in the anterograde direction occurs mainly through non-vesicular pathways[44,45], our results imply that organelle membrane lipid compositions are maintained mainly through a combination of fast, species-specific, non-vesicular lipid transport. We found that specific non-vesicular lipid transport and ATP-dependent lipid flipping by P4ATPases are coupled, suggesting P4ATPases or lipid asymmetry more generally as a likely source of the energy required for directional lipid transport. This finding provides a possible explanation for the much higher rates of lipid exchange observed between cellular membranes than observed for in vitro membrane models[46].

Our lipid-imaging technique relies on photochemical cross-linking for generating covalent lipid–protein conjugates. Lipid–protein interactions typically have lifetimes in the nanosecond–microsecond range[40], which implies that, during a 1–10 s photo-crosslinking pulse, lipids switch thousands of times between protein-associated and free

states. This fast sampling of membrane proteins by the lipid probe is expected to minimize any potential protein interaction bias in the final cross-linked lipid distributions.

There are two main limitations with our current approach to measuring lipid transport. First, our data indicate that the C16 bifunctional fatty acid used in this study resembles a native monounsaturated fatty acid (C16:1). It is therefore unlikely that fully saturated lipids can be faithfully mimicked using the bifunctional fatty acid that we used in the current study. To an extent, it may be possible to mitigate this by placing the diazirine moiety at either end of fatty acid rather than in the centre of the aliphatic chain[47]. Second, by delivering probes to the outer PM leaflet, we obtain detailed information on the early steps of retrograde transport, whereas lipid exchange between organelles further removed from the PM in the context cellular lipid transport is captured less well. In the future, this can be addressed by alternative delivery protocols or organelle-specific trifunctional lipid probes, which equip bifunctional lipids with an organelle-directing photocaging group and offer much higher flexibility with regard to intracellular starting points at the cost of lengthier synthetic routes[11,13,14].

Taken together, our findings suggest that non-vesicular lipid transport has a key role in the maintenance of organelle identity. Combining our approach with genetic interventions will shed light on the molecular mechanisms that underpin species-selective lipid transport and metabolism. We anticipate that this work will have a major impact for revealing the functions of lipids in cell biology.

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

## Reporting summary

Further information on research design is available in the Nature Portfolio Reporting Summary linked to this article.

## Data availability

The complete lipid flux dataset can be interactively accessed online (https://doi.org/21.11101/0000-0007-FCE5-B), and all original data can be downloaded (https://doi.org/21.11101/0000-0007-FCE4-C and https://doi.org/10.6019/S-BIAD1695). A demo dataset is available online (https://doi.org/10.17617/3.BRSGLA). Source data are provided with this paper.

## Code availability

The Python code, the Ilastik models and the MATLAB scripts are provided online (https://doi.org/21.11101/0000-0007-FCE3-D). Custom code was used to perform the image analysis, to calculate the intracellular lipid distribution in the segmented images, to analyse the MS data and to model the transport of lipid in cells.

**Acknowledgements** A.N. acknowledges financial support by the European Research Council (ERC) under the European Union's Horizon 2020 research and innovation program (grant agreement no. GA 758334 ASYMMEM and AURORA). A.N., A.H. and A.S. acknowledge financial support by the Deutsche Forschungsgemeinschaft (DFG) through the TRR83 consortium. K.B., A.H. and A.N. were supported by the Volkswagen Foundation, project number A133289. This research was supported by an Allen Distinguished Investigator Award and a Paul G. Allen Frontiers Group advised grant of the Paul G. Allen Family Foundation to A.N. and A.H.; M.H. and R.Š. acknowledge the financial support provided by the Advanced Multiscale Materials for Key Enabling Technologies project, supported by the Ministry of Education, Youth and Sports of the Czech Republic, project no. CZ.02.01.01/00/22_008/0004558, co-funded by the European Union. M.S. is supported by the ELISIR program of the EPFL School of Life Sciences and acknowledges funding from the Swiss National Science Foundation (SNSF grant IC00IO-227891). We thank C. Nadler for discussions and providing liposome preparations during the initial lipid-imaging experiments and M. Marass for feedback during manuscript preparation; the staff at the following services and facilities at MPI-CBG Dresden for their support: Protein Expression Facility, Mass Spectrometry Facility, Scientific Computing Facility, Genome Engineering Facility, Organoid and Stem Cell Facility and the Light Microscopy Facility; and J. Peychl, B. Schroth-Diez, S. Bundschuh and H. K. Moon for support and expert advice on fluorescence microscopy and data visualization in the online lipid flux atlas.

**Author contributions** K.B., M.S., C.J.L., A.L. and A.N. synthesized lipid probes. A.N. and A.H. developed the initial lipid-delivery and imaging protocol. J.M.I.-A developed the final, optimized lipid-imaging protocol, optimized colocalization experiments with organelle markers and developed the image analysis pipeline. J.M.I.-A., K.B. and K.C.C. performed lipid-imaging experiments. K.S., V.G. and K.C.C. performed lipidomics experiments. K.S., J.M.I.-A., K.C.C., A.S. and A.N. analysed MS data. J.M.I.-A., K.C.C., K.B., A.N. and A.H. analysed fluorescence microscopy data. R.Š., P.R. and M.H. performed and analysed biophysical characterization of model membrane systems. H.M.L., M.S., K.P.-G. and P.B. contributed to lipid-imaging protocol development. P.B. performed protein purification of the eGFP–LactC2 PS-sensor. P.B. and H.M.L. characterized the HCT116 *TMEM30A*-KD line and assessed lipid-transport changes. B.D. performed kinetic modelling. A.N. prepared figures with contributions from J.M.I.-A., A.H., B.D., K.C.C., K.B., K.S. and P.B.; A.N. and A.H. wrote the manuscript with contributions from J.M.I.-A., K.B. and M.S.; J.M.I.-A., A.H. and A.N. designed the project. All of the authors read and commented on the manuscript.

**Funding** Open access funding provided by Max Planck Society.

**Competing interests** A.N. and J.M.I.-A. have received a proof-of-concept grant from the ERC to explore the commercial potential of the lipid-imaging methodology. The other authors declare no competing interests.

**Additional information**
**Correspondence and requests for materials** should be addressed to Alf Honigmann or André Nadler.

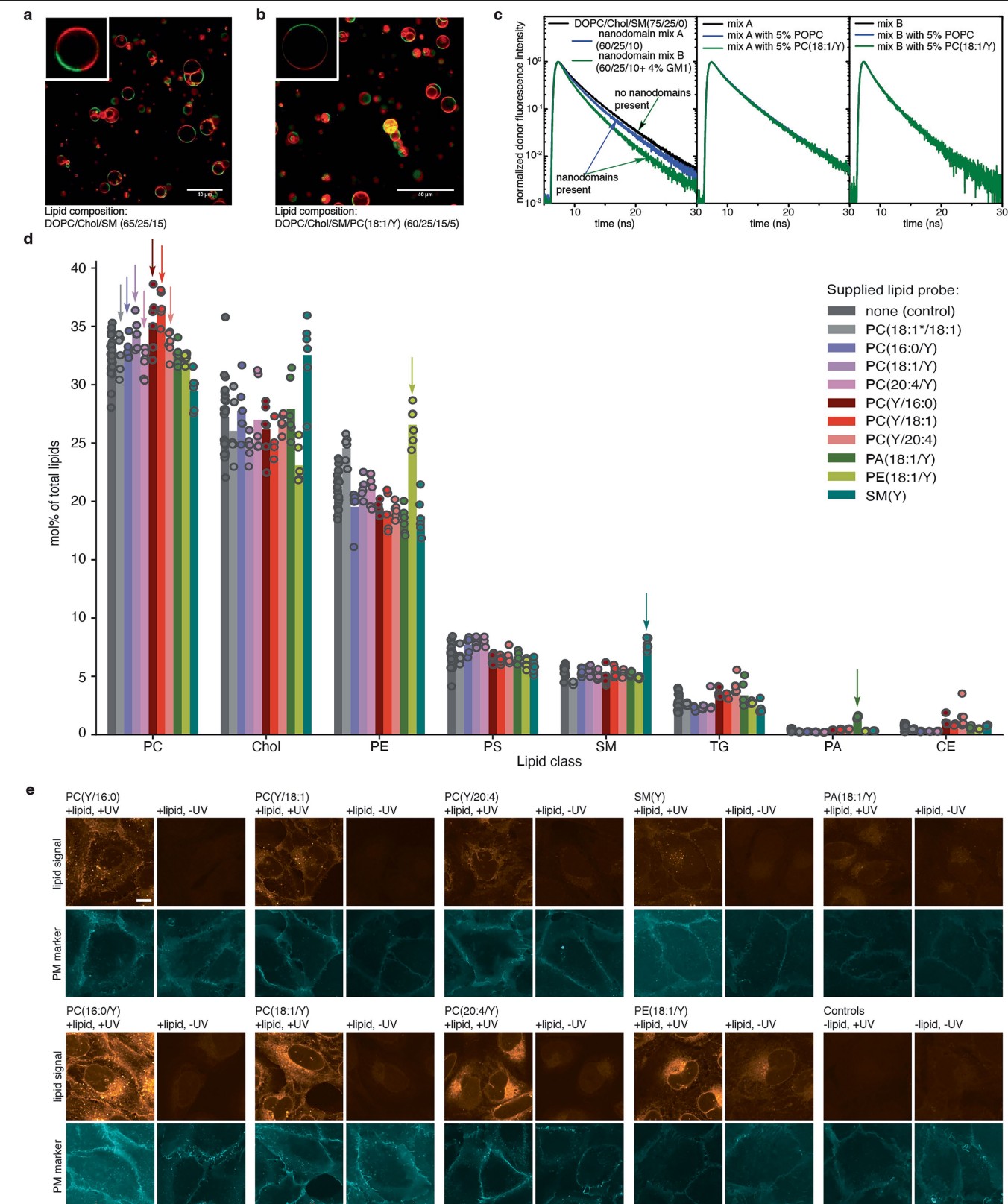

**Extended Data Fig. 1** | See next page for caption.

**Extended Data Fig. 1 | Biophysical characterization of bifunctional lipid containing model membranes, lipidome assessment after bifunctional lipid loading and lipid imaging signal comparison for all probes.**
**a, b**. Formation of liquid ordered $L_o$ (stained by Bodipy-FL-GM$_1$; green) and liquid disordered $L_d$ (stained by DiD; red) microdomains is unaffected by replacing 5% of DOPC content with PC(18:1/Y) in giant unilamellar vesicles GUVs. Scale bars: 40 μm. **c**. Formation of ganglioside nanodomains leading to faster deexcitation of Bodipy-FL-GM$_1$ donors via FRET is unaffected by replacing 5% of POPC content with PC(18:1/Y) in GUVs. Scale bar is 40 μm. **d**. Comparison of lipidome composition directly after lipid loading bifunctional lipid probes (4 min timepoint) with control lipidome. Arrows indicate supplied lipid type. Bars show the mean of 3 biological repeats containing 2 technical replicates each. **e**. + UV lipid signal vs -UV lipid signal for all probes, 30 min timepoint shown. Note: The high intensity in -UV conditions for PE(18:1/Y) is explained by the fact that PE can be chemically fixed with formaldehyde due to its primary amine group, which is not the case for the other lipids. Scale bar: 10 μm, all images shown at the same magnification.

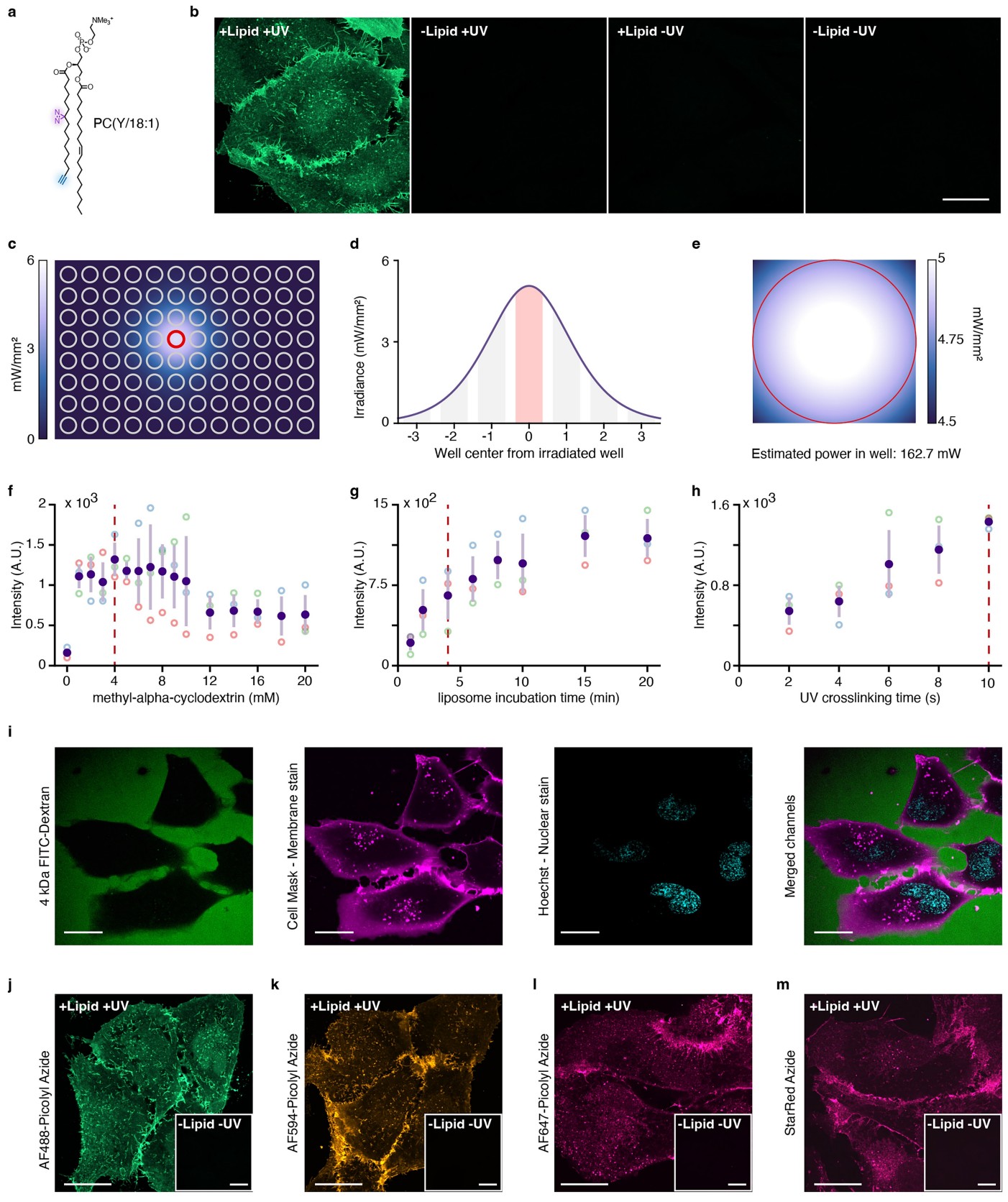

**Extended Data Fig. 2 | See next page for caption.**

**Extended Data Fig. 2 | Optimization of the lipid imaging protocol.**
**a**. Structure of PC(18:1/Y) used for protocol optimization. **b**. Representative imaging results using optimized lipid loading, crosslinking and click chemistry conditions. Images from experiment and control samples are adjusted to the same intensity. Scale bar: 20 μm, all images shown at the same magnification. **c-e**. Characterization of UV illumination in the 96-well plate format used for this study. **f-h**. Optimization of lipid loading and crosslinking conditions. Dashed red lines indicate chosen conditions. Dashed red lines indicate the selected condition for further experiments. Cells were fixed directly after incubation with the loading solution. For **f** and **h** samples were incubated with liposomes for 4 min. **f-h**. Mean and 68% CI of 3 independent experiments. **i**. Lipid loading does not compromise cell membrane integrity as demonstrated by exclusion of 4 kDa FTIC-Dextran from cell interior. Lipid loading solution was incubated for 10 min before imaging. **j-m**. Lipid signal visualization using different Picolyl-Azide dyes. AF594-Picolyl-Azide was used for this study. Scale bars: 20 μm.

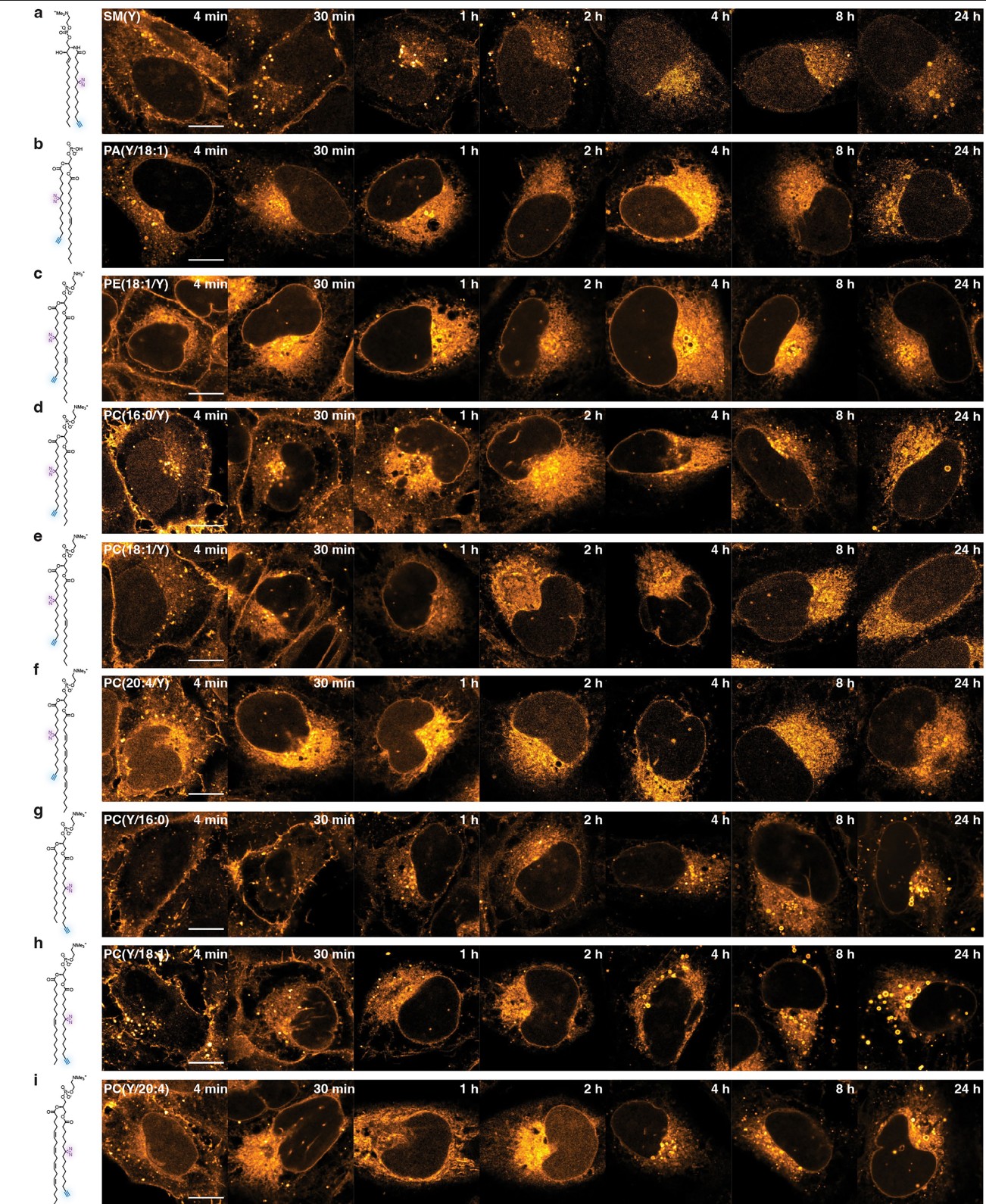

**Extended Data Fig. 3 | a-i Lipid transport time courses for all probes and timepoints.** Representative images for lipid transport time course experiments. Scale bars: 10 μm, all images shown at the same magnification. Images are brightness-contrast adjusted to facilitate comparing intracellular lipid localization. The full dataset 3D dataset including marker channels can be accessed on https://lipidimaging.org/. N = 3 for each lipid and timepoint.

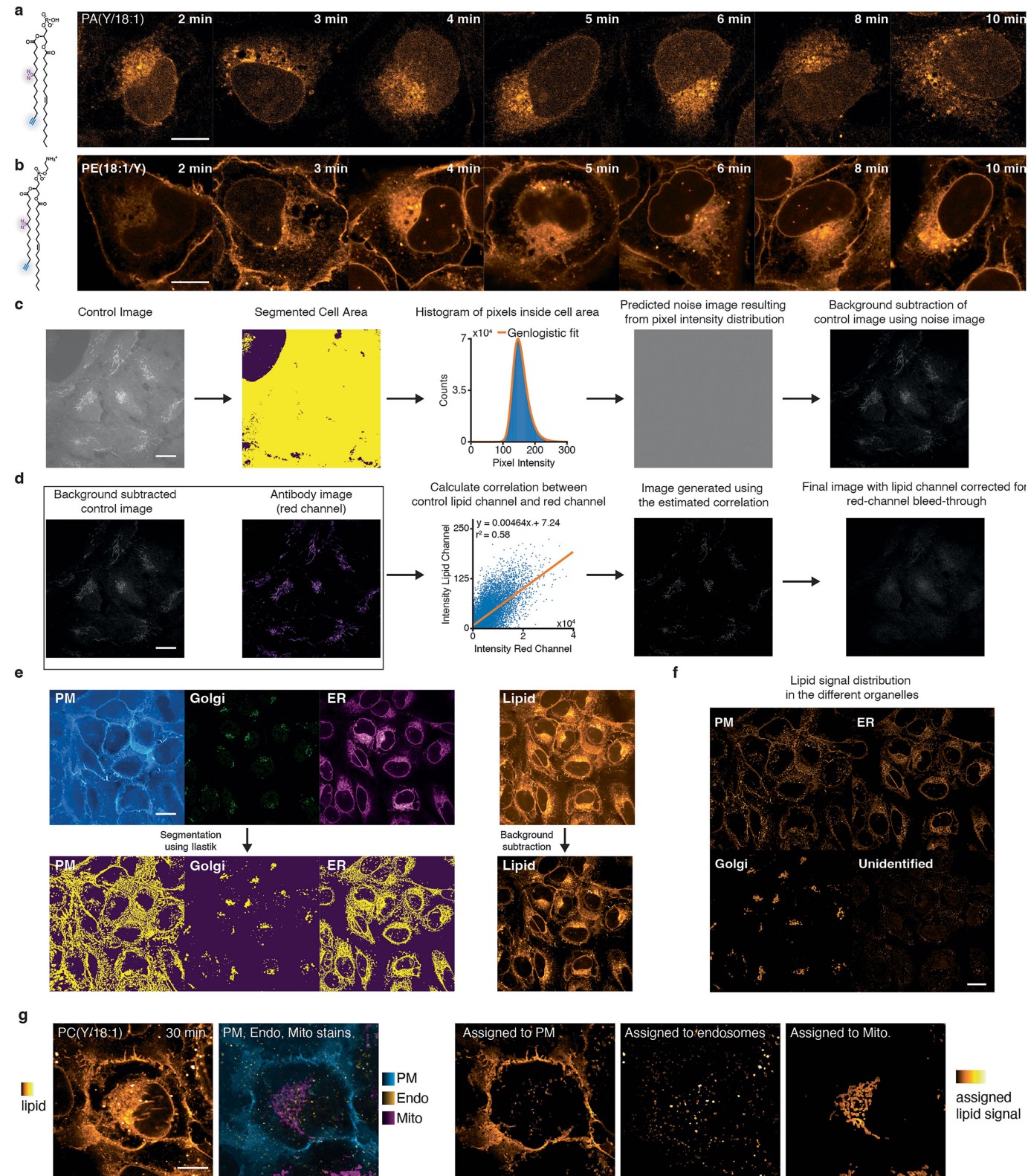

**Extended Data Fig. 4** | See next page for caption.

**Extended Data Fig. 4 | High time-resolution time courses for PE (18:1/Y) and PA(18:1/Y) and image analysis pipeline. a, b**. Representative images for lipid transport time course experiments at higher time resolution using PA(18:1/Y) (**a**) and PE(18:1/Y) n = 3 (**b**). Scale bars: 10 μm. Images are brightness-contrast adjusted to facilitate comparing intracellular lipid localization. n = 3. **c, d**. Background subtraction strategy. For most data, background was removed using a predicted noise image derived from control images (+ UV, -lipid). In cases where a AF647-Tom20 antibody was used as a mitochondrial stain, we observed a faint mitochondrial signal in the AF594 (lipid) channel in control conditions. For the corresponding +lipid images we estimated the extent of the bleed-through signal by determining the correlation between the mitochondrial signal in the marker channel and the lipid channel, using these parameters to generate an image for the expected artefactual mitochondrial signal in +lipid images & subtracting it from the raw +lipid image. Scale bars: 20 μm. **e**. Segmentation of marker channels to generate probability masks and representative result of lipid channel background removal. Scale bar: 20 μm. **f**. Lipid signal assignment to individual organelles shown in **e**. Scale bar: 20 μm. **g**. Representative image (PC(Y/18:1), 30 min timepoint) showing lipid signal and individual organelles markers (PM, endosomes, mitochondria) by four-colour fluorescence imaging (left panels). Scale bar: 10 μm. Right panels: Lipid signal assignment for cells shown based on automated image segmentation.

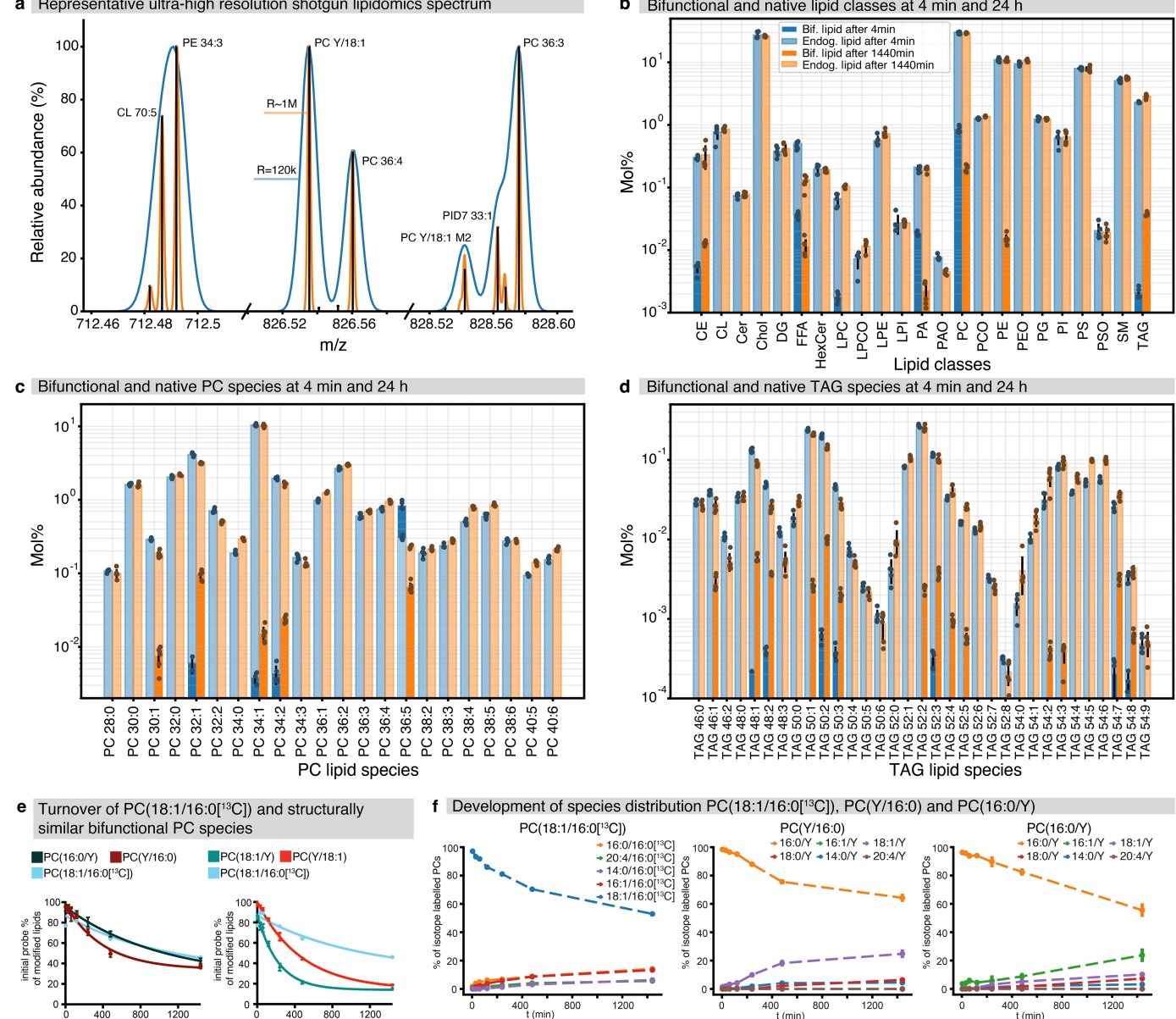

**Extended Data Fig. 5 | Ultra-high resolution (UHR) shotgun lipidomics of bifunctional lipid probes. a**. Shotgun UHR mass spectrometry resolves lipid peaks spaced by a few mDA and matches bifunctional precursors and their metabolites in multiple lipid classes. Blue line: Section of the spectrum acquired at the conventional ($R_s$ 120,000) resolution on Q Exactive mass spectrometer; orange line: Same spectrum section acquired at $R_s$ ~ 1 M resolution using optional Booster X2 data processing system and extended (2 s) transients. Vertical lines are peak centroids. **b**. Mol% profile acquired at two time points (see inset for colour coding) of 23 lipid classes (light bars), of which 7 classes comprise lipids with bifunctional lipid moieties (dark bars) produced from PC Y/20:4. **c**. PC profile covering 22 species with 5 species containing the bifunctional fatty acid. PCs bearing a bifunctional fatty acid (16:1) are annotated as endogenous lipids having the same number of carbons and double bonds in both FA moieties, albeit having different (+28.0061 Da) masses. **d**. Bifunctional fatty acids from the source PC(Y/20:4) are, incorporated into different lipid classes e.g., the cellular TAG pool consisting of 33 species with 14 species bearing the bifunctional fatty acid. The molar abundance of PC species containing the bifunctional fatty acid other than PC(Y/20:4) (**c**) and TAG (**d**) species increases with time, while the abundance of the starting PC(Y/20:4) decreases. Molar% profiles of native lipid classes (**b**), but also the species profile within PC and TAG classes (**c**, **d**) are not perturbed, indicating that the supplemented bifunctional lipids act as true tracer compounds and do not change the overall lipidome compositions. **e**. Comparison of bifunctional lipid metabolism with the native, isotope labelled, monounsaturated PC species PC(18:1/16:0[$^{13}$C]). **f**. Development of the PC species distribution of PC(18:1/16:0[$^{13}$C]) and palmitate-containing bifunctional PC species shows similar persistence of the original species in the labelled lipidome and similar product species forming. Mean and 95% CI of 3 biological repeats containing 2 technical replicates each are shown for **b**-**f**.

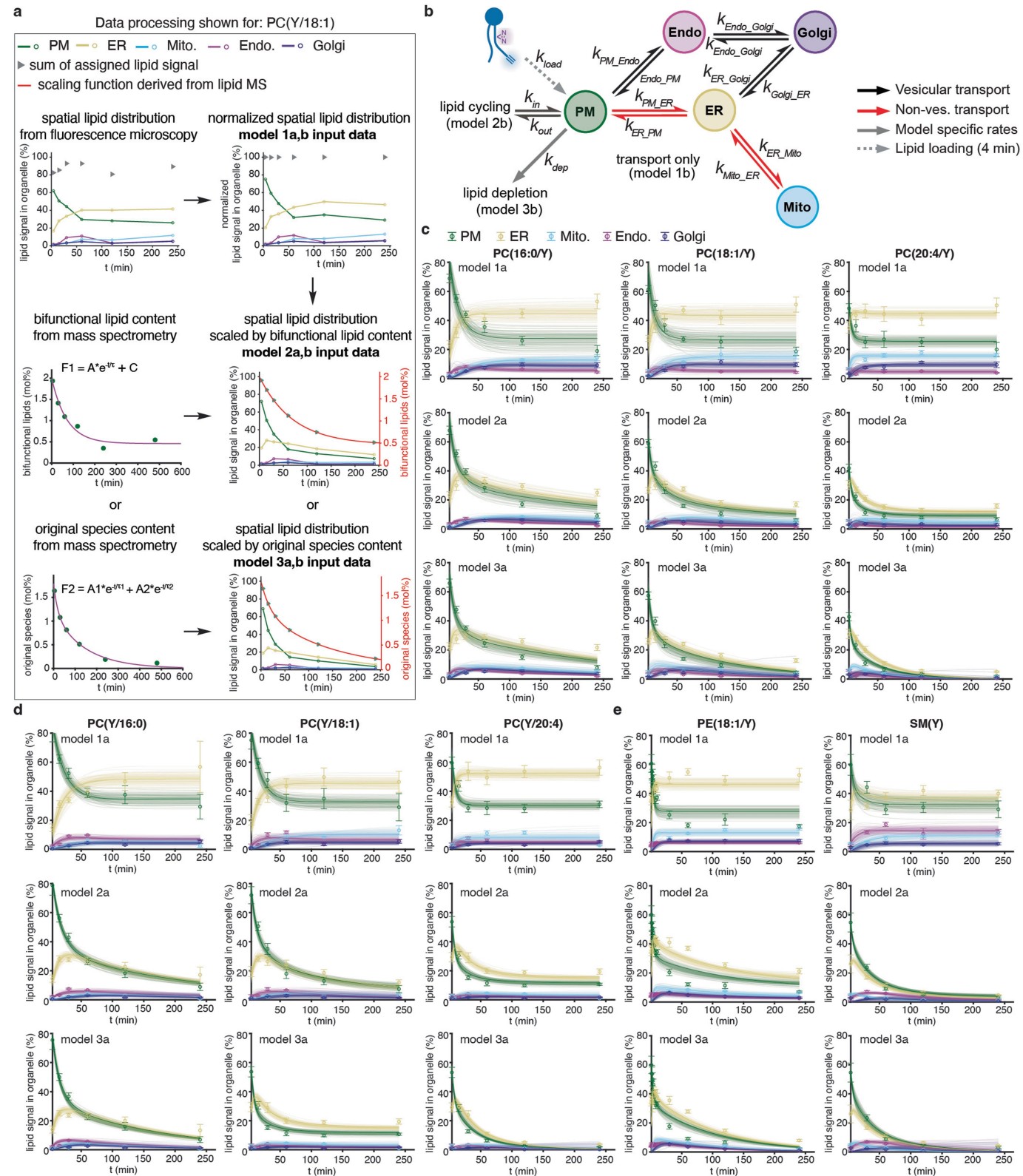

**Extended Data Fig. 6 | Kinetic analysis of lipid imaging and lipid MS data.**
**a.** Data processing steps for quantification results from lipid imaging time course experiments exemplary shown for PC(Y/18:1) (see Supplementary Information for details). **b.** Transport scheme detailing kinetic models 1b-3b. **c.** Model 1a-3a fits for PC(16:0/Y), PC(18:1/Y), PC(20:4/Y). **d.** Model 1a-3a fits for PC(Y/16:0), PC(Y/18:1), PC(Y/20:4). **e.** Model 1a-3a fits for PE(18:1/Y), SM(Y). Datapoints in were constructed from 5 independent time points (Mean and SD). Individual n-numbers can be found in the source data files.

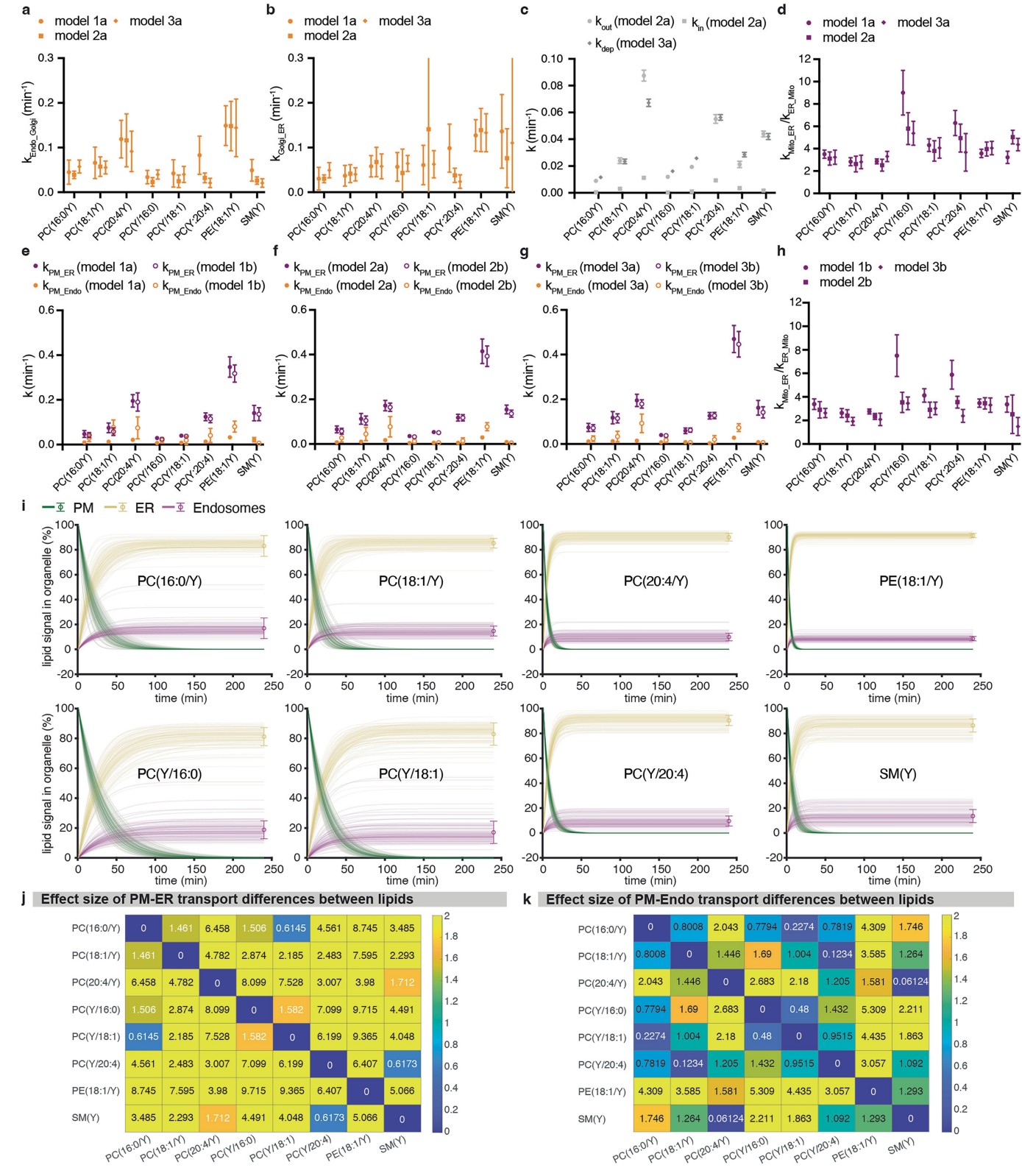

**Extended Data Fig. 7** | See next page for caption.

**Extended Data Fig. 7 | Results of kinetic analysis and estimation of lipid flow through vesicular and non-vesicular pathways. a, b**. Rate constants for vesicular transport from endosomes to the Golgi and from the Golgi to the ER derived from models 1a-3a. **c**. Rate constants describing lipid cycling (lipid exchange with the extracellular space, model 2a) and lipid depletion (model 3a). **d**. Ratio of rate constants describing lipid exchange between the ER and mitochondria (models 1a-3a). Note: Individual rate constants could not be identified from the data, presumably as preceding lipid transport steps were rate-limiting. **e-g**. Comparison of rate constants describing retrograde vesicular transport from the PM to endosomes and retrograde non-vesicular transport from the PM to the ER for all analysed lipid probes, corresponding models 1a and 1b, 2a and 2b, 3a and 3b shown together. **h**. Ratio of rate constants describing lipid exchange between the ER and mitochondria (models 1b, 2b, 3b). Note: Individual rate constants could not be identified from the data, presumably as preceding lipid transport steps were rate-limiting. **a-h** Error bars: mean and SD from 100 MC model runs. **i**. Fraction of bifunctional lipids transported via the non-vesicular route to the ER and the vesicular route to endosomes during retrograde transport, model 1a rate constants used for simulations. Plots show 100 individual model trajectories as well as the mean and SD of the endpoint distribution from 100 MC model runs. **j,k**. Effect size comparison for differences between lipid species in retrograde transport from the PM to the ER ($k_{PM\_ER}$) and endosomes ($k_{PM\_Endo}$), respectively, numbers in the heatmaps are Cohen's d. Cohen's d values: >0.01 very small (VS), >0.20 small (S), >0.50 medium (M), >0.80 large (L), >1.20 very large (VL), >2.00 Huge (H).

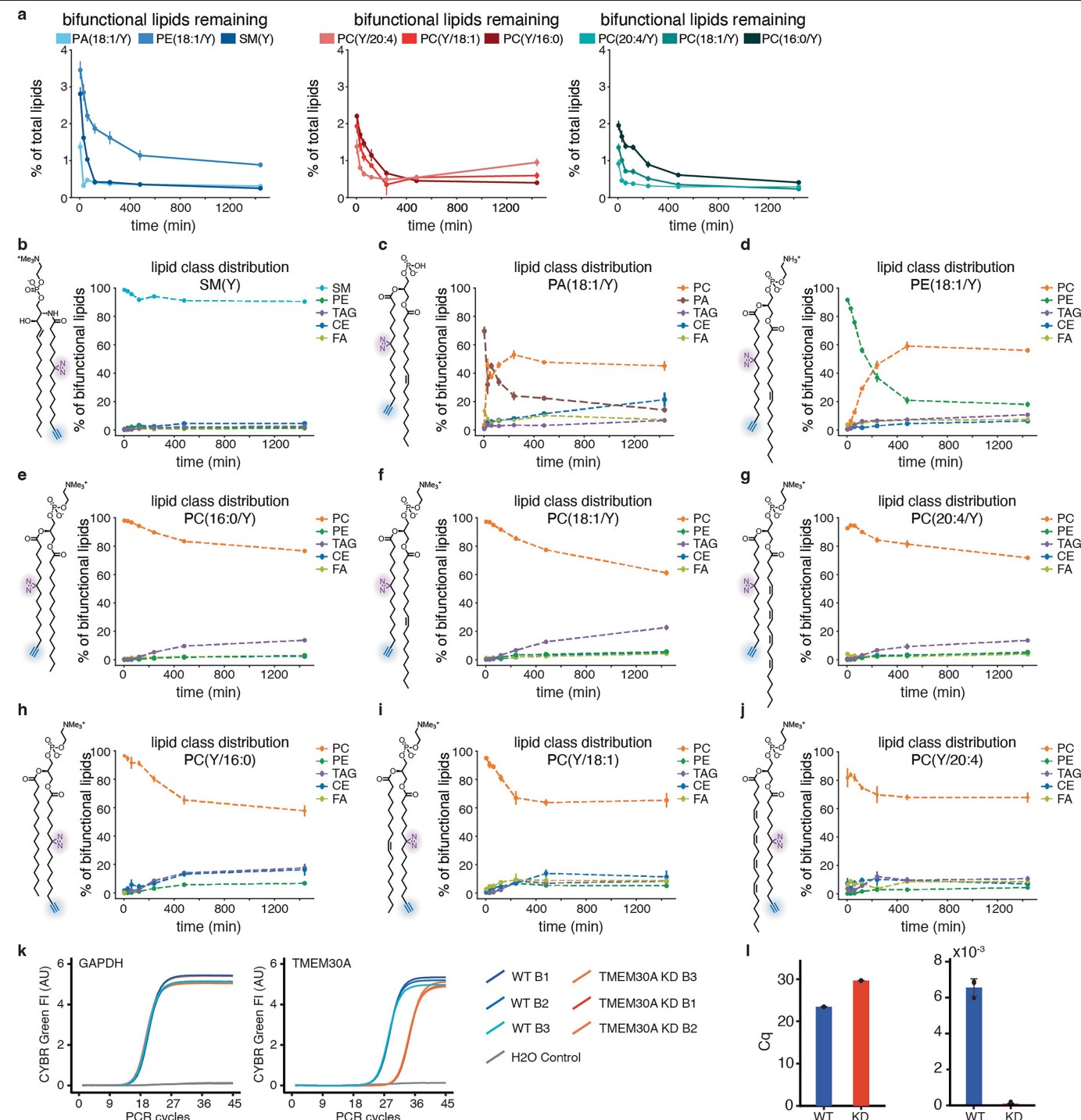

**Extended Data Fig. 8 | Determination of overall bifunctional lipid content and metabolism of the lipid class level and characterization of TMEM30A KD. a**. Bifunctional lipid incorporation and subsequent depletion over 24 h determined by shotgun lipidomics. Error bars: Mean and 95% CI of 3 biological repeats containing 2 technical replicates each. **b-j**. Development of bifunctional lipid class distribution over 24 h for all lipid probes. Note that final distributions are not identical, even for closely related species. **b-j** show the mean and 95% CI n = 3 biological repeats containing 2 technical replicates each. **k**. Confirmation of TMEM30A KD in HCT116 cells shown by qPCR of GAPDH and TMEM30A in WT and KD cells. **l**. Quantification cycle (Cq) and Cq normalized to GAPDH (DCq). Mean and SD of 3 biological replicates with 4 technical replicates each.

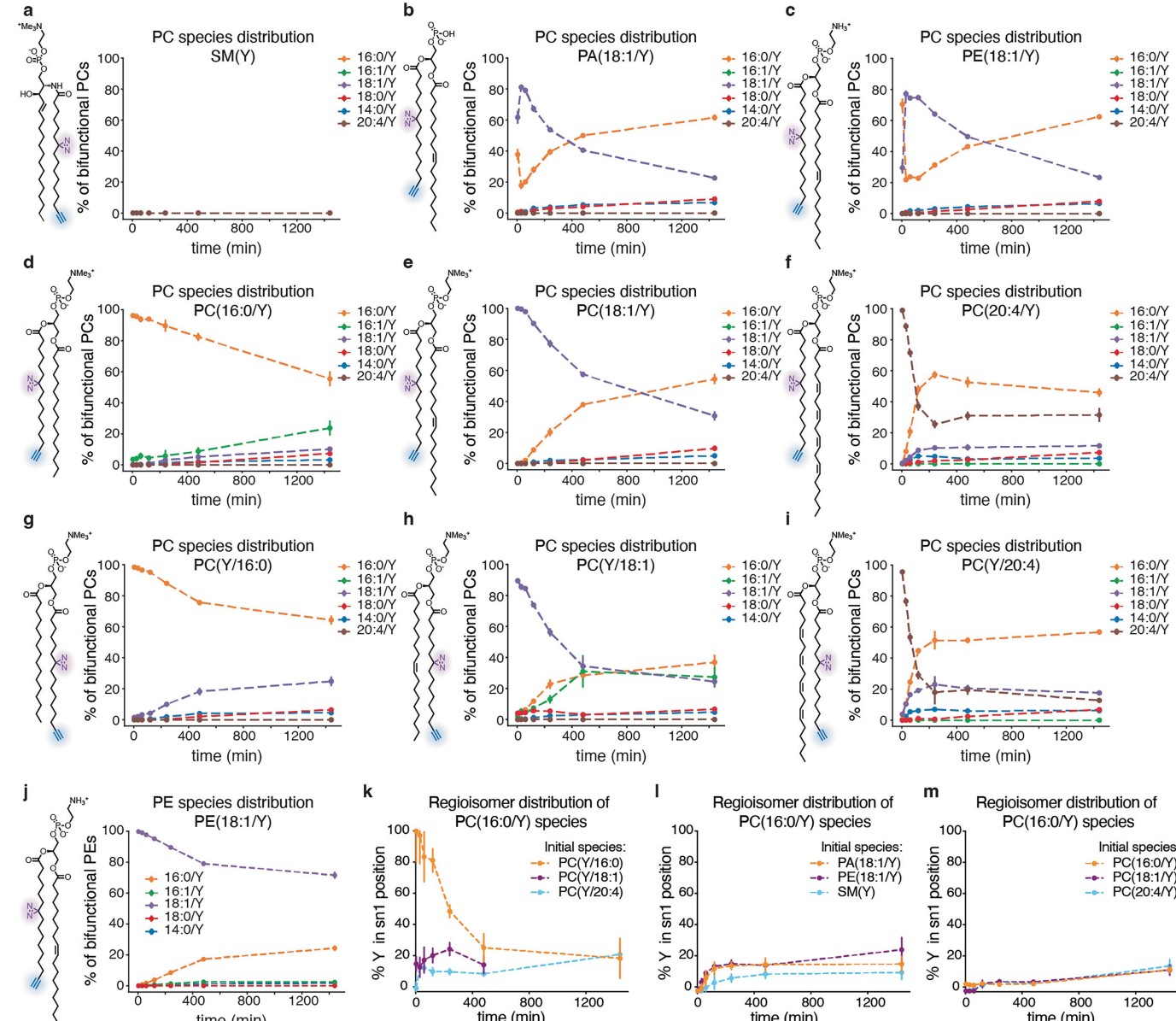

**Extended Data Fig. 9 | Analysis of PC species distribution. a-i.** Development of PC species distribution over 24 h for all lipid probes. Note that some species, notably PC(16:1/Y) are only produced from a subset of the initially supplied lipids. For SM(Y), no detectable amount of PC was observed. **j.** Development of PE species distribution over 24 h after loading PE(18:1/Y). **k-m.** Development of the regioisomer distribution of the most common PC species PC(16:0,Y) estimated via the MS/MS- fatty acid neutral loss fragments. The bifunctional fatty acid is primarily incorporated at the *sn2*-position. **a-m.** Mean and 95% CI, n = 3 biological repeats containing 2 technical replicates each.

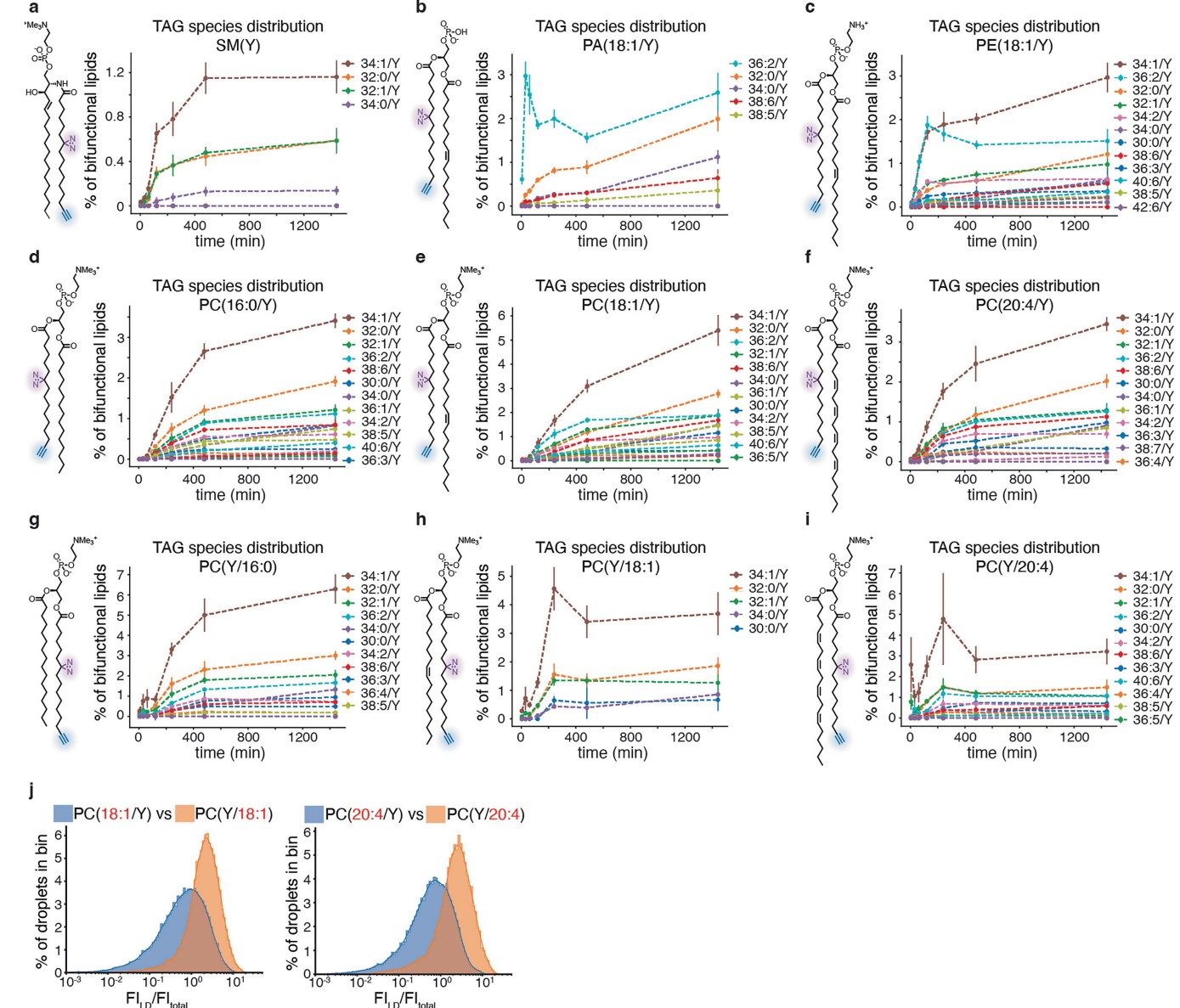

**Extended Data Fig. 10 | Analysis of TAG species distribution and lipid droplet populations. a-i.** Development of TAG species distribution over 24 h for all lipid probes. Note that PA (**c**) is the only lipid that initially gives rise to a single TAG species, whereas all other probes yield a spectrum of TAGs.

Mean and 95% of 3 biological repeats containing 2 technical replicates each. **j.** Comparison of intensity distribution of individual lipid droplets for PC(18:1/Y) and PC(Y/18:1) and PC(20:4/Y) and PC(Y/20:4). Distribution drawn from 5 fields of view for each lipid.

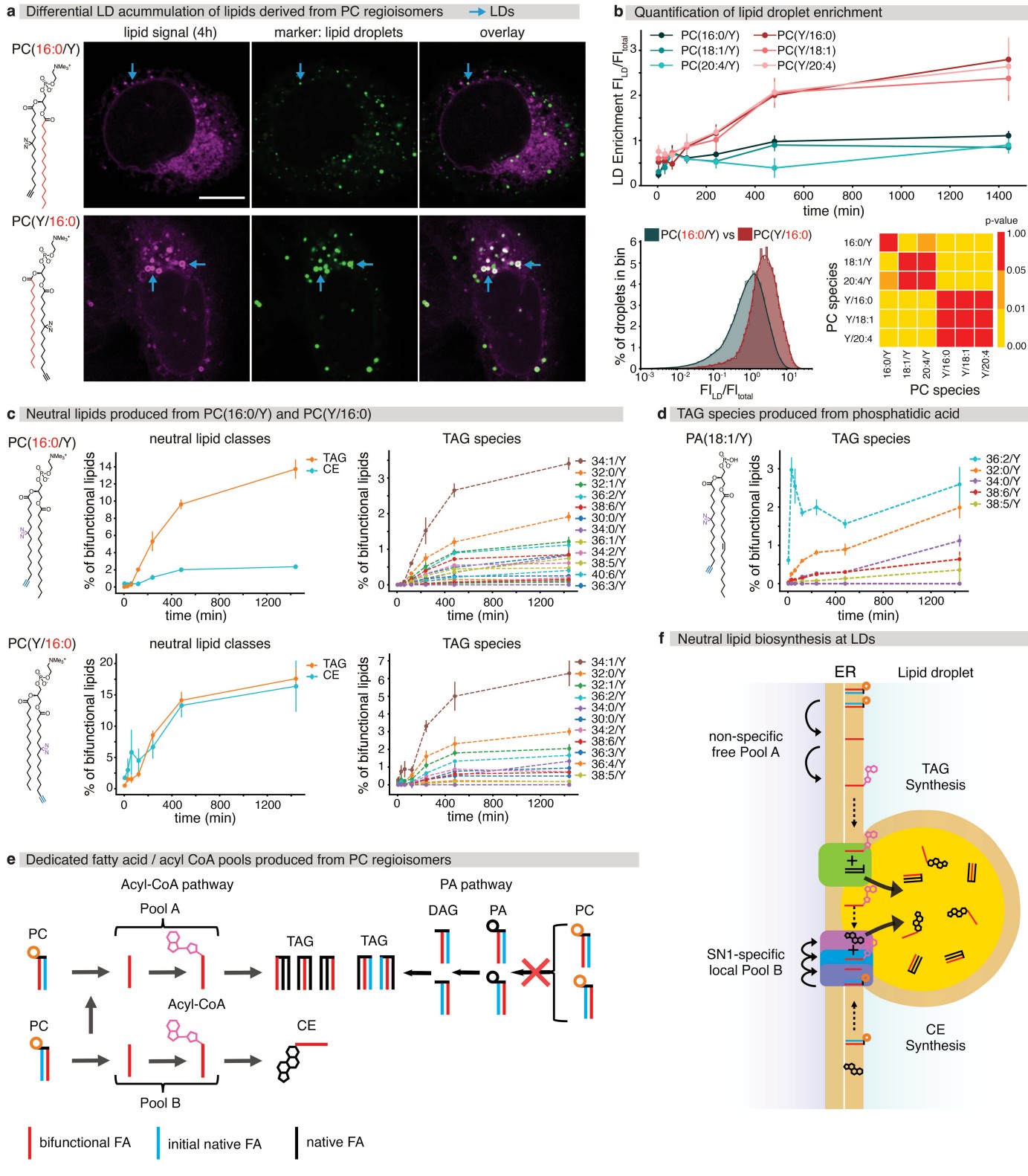

**a** Differential LD acummulation of lipids derived from PC regioisomers

**b** Quantification of lipid droplet enrichment

**c** Neutral lipids produced from PC(16:0/Y) and PC(Y/16:0)

**d** TAG species produced from phosphatidic acid

**e** Dedicated fatty acid / acyl CoA pools produced from PC regioisomers

**f** Neutral lipid biosynthesis at LDs

**Extended Data Fig. 11 | Dedicated pools of fatty acids are utilized during neutral lipid biogenesis. a**. Lipid droplets (stained with LipidSpot 610, green) exhibit a bright lipid signal (magenta) 4 h after loading PC(Y/16:0), bottom panels which is not observed after loading PC(16:0/Y), top panels. Scale bar: 10 µm. **b**. Upper panel: Quantification of fluorescence intensity of cellular lipid droplets for all PCs over time. Lower panels: Comparison of intensity distribution of individual lipid droplets for PC(16:0/Y) and PC(Y/16:0), 8 h after loading and statistical analysis of the similarity of the respective distributions for all PC species. Error bars: Mean ± SE of 5 fields of view. **c**. Mass spectrometric determination of bifunctional lipid content in neutral lipids demonstrates that significantly more TAG than CE is generated after loading PC(16:0/Y) whereas similar amounts of CE and TAG are generated after loading PC(Y/16:0) (left panels). Both species yield complex TAG patterns (right panels) and all TAG species are produced with similar kinetics. Mean and 95% confidence intervals of 3 biological repeats containing 2 technical replicates each. **d**. A single TAG species is initially produced after supplying PA (18:1/Y), whereas all other species are produced with slower kinetics. Mean and 95% confidence intervals of 3 biological repeats containing 2 technical replicates each. **e**. Schematic overview of neutral lipid biosynthesis at lipid droplets **f**. Proposed neutral lipid biosynthetic pathway model featuring dedicated free fatty acid / Acyl-CoA pools.

# Reporting Summary

## Statistics

For all statistical analyses, confirm that the following items are present in the figure legend, table legend, main text, or Methods section.

| n/a | Confirmed | |
|---|---|---|
| ☐ | ☒ | The exact sample size (*n*) for each experimental group/condition, given as a discrete number and unit of measurement |
| ☐ | ☒ | A statement on whether measurements were taken from distinct samples or whether the same sample was measured repeatedly |
| ☐ | ☒ | The statistical test(s) used AND whether they are one- or two-sided *Only common tests should be described solely by name; describe more complex techniques in the Methods section.* |
| ☒ | ☐ | A description of all covariates tested |
| ☒ | ☐ | A description of any assumptions or corrections, such as tests of normality and adjustment for multiple comparisons |
| ☐ | ☒ | A full description of the statistical parameters including central tendency (e.g. means) or other basic estimates (e.g. regression coefficient) AND variation (e.g. standard deviation) or associated estimates of uncertainty (e.g. confidence intervals) |
| ☐ | ☒ | For null hypothesis testing, the test statistic (e.g. *F*, *t*, *r*) with confidence intervals, effect sizes, degrees of freedom and *P* value noted *Give P values as exact values whenever suitable.* |
| ☒ | ☐ | For Bayesian analysis, information on the choice of priors and Markov chain Monte Carlo settings |
| ☒ | ☐ | For hierarchical and complex designs, identification of the appropriate level for tests and full reporting of outcomes |
| ☐ | ☒ | Estimates of effect sizes (e.g. Cohen's *d*, Pearson's *r*), indicating how they were calculated |

*Our web collection on statistics for biologists contains articles on many of the points above.*

## Software and code

Policy information about availability of computer code

| Data collection | Olympus cellSens Dimension (Version 4.1), ChipSoftManager 8.3.1.1018, Thermo Scientific Q Exactive 2.9, Bruker TopSpin 3.6.2, Thermo Scientific Xcalibur 4.4.16.14, FTMS Booster Control Software 2018.6.0, Olympus FV10-ASW 4.2, SymPho Time 64 2.6, Sepia II |
|---|---|
| Data analysis | Python 3.9.7 with the following packages: tifffile 2020.9.3, python-javabridge 4.0.2, python-bioformats 4.0.3, numpy 1.19.3, apeer-ometiff-library 1.8.2, pandas 1.1.5, scipy 1.5.4, scikit-image 0.17.2, scikit-learn 0.24.2, matplotlib 3.3.4, Pillow 8.4.0, opencv-python 4.7.0.72. Ilastik 1.4.Thermo Scientific Xcalibur 4.4.16.14, MestReNova 14.3. PeakbyPeak (SpectroSwiss) 2020.10.0.b1, LipidXplorer 1.2.4, Simtrim, Jupyter Notebook. ImageJ 1.54h, SymPhoTime 5.3.2.2, OriginPro 8.5, Matlab R2020b, Matlab Optimization toolbox<br><br>The code can be found at http://doi.org/21.11101/0000-0007-FCE3-D |

For manuscripts utilizing custom algorithms or software that are central to the research but not yet described in published literature, software must be made available to editors and reviewers. We strongly encourage code deposition in a community repository (e.g. GitHub). See the Nature Portfolio guidelines for submitting code & software for further information.

## Data

Policy information about availability of data

All manuscripts must include a data availability statement. This statement should provide the following information, where applicable:

- Accession codes, unique identifiers, or web links for publicly available datasets
- A description of any restrictions on data availability
- For clinical datasets or third party data, please ensure that the statement adheres to our policy

> The complete lipid flux dataset can be interactively accessed under http://doi.org/21.11101/0000-0007-FCE5-B and the original data can be downloaded from http://doi.org/21.11101/0000-0007-FCE4-C and https://doi.org/10.6019/S-BIAD1695. A demo-dataset is available at https://doi.org/10.17617/3.BRSGLA. The source data for all the plots presented in the main and extended data figures is provided as separate Excel files.

## Research involving human participants, their data, or biological material

Policy information about studies with human participants or human data. See also policy information about sex, gender (identity/presentation), and sexual orientation and race, ethnicity and racism.

| | |
|---|---|
| Reporting on sex and gender | n/a |
| Reporting on race, ethnicity, or other socially relevant groupings | n/a |
| Population characteristics | n/a |
| Recruitment | n/a |
| Ethics oversight | n/a |

Note that full information on the approval of the study protocol must also be provided in the manuscript.

# Field-specific reporting

Please select the one below that is the best fit for your research. If you are not sure, read the appropriate sections before making your selection.

☒ Life sciences  ☐ Behavioural & social sciences  ☐ Ecological, evolutionary & environmental sciences

For a reference copy of the document with all sections, see nature.com/documents/nr-reporting-summary-flat.pdf

# Life sciences study design

All studies must disclose on these points even when the disclosure is negative.

| | |
|---|---|
| Sample size | No sample size calculation was performed. The low variability between the different labeling conditions shows that a small number of independent repeats is sufficient. Mass spectrometry measurements contained 3 biological replicates and 3 technical replicates. For microscopy experiments, at least 5 fields of view were acquired per condition to ensure statistical power for analysis. |
| Data exclusions | Images were excluded if the acquisition process failed by either being out of focus or if unwanted cellular debris or clamps precluded image analysis. No data was excluded from mass-spectrometry. |
| Replication | Mass spectrometry experiments were carried in 3 biological replicates with 2 technical replicates. Microscopy experiments for U2OS cells that were used to calculate the amount of lipid in the plasma membrane were replicated 3 times with at least 10 fields of view; ER experiments were replicated 2 times with at least 10 fields of view; Endosomes, Mitochondia and Golgi were measured once with at least 5 fields of view. Microscocopy experiments using HCT116 involving WT and TMEM30A mutants were replicated at least 2 times with at least 10 fields of view. Drug experiments to block endocytosis were performed once and at least 10 fields of view were captured. |
| Randomization | Images were taken at random positions in the wells of a 96 well-plate. For this we made use of the CellSens software. Mass spectrometry experiments cannot be randomized, the lipidome of the whole sample was quantified. |
| Blinding | Not relevant for this study. The microscopy data was acquired in an unbiased way. For mass spectrometry the whole lipidome of the sample was studied |

# Reporting for specific materials, systems and methods

We require information from authors about some types of materials, experimental systems and methods used in many studies. Here, indicate whether each material, system or method listed is relevant to your study. If you are not sure if a list item applies to your research, read the appropriate section before selecting a response.

## Materials & experimental systems

| n/a | Involved in the study |
|-----|----------------------|
| ☐ | ☒ Antibodies |
| ☐ | ☒ Eukaryotic cell lines |
| ☒ | ☐ Palaeontology and archaeology |
| ☒ | ☐ Animals and other organisms |
| ☒ | ☐ Clinical data |
| ☒ | ☐ Dual use research of concern |
| ☒ | ☐ Plants |

## Methods

| n/a | Involved in the study |
|-----|----------------------|
| ☒ | ☐ ChIP-seq |
| ☒ | ☐ Flow cytometry |
| ☒ | ☐ MRI-based neuroimaging |

# Antibodies

| | |
|---|---|
| Antibodies used | Rab5 (rabbit), Cell Signaling Technology CST-3547S, Clone: C8B1, Lot: 7<br>Rab7 (rabbit), Cell Signaling Technology CST-9367S, Clone: D95F2, Lot: 3<br>Tom20 (mouse), Santa Cruz sc-17764, Clone: F-10, Lot: H0320<br>ATP5A (mouse), Abcam ab14748, Clone: 15H4C4, Lot: 2101038509<br>Golgin97 (rabbit), Abcam ab84340, Clone:Polyclonal, Lot:1<br>Giantin (rabbit), Abcam ab80864, Clone: Polyclonal, Lot:GR3209923-1<br>GM130 (rabbit), Cell Signaling Technology CST-12480S, Clone:D6B1 Lot: 3<br>Lamp1 (rabbit), Cell Signaling Technology CST-9091S, Clone: D2D11 Lot: 6<br>Tom20 (rabbit), Cell Signaling Technology CST-42406S,  Clone:D8T4N Lot: 4<br>ATP5A (rabbit), Abcam ab176569, Clone: EPR13030(B), Lot:GR3291066-12<br>Calnexin 1 (mouse) Abcam ab112995, Clone: 6F12BE10, Lot: GR3246794-6<br>Lamin A/C (mouse), Cell Signaling Technology CST-4777S, Clone:4C11 Lot:5<br>BAP31 (mouse), Enzo Life Sciences ALX-804-601-C100,  Clone: A1/182 Lot: L15093<br>Calreticulin (mouse), Abcam ab22683, Clone: FMC 75, Clone: Lot:GR3361946-5 |
| Validation | All primary antibodies were used directly from the manufacturer. All antibodies were validated by the manufacturers to be suitable for immunofluorescence (IF) and reactive with Human samples.<br><br>Rab5, CST-3547S -> Applications: WB, IF Species Reactivity: Human, Mouse, Rat, Monkey<br>Rab7, CST-9367S -> Applications: WB, IP, IF. Species Reactivity: Human, Mouse, Rat, Monkey<br>Tom20, sc-17764 -> Applications: WB, IP, IF, IHC(P), ELISA. Species Reactivity: Human, Mouse, Rat<br>ATP5A, ab14748 -> Applications: WB, IF, IHC(P), FC. Species Reactivity: Human, Rat, Cow, Mouse, Drosophila melanogaster.<br>Golgin97, ab84340 -> Applications: IF, HC(P), ICC. Species Reactivity: Human<br>Giantin, ab80864 -> Applications: WB, IF, ICC, IHC(P). Species Reactivity: Human<br>GM130, CST-12480S -> Applications: WB, IF, IP. Species Reactivity: Human, Monkey<br>Lamp1, CST-9091S -> Applications: WB, IF, IP, IHC, FC. Species Reactivity: Human, Monkey<br>Tom20, CST-42406S -> Applications: WB, IF, IHC, IP. Species Reactivity: Human, Mouse, Rat, Monkey<br>ATP5A, ab176569 -> Applications: WB, IF, ICC, IHC(P), FC. Species Reactivity: Human, Mouse, Rat<br>Calnexin, ab112995 -> Applications: WB, IF, ICC, IHC(P), FC. Speacies Reactivity: Human<br>Lamin A/C, CST-4777S -> Applications: WB, IF, IP, IHC, IP. Species Reactivity: Human, Mouse, Rat, Monkey<br>BAP31, ALX-804-601-C100 ->Applications: WB, IF, IP, ICC, FC, ELISA, IHC. Species Reactivity: Human, Monkey<br>Calreticulin, ab22683 -> Applications: IF, ICC, FC, IHC(P). Species Reactivity: Human |

# Eukaryotic cell lines

Policy information about cell lines and Sex and Gender in Research

| | |
|---|---|
| Cell line source(s) | U-2 OS cells were purchased from ATCC and were originally obtained from a female.  HCT-116 cells were purchased from ECCAC and have a male origin. |
| Authentication | U-2 OS cells were authenticated by eurofins genomics using 16 independent PCR-systems. HCT-116 cells were not authenticated after purchase. |
| Mycoplasma contamination | All cell lines are routinely tested for Mycoplasma contamination. Data presented in this study was collected only from negative samples. |
| Commonly misidentified lines<br>(See ICLAC register) | No commonly misidentified cell lines were used in this study. |

## Plants

Seed stocks

*Report on the source of all seed stocks or other plant material used. If applicable, state the seed stock centre and catalogue number. If plant specimens were collected from the field, describe the collection location, date and sampling procedures.*

Novel plant genotypes

*Describe the methods by which all novel plant genotypes were produced. This includes those generated by transgenic approaches, gene editing, chemical/radiation-based mutagenesis and hybridization. For transgenic lines, describe the transformation method, the number of independent lines analyzed and the generation upon which experiments were performed. For gene-edited lines, describe the editor used, the endogenous sequence targeted for editing, the targeting guide RNA sequence (if applicable) and how the editor was applied.*

Authentication

*Describe any authentication procedures for each seed stock used or novel genotype generated. Describe any experiments used to assess the effect of a mutation and, where applicable, how potential secondary effects (e.g. second site T-DNA insertions, mosiacism, off-target gene editing) were examined.*

