## [Peer Review File · Nature]

Quantitative imaging of lipid transport in mammalian cells

Corresponding Author: Dr André Nadler

Version 0:

Reviewer comments:

Referee #1

(Remarks to the Author)

In this work, Iglesias-Artola and coworkers develop a novel, coherent and robust pipeline to investigate retrograde lipid flux in cells. Their approach, based on the use of bifunctional lipids, allows to couple organellar fluorescence microscopy with quantitative lipidomics, and it is achieved thanks to the careful optimization of the experimental parameters. Mathematical modeling of the observed fluxes using straightforward kinetic models allows the authors to interpret their data and to extract meaningful information on the fate of different lipid species upon their cellular internalization.

From a technical and methodological point of view, this work is a remarkable tour-de-force that significantly advances our current technology to investigate retrograde lipid fluxes, and I expect it to become a landmark methodology in the field.

Specifically, I have no criticisms on this aspect and I find it an amazing technical work.

From a biology point of view, on the one hand the authors nicely confirm previous results/expectations, such as the prevalence of non-vesicular trafficking or the key role of PM flippases, on the other hand they provide novel insights, for example into the spatial and temporal separation between lipid transport and metabolism or into the head and chain specificity of lipid internalization. Notably, because of the nature of their approach, they provide a quantitative estimation of these processes, which was mostly not available with previous methods.

At the same time, I find that the coupling of the newly developed technology with genetic perturbations is severely underdeveloped, causing most of the novel observations to remain somewhat superficial. Specifically, all the constants extracted from the kinetic models are "effective" ones, likely resulting from multiple pathways. In order to identify the origin of the observed differences (e.g. between lipid species), and hence to investigate causative relationship in lipid biology, the authors should extensively and systematically couple their approach with genetic perturbations, and identify the origin of the observed differences. Some examples/suggestions:

1. Does the observed difference between lipid species originate from differences in lipid flipping at the PM or from specificity in lipid transport?
2. What is the role of the the different lipid transport proteins proposed to work at the ER-PM contact sites?
3. What is the contribution of shuttle (ORP...) versus bridge (VPS13A-D) proteins?
4. For lipid metabolism: what is the relative contribution of ACAT1 and ACAT2? And of DGAT1 and DGAT2? What about the enzymes upstream of the last step of sterol-ester and triacylglycerol synthesis?

While I understand that most of these investigations could be explored in future studies, I believe that the article would strongly benefit from a deeper investigation (achieved with genetic perturbations) in at least one of these areas. This will highlight the potential of this beautiful new approach (which in its current form is more suited to more method development journals, such as Nature Method, for example) to gain novel important biological insights.

(Remarks on code availability)

Most areas of the code are clear and function appropriately. However, small demo datasets to test that the python and matlab scripts used for the mass spectrometry data analysis and for the kinetic model building, respectively, are currently missing. Without these datasets, it is not possible to check whether the results presented in the paper are reproducible.

Referee #2

(Remarks to the Author)

This study investigates retrograde lipid transport and lipid metabolism using exogenous clickable phospholipid. The lipids are solubilized with alpha-methyl-cyclodextrin and added to cells, initially incorporating into the plasma membrane (PM). The transport of the lipids (or their derivatives) is followed over time by crosslinking, fixation, and detection using click chemistry. State of art ultra-high resolution mass spec is used to determine how the lipids are metabolized over time. This is a novel strategy. Previous studies have used labeled lipids, typically fatty acids or cholesterol, and fractionation or enzymatic conversion to estimate rate of lipid movement from the PM to various compartments. There have been few studies in which diacyl phospholipids are added to cells and using cyclodextrin deliver the lipids is a good idea. There are four main conclusions: (1) retrograde transport is primarily non-vesicular, (2) flippases play a role in lipid transport, (3) lipid metabolism is about 10x slower than transport, and (4) the rate of incorporation of exogenous lipids into neutral lipids is affected by the saturation of the acyl chains. While the work is well done and highly technically innovative, there are concerns about the significance of the conclusions and the interpretation of the findings. Here are the major concerns.

1. It is already well established that exogenous cholesterol, fatty acids, and lysolipids are rapidly moved by non-vesicular mechanisms from the PM to the ER. The exogenous lipids are initially incorporated into the PM and converted to other lipids in the ER, an indirect indication of transport; for example, exogenous cholesterol is rapidly converted to cholesteryl ester, which happens in the ER. The rate of conversion is too fast to be entirely explained by vesicular transport and is not blocked by genetic or chemical treatments that block vesicular trafficking, indicating that transport is largely non-vesicular. This study uses phospholipids with 2 acyl chains, which has only been done in a few previous studies and not nearly as carefully as here. There have also been several papers on the transport of newly synthesized lipids from the to the PM and other organelles that suggest lipid can be moved by non-vesicular pathways that have a high capacity to move lipids. Therefore, the finding that exogenous phospholipids with two acyl chains also move by non-vesicular pathways to internal membranes is not terribly surprising. However, previous studies have not estimated what fraction of exogenous lipid (or endogenous lipid) moves by vesicular and non-vesicular mechanisms, which is a notable advance provided here. There are some concerns about the transport assays that should be addressed. One is the rate of conversion of the exogenously added lipids to other species and how this affect the calculated transport rate. For each time point in the transport assays (for example in Fig 2c), the percent of the added lipid that has been converted to other species should be shown. At the later time points, is a significant fraction of the signal in an organelle from lipid species other than the one added to the cells? This data and the transport data should be given for all the lipids shown in Fig 1b.

Another concern about the transport assay are the results with sphingomyelin (SM). Since SM (and other complex sphingolipids) are synthesized on the extracytoplasmic leaflet of the Golgi and do not flip, it is widely thought they are not available to lipid transport proteins operating in the cytoplasm. There are also no lipid transport proteins known to bind SM. Therefore, the transport of intact SM should be almost entirely vesicular, which does not seem to be the case here. The ability of SM to be moved by non-vesicular pathways should be assessed by blocking vesicular transport, for example by depleting cells of NSF. If it turns out that SM transport is not blocked by inhibiting vesicular transport, the authors should explain why previous studies are incorrect.

2. The finding that knock down of TMEM30A slows transport is interesting, but there are no controls to show that the effect is direct. It remains possible that TMEM30A affects membrane properties or the function of proteins other than lipid transport proteins. TMEM30A depletion could also affect vesicular transport. Other flippases should also be investigated.

3. It is already well known that the saturation level of fatty acids affects their rates of incorporation of into neutral lipid. It is not clear what the findings here add to what is already known or how this is a significant advance.

4. More needs to be done to verify that biophysical properties of the lipids shown in Fig. 1b. The biophysical properties of membrane composed entirely of the PC or PA species shown in Fig 1b should be compared to membranes composed of the identical lipids lacking the diazirine and alkyne groups. For the PC species, their ability to be transported in vitro by a PC lipid transport protein like Sec14 should be assessed and compared PC species with the diazirine and alkyne groups.

Referee #3

(Remarks to the Author)

In this manuscript, André Nadler's group has adapted innovative methods to study lipid transport between membranes in prototypical human cells. The method is based on the chemical synthesis of bifunctional lipids that contain both a photoactivatable and a clickable chemical label. The method is not new, but the authors have adapted it to systematic approaches and integrated pulsed-phase fluorescence imaging, ultra-high resolution mass spectrometry and mathematical modeling. The approach is well thought-out and elegant. The authors have created a library of 9 bifunctional lipid probes covering four different lipid classes: phosphatidylcholine (PC), phosphatidic acid (PA), phosphatidylethanolamine (PE) and sphingomyelin (SM). For the PC class, they also included 6 species with different fatty acids. The strength of this approach lies in the fact that it is based on pulse-chase experiments, enabling the authors to measure (and compare) the transport kinetics of different lipid classes and species. Such a quantitative analysis of lipid transport in a cell had, to my knowledge, never been carried out on this scale. This led to several interesting findings: FA composition is important and affects lipid transport kinetics, and retrograde lipid transport from the PM to the ER is faster and more selective than vesicular transport. The authors then address mechanistic questions concerning the directionality of non-vesicular transport (often against lipid gradients) and follow the hypothesis that active reversal of the trans layer between membrane sheets catalyzed by flippases and floppases may contribute. Overall, this work represents a tour de force and the first attempt to understand the general principles of non-vesicular lipid transport in a quantitative way. The study of lipids and their transport remains poorly understood due to the difficulty of analyzing and visualizing them in the cell. This work is a first attempt to answer this question and may lead/inspire many follow-up studies. Below are a number of points that I feel need to be addressed.

General points:

1) The reasons for selecting lipids for chemical engineering (and the scope of this work) should be explained. Are certain classes of lipids (i.e. those that are lacking: PS, PI, PG, cardiolipins) ceramides) not suitable for chemical synthesis, or have they been omitted because they were less relevant in the context of plasma membrane lipid supply, or because they are toxic (in high abundance and/or when found in the plasma membrane)? Why have the various FAs only been studied in the context of PC (and not other GPL/sphingolipids)? This information that could be useful for follow-up studies.

2) As a control, the authors measured the possible effects of bifunctional lipid probes on general membrane properties in an in vitro biochemical assay, which is an important control - they claim to find no effect. This requires some explanation. Why is it that membranes containing PC(20:4) have properties similar to those of PC(16:0) or PC(18:1)?

They use the same (liposome-based) delivery system for all probes, but can they really rule out that liposomes containing different bifunctional lipids lead to similar difunctional lipid uptake mechanisms? As bifunctional lipids can be cross-linked to proteins, pull-down analyses followed by proteomics (after cross-linking) (at early time points) would help answer this question, at least for a few important cases (i.e. identified proteins/machines involved in bifunctional lipid uptake/manipulation). This could also provide evidence and further validation of their transport by non-vesicular transport pathways (i.e. LTP).

4) The work actually focuses on a specific pool of bifunctional lipids, i.e. those that are cross-linked to interacting proteins, i.e. those embedded in biomolecular interactions. I understand that the crosslinking step is unavoidable, as the permeabilization of the cell (with detergent if I understand correctly) required for click chemistry will result in the loss of all non-crosslinked lipids, including bifunctional probes that do not interact with proteins. I think this should be explained and described, could it be that it includes a bias? When they normalize for total bifunctional lipids, it's actually for total bifunctional lipids associated with proteins (isn't it?) - could this include artifacts? Could it be that some of the observations described later stem indirectly from this ability (or inability) to interact with proteins (and not so much about transport)? More generally, what about "free bifunctional lipids" (i.e. not associated with proteins)? Do they exist? Are we learning anything from them? Is it acceptable to ignore them? This notion could be addressed as it could represent an advantage of their approach.

For a few important cases, they could perform cell fractionation and biochemically measure bifunctional lipids in the different organellar fractions (without cross-linking). This would also represent (for a few selected cases) an orthogonal verification/validation of their cross-linking/click chemistry approaches measuring the accumulation of bifunctional lipids in the different organelles.

Points on bifunctional lipids metabolism:

The authors have carried out an important analysis of the metabolic fate of bifunctional lipids supplied to the cell at the plasma membrane. This section is currently mainly to be found in the supplementary information. In my opinion, this section is very important and interesting for several reasons. It describes the limitations of the approach (possible effect of chemical markers on lipid metabolism and transport). Such limitations are inevitable and not necessarily problematic as long as we understand them precisely. From a more biological point of view, this may also highlight metabolic biases, i.e. lipids from PM do not have the same metabolic fate - which is interesting and also alludes to transport mechanisms. However, this section remains limited to additional information, and I think these aspects would merit presentation in the main text.

5) They mention that the SM probe was metabolically stable. Does this mean that it is not a good substrate for sphingomyelinase, or could another possibility be that its transport is somehow affected? More generally, a description of the metabolic fate of the different probes including a comparison to what is known/expected and a discussion concerning possible short comings would greatly help. More generally what results in the Supplementary Material 1 is important and a few Figure panels describing the most salient properties/behaviors of the different probes would be extremely useful for the community and other groups who may want to apply the methods in the future.

Maybe interesting to also analyze the secretome for SM-derived bifunctional probes – authors may still have the conditioned culture media.

6) The authors also allude to the fact that bifunctional lipids remain surprisingly well represented in the total lipidome (even for short-lived lipids such as PA). As the authors point out, this is surprising. They propose that this is due to the formation of lipid pools less likely to be metabolized. The authors suggest many hypotheses, but seem to rule out the possibility that this is due to chemical engineering of lipids (i.e. bifunctional lipids have a different metabolic fate/rate) and/or to less susceptibility to transport (e.g. linked to the flippase mechanism they describe later- bifunctional lipids are supplied on liposomes and on both sides of the leaflet). This point should be discussed. Here again, a description of the metabolic fate of the different probes in the context of a comparison with what we expect would greatly help to understand the possible limitations of the analysis. For example, do we expect bifunctional PC not to be metabolized into other glycerophospholipids? If not, can we learn from these discrepancies (origin of PC (PM instead of ER), effect of chemical labels, etc.). This part is important because it delimits the scope and limits.

Part on lipid transport:

7) One interesting finding is that specific retrograde transport takes place via non-vesicular pathways. However, this part is currently based mainly on kinetic models; a validation step would make it more convincing. What if they inhibit vesicular transport, etc.?

Later, they claim that polyunsaturated PC species were transported up to 10 times faster by the non-vesicular route than saturated PC species. Can they be sure that saturated species were not transported by the vesicular route? It seems that the model in figure 2e does not consider non-vesicular transport from endosomes to the ER. In general, the author should better define the criteria used by their model to predict vesicular versus non-vesicular transport, and validate their model experimentally.

8) The authors then explore an interesting hypothesis that trans-layer transport may play a role in non-vesicular lipid transport. However, many other mechanisms could explain the directionality of lipid transport (often against concentration gradients), such as metabolism (i.e. once transported, the lipid is rapidly metabolized into products that are not transported by the same LTP), which should probably also be mentioned. Their metabolic data (i.e. the metabolic fate of bifunctional

lipids) could be re-examined in the light of this (alternative) hypothesis.

9) Validation should be provided to show that knockout of the common PM flippase subunit does indeed lead to a flipping defect (and that this function is not redundant, etc.) - i.e. show that PE/PS is exposed at the cell surface (in non-permeabilized cells using specific probes). Furthermore, as these experiments are carried out in the presence of excess extracellular PE present on liposomes, i.e. presented on both leaflets of artificial membranes, what's the idea/model? PE is loaded into the cell on both sheets anyway. So, could it be that what is observed here is a consequence of the way lipids are delivered to the cell, but is not at play under more physiological conditions when endogenous PE comes from the inner leaflet of the PM or the ER?

Minor points:

- In paragraph "" lines 70-78, the authors briefly describe alternative approaches to studying lipids. This should also include tri-functional lipids, i.e. combining photocaging with cross-linking/click chemistry labels, as they also represent interesting alternatives for such analyses allowing targeting the release of the lipid from different organelle (not only the PM) in the future. They should also be mentioned.

- Figure 1 Panel e. The color code for the different locations is difficult to "decode" as the ER and lipids have very similar ranges of colors (at least on my screen). We can not see cases of assigned to ER.

- Line 171-173; the authors state that "...transport rate constants for vesicular transport via endosomes with non-vesicular transport to the ER revealed that non-vesicular trafficking was up to an order of magnitude faster for all lipids compared to vesicular transport." This seems to me to be a circular argument, as both vesicular and non-vesicular transport were initially modeled from transport kinetics. This needs clarification and/or rewording. Perhaps with a graph describing how the model has been constructed and/or what are the key parameters that define vesicular and non-vesicular transport?

- Line 223-224 the authors claim "While some lipid transfer proteins can move cholesterol, PA and PS against concentration gradients by PI(4)P2 co-transport^{8,31}, it is unclear which process provides the energy for directional transport of other lipids, in particular via bridge-like lipid transfer proteins" is not completely exact. In those cases, we know that the directionality of PS or cholesterol transport is driven to the counter transport of PI(4)P back to the donor membrane and its dephosphorylation into PI in this donor membrane. Energy/ATP hydrolysis is provided by rounds of PI phosphorylation dephosphorylation and PI(4)P. The authors should clearly mention this alternative hypothesis. i.e. that directionality is driven by lipid metabolism.

- Line 225: BTW it is not PI(4)P2 but PI(4)P

Version 1:

Reviewer comments:

Referee #1

(Remarks to the Author)

The authors have successfully addressed my previous concerns, and the new experiments contribute sufficient novel insights into the biology of lipid transport and metabolism for me to recommend publication of this work, which is now significantly improved. This is an amazing tour-de-force in term of methodological developments that I expect will contribute to important further future discoveries in the field of lipid biology.

I only have a few minor points to improve the clarity of the manuscript:

Abstract

- "Here we measured transport and metabolism"  I think should be replaced by "Here we measured retrograde transport and metabolism" as forward transport is not directly estimated in this work

- "active lipid flipping and passive non-vesicular transport" should be replaced by the more neutral and precise terms "energy-dependent" and "energy independent"

Figure 3c: the color code of the 3 treatments (grey, pink, cyan) could be also explained directly in the figure, rather than only in the legend.

Stefano Vanni

Referee #2

(Remarks to the Author)

My concerns have been carefully and thoughtfully addressed and I now favor publication of this study.

Referee #3

(Remarks to the Author)

The authors have taken into account all my points and comments. The revised manuscript has been considerably improved. This is an important work, the approach is powerful and elegant. The results show that it allows us to tackle aspects hitherto unexplored in lipid biology.

General remarks:

We thank all reviewers for their comments, which provided helpful and constructive feedback. We believe that the revisions made the manuscript substantially stronger.

We have addressed all points experimentally that were feasible in the time frame of the revision. We have acquired extensive new data, which are shown in a new main text figure, two significantly revised and expanded main text figures, an expanded ED figure and 7 new supplementary figures. Our main new findings are:

- Pharmacological interference with vesicular trafficking has no significant effect on retrograde transport of all tested lipids. The only observable effect was to anterograde transport of SM at 30 minutes post-lipid addition, specifically, which accumulates at endolysosomal compartments. This finding strengthens our conclusion that non-vesicular transport is the primary mechanism for the maintenance of organelle lipid compositions (New Figure 3).
- TMEM30A associated flippases affect lipid transport kinetics in a species-specific manner, as transport of PE is affected to a larger degree than transport of different PC species. The results provide evidence that the lipid asymmetry machinery plays an important part in the species selectivity of retrograde lipid transport. (Revised Figure 4, originally Figure 3)
- We tested the role of two lipid transfer proteins (LTP) GLTPD1 (cup-shaped, sphingomyelin-specific LTP) and VSP13C (bridge-like-LTP) for retrograde lipid transport using genetic perturbations. The results suggest that lipid transport kinetics of SM depend on GLTPD1, whereas no effect was observed for the bridge-like LTP, possibly due to redundancy. (New Figure 3, Reviewer Figure 1).

Furthermore, we have carried out experiments to validate our methodological approach and provide additional context:

- We have expanded the biophysical characterization of bifunctional lipids in model membranes. This dataset now covers all probes which were extensively compared to their structurally closest native counterparts in partitioning FRET assays. We included additional measurements of membrane order by highly sensitive Laurdan temperature scans. These datasets allow us to conclude that the combined diazirine and alkyne modifications have an effect on lipid behavior that is either smaller or equal to that of one additional double bond, i.e. the bifunctional acyl chain is most similar to a 16:1 chain (Supplementary Figure 1).
- We performed complementary lipid flux experiments using an isotope labelled, native PC species and compared the metabolic turnover to its bifunctional PC counterparts. We find that the turnover of PCs containing the bifunctional fatty acid is very similar to native lipids containing a monounsaturated fatty acid in its place (Extended Data 5).
- We have carried out cellular fractionation experiments for two PC species at different time points. Analysis of organelle fractions by quantitative lipidomics confirms our lipid imaging results, with retention of higher relative amounts of a saturated PC species at the plasma membrane compared to a polyunsaturated PC species, which is internalized to endomembranes at earlier time points (Supplementary Figure 7).

The following section outlines the structure of this document and gives an overview for the restructuring of the original manuscript version which was required to accommodate the new datasets.

1. Full reviewer comments are provided together with our point-by-point answers to maintain the context. Key results are included here as revised / new figures and the primary conclusion from a new experiment/dataset is highlighted. Our answers to individual reviewer comments are marked by “**Answer**” and are highlighted in blue to enhance clarity.

2. We have added a new main text Figure (Figure 3) to present new results on inhibition of vesicular transport and genetic interference with non-vesicular lipid transport

3. The following main text figures have been revised to include new datasets or datasets which were previously displayed in the supporting information: original Figure 3 (Now Figure 4), original Figure 4 (Now Figure 5)

4. The following ED figures have been revised to include new datasets or to move datasets into a supplementary information:
ED5, ED7, ED8

5. The following new figures have been added to the Supplementary Information

Supplementary Figure 1: Biophysical characterization of bifunctional lipid probes by comparison to native counterparts.

Supplementary Figures 2-7: Additional datasets covering the effects of pharmacological perturbations on lipid and protein transport, influence of genetic perturbations on lipid transport and additional validation experiments, organelle fractionation experiments and further characterization of the TMEM30A effect on lipid flipping

6. All textual changes in the submitted documents are highlighted in **yellow**

7. We have included effect size tests for modelling results which are now included in figures and main text. We opted against regular null hypothesis testing since the sample size for rate constants is the number of model runs. (Detailed explanation in SI)

Answers to specific reviewer questions:

Referee #1 (Remarks to the Author):

In this work, Iglesias-Artola and coworkers develop a novel, coherent and robust pipeline to investigate retrograde lipid flux in cells. Their approach, based on the use of bifunctional lipids, allows to couple organellar fluorescence microscopy with quantitative lipidomics, and it is achieved thanks to the careful optimization of the experimental parameters. Mathematical modeling of the observed fluxes using straightforward kinetic models allows the authors to interpret their data and to extract meaningful information on the fate of different lipid species upon their cellular internalization.

From a technical and methodological point of view, this work is a remarkable tour-de-force that significantly advances our current technology to investigate retrograde lipid fluxes, and I expect it to become a landmark methodology in the field. Specifically, I have no criticisms on this aspect and I find it an amazing technical work.

From a biology point of view, on the one hand the authors nicely confirm previous results/expectations, such as the prevalence of non-vesicular trafficking or the key role of PM flippases, on the other hand they provide novel insights, for example into the spatial and temporal separation between lipid transport and metabolism or into the head and chain specificity of lipid internalization. Notably, because of the nature of their approach, they provide a quantitative estimation of these processes, which was mostly not available with previous methods.

Answer:

We thank reviewer #1 for the positive overall assessment of our work, in particular with regard to the methodological novelty of our work.

At the same time, I find that the coupling of the newly developed technology with genetic perturbations is severely underdeveloped, causing most of the novel observations to remain somewhat superficial. Specifically, all the constants extracted from the kinetic models are “effective” ones, likely resulting from multiple pathways. In order to identify the origin of the observed differences (e.g. between lipid species), and hence to investigate causative relationship in lipid biology, the authors should extensively and systematically couple their approach with genetic perturbations, and identify the origin of the observed differences. Some examples/suggestions:

1. Does the observed difference between lipid species originate from differences in lipid flipping at the PM or from specificity in lipid transport?
2. What is the role of the the different lipid transport proteins proposed to work at the ER-PM contact sites?
3. What is the contribution of shuttle (ORP...) versus bridge (VPS13A-D) proteins?
4. For lipid metabolism: what is the relative contribution of ACAT1 and ACAT2? And of DGAT1 and DGAT2? What about the enzymes upstream of the last step of sterol-ester and triacylglycerol synthesis?

While I understand that most of these investigations could be explored in future studies, I believe that the article would strongly benefit from a deeper investigation (achieved with genetic perturbations) in at least one of these areas. This will highlight the potential of this beautiful new approach (which in its current form is more suited to more method development journals, such as Nature Method, for example) to gain novel important biological insights.

Answer:

We agree that the combination of lipid imaging with genetic and pharmaceutical perturbations provides a promising avenue to delineate the mechanisms of the lipid transport networks. While we agree that a full investigation of the entire network is an exciting application of our work, it is currently beyond the scope of this manuscript and the throughput of the method. Therefore, taking into account suggestions from all three reviewers, we have decided to focus our new experiments on two main areas: The respective contributions of lipid flippases and lipid transfer proteins to selective lipid transport and the prevalence of non-vesicular lipid transport over vesicular lipid transport.

1. Impact of lipid flippases on species specific lipid transport rates

We tested the lipid selectivity of the TMEM30A flippase subunit in the CRISPR-KD cell line. To this end, we compared the transport kinetics of PE to two additional PC species (fully saturated and polyunsaturated). We found that the transport of both PC species was less affected compared to the 3-fold slower transport of PE in TMEM30A KD cells. Transport of the saturated PC was hardly affected at all, while transport of the polyunsaturated PC species was slightly increased and its steady-state distribution was shifted towards the PM compared to WT cells (Revised Figure 4).

These findings demonstrate that lipid flippases do indeed contribute to selective and fast non-vesicular lipid transport, both with regard to the speed of transport and the steady state lipid distributions. Since TMEM30A is a subunit of many aminophospholipid flippases (ATP8A1, ATP8A2, ATP8B1, ATP8B2, ATP10A, ATP10B, ATP10D, ATP11A, ATP11B, ATP11C, according to the STRING database) a dedicated project will be required to unravel the exact species selectivity of the different flippases. The new data is now displayed in the revised Figure 4 (Figure 3 in the original submission), shown below and new Supplementary Figure 6.

Figure 4 Genetic perturbation experiments confirm involvement of flippases in species-specific directional lipid transport. **a.** Schematic representation of lipid trans-bilayer movement (lipid flipping) and non-vesicular lipid transport by lipid transfer proteins. **b.** Kinetic model for exchange of lipids between the PM and the ER. **c.** Comparison of time-course experiments for PE(18:1/Y) show that lipid internalization dynamics are slower in HCT116 TMEM30A KD cells compared to HCT116 wild type cells. Coloured arrows indicate lipid localization in different membranes types (green: PM, yellow: ER). Scale bars: 10 μ m. Images are brightness-contrast adjusted to facilitate comparing intracellular lipid localization. **d.** Quantification of PE(18:1/Y) internalization kinetics and

model fits. **e.** Quantification of PC(Y/16:0) internalization kinetics and model fits. **f.** Quantification of PC(Y/20:4) internalization kinetics and model fits. **d-f.** Kinetics were constructed from 5 independent time points (mean and SD) containing 3-15 field of views each with 5-10 cells. **g.** Rate constants and quasi-equilibrium constants for retrograde and anterograde plasma membrane-ER lipid transport for PC(Y/16:0), PC(Y/20:4) and PE(18:1/Y). Mean and SD were calculated from 100 MC model runs. Pairwise effect size tests, Cohen's d values: >0.01 very small (VS), >0.20 small (S), >0.50 medium (M), >0.80 large (L), >1.20 very large (VL), >2.00 Huge (H).

2. Impact of lipid transfer proteins on specific retrograde lipid transport

The high number and potential redundancy of lipid transfer proteins makes it difficult to provide a comprehensive answer to this question, as there are > 130 individual cup-shaped LTPs alone which would have to be investigated, according to Titeca et al.:

<https://www.biorxiv.org/content/10.1101/2023.12.21.572821v1.full.pdf>). While we agree that a full investigation of the entire network is an exciting application of our work, it is currently beyond the scope of this manuscript. Instead, we have chosen to perturb two lipid transfer protein (GLTPD1 and VPS13C), which allowed us to address specific mechanistic aspects that are particularly relevant for the current story.

GLTPD1 is one of the few cup-shaped LTPs that has been suggested to have a single primary cargo lipid (Sphingomyelin) by Titeca et al.. The effect of a GLTPD1 KO would therefore provide (i) a useful estimate of the maximal expected effect size for a single gene KO of a cup-shaped LTP and (ii) constitute a direct cross-check of our finding that there is significant retrograde non-vesicular transport of sphingomyelin. VPS13C is a known bridge-like lipid transfer protein for bulk lipid transport of structural membrane lipids on several ER-organelle membrane contact sites, but it appears to be primarily localized at Endosome-ER contact sites. Therefore, a VSP13C KO could provide a means to estimate non-vesicular lipid transport between endosomes and the ER.

We generated KO cell lines for both genes and quantified the effects on lipid transport of PE, PC and SM. We were excited to see that the GLTPD1 KO specifically reduced SM transport whereas transport of PE and PC lipids was less affected (New Figure 3, shown below). Our new results indicate that individual cup-shaped lipid transport proteins can indeed affect the speed of bulk intracellular lipid transport of individual lipids and thereby contribute to specificity. We complemented this dataset with pharmacological inhibition of vesicular trafficking (requested by reviewers #2 and #3). We find that retrograde transport of all lipids is unaffected by blocking vesicular trafficking, whereas anterograde transport of SM involves vesicular steps, as previously reported. Together, these data confirm one of the key findings of the paper that retrograde transport is indeed primarily non-vesicular and we show that GLTPD1 that contributes significantly to overall SM flux in the cell.

Figure 3 Pharmacological and genetic perturbations confirm non-vesicular transport as the primary retrograde lipid transport route. **a.** Inhibition of vesicular trafficking between the Golgi and the ER (Brefeldin A) and of endosome formation and trafficking (Wortmannin). **b.** Transferrin uptake and localization and lipid localization in control cells and drug treated cells at 4 min and 30 min after lipid loading. Scale bar: 10 μ m. Transferrin images are shown at identical settings, lipid images are brightness-contrast adjusted to allow for comparing lipid distributions at different timepoints. **c.** Quantification of lipid distribution between the ER and endo-lysosomes (mean and SD of 8-10 fields of view) after treatment DMSO (gray), Brefeldin A (pink), Wortmannin (light blue). **d.** Schematic representation of GLTPD1-mediated lipid transfer and kinetic model used to assess effects of GLTPD1 KO on non-vesicular lipid transport. **e.** Comparison of SM(Y) time-course experiments in GLTPD1 KO and U2OS WT cells. Scale bar: 10 μ m. **f.** Rate constants for retrograde and anterograde plasma membrane-ER lipid transport for PC(18:1/Y), PE(18:1/Y) and SM(Y). Kinetics were constructed from 6 independent time points (mean and SD) containing 5 field of views each with 5-10 cells. Mean and SD were calculated from 100 MC model runs. Pairwise effect size tests, Cohen's d values: >0.01 very small (VS), >0.20 small (S), >0.50 medium (M), >0.80 large (L), >1.20 very large (VL), >2.00 huge (H).

Effects of the VPS13C KO on transport of PC, PE and SM species between the endosomes and the ER and the plasma membrane and the ER were smaller than for the GLTPD1 KO and overall inconclusive. We include the results for the reviewers (see Reviewer Figure 1), but decided to not include this dataset in the manuscript due to the space limitations. Currently it is difficult to interpret this result, as there are multiple possible explanations. These include for example redundancy, low non-vesicular ER-Endosomes transport as a fraction of overall lipid

flux and the fact that later steps in the transport network are less well resolvable due to the PM starting point for the probes.

Reviewer Figure 1. VPS13C KO effects on lipid transport are small and inconclusive. **a.** Schematic representation of lipid transport by bridge-like LTPs. **b.** Kinetic model used to assess transport between the plasma membrane, the ER and Endosomes. **c.** Model fits of lipid transport for PC(18:1/Y), PE(18:1/Y) and SM(Y) in U2OS WT and VPS13C KO cells. Rate constants describing PM-ER lipid transport for PC(18:1/Y), PE(18:1/Y) and SM(Y) in U2OS WT and VPS13C KO cells. Mean and SD were calculated from 100 MC model runs. Pairwise effect size tests, Cohen's d values: >0.01 very small (VS), >0.20 small (S), >0.50 medium (M), >0.80 large (L), >1.20 very large (VL), >2.00 Huge (H). **e.** Quasi-equilibrium constants describing PM-ER lipid transport for PC(18:1/Y), PE(18:1/Y) and SM(Y) in U2OS WT and VPS13C KO cells.

Overall, after performing the additional experiments suggested by reviewer #1 we now conclude that both flippases and LTPs contribute to the specificity of intracellular lipid transport. We believe that, in the long run, our lipid imaging approach in combination with genetics has the potential to untangle the full network of the lipid transport machinery and determine actual rates of lipid flipping and transport instead of effective rates. A major genetic engineering effort in combination with a high-throughput version of the lipid imaging pipeline will be required to crack this problem over the next decade.

Referee #1 (Remarks on code availability):

Most areas of the code are clear and function appropriately. However, small demo datasets to test that the python and matlab scripts used for the mass spectrometry data analysis and

for the kinetic model building, respectively, are currently missing. Without these datasets, it is not possible to check whether the results presented in the paper are reproducible.

Answer:

We apologize for this oversight. All quantified MS data as well as raw lipid imaging data were uploaded shortly after the original version was sent out for review and should now be available. There is now a matlab demo including a test dataset for the kinetic modelling. The overall size of the MS dataset that was used to adjust the kinetic models for metabolism is not large, so it should be possible to work with the full datasets. Most of the analysis of raw MS data was done using published software (Lipidexplorer), and the uploaded python scripts were used for data formatting and filtering. We have uploaded the full Lipidexplorer output dataset, so that this code can now also be tested. The new version of the demo dataset can be found here: <https://doi.org/10.17617/3.BRSGLA>

Referee #2 (Remarks to the Author):

This study investigates retrograde lipid transport and lipid metabolism using exogenous clickable phospholipid. The lipids are solubilized with alpha-methyl-cyclodextrin and added to cells, initially incorporating into the plasma membrane (PM). The transport of the lipids (or their derivatives) is followed over time by crosslinking, fixation, and detection using click chemistry. State of art ultra-high resolution mass spec is used to determine how the lipids are metabolized over time. This is a novel strategy. Previous studies have used labeled lipids, typically fatty acids or cholesterol, and fractionation or enzymatic conversion to estimate rate of lipid movement from the PM to various compartments. There have been few studies in which diacyl phospholipids are added to cells and using cyclodextrin deliver the lipids is a good idea. There are four main conclusions: (1) retrograde transport is primarily non-vesicular, (2) flippases play a role in lipid transport, (3) lipid metabolism is about 10x slower than transport, and (4) the rate of incorporation of exogenous lipids into neutral lipids is affected by the saturation of the acyl chains. While the work is well done and highly technically innovative, there are concerns about the significance of the conclusions and the interpretation of the findings.

Answer:

We thank reviewer #2 for the overall positive assessment of our work, in particular with regard to the methodological novelty and highlighting the conceptual advance of an approach to measure the rates non-vesicular and vesicular transport in the same experiment. We address the points mentioned by reviewer #2 in our answers below.

Here are the major concerns.

1. It is already well established that exogenous cholesterol, fatty acids, and lysolipids are rapidly moved by non-vesicular mechanisms from the PM to the ER. The exogenous lipids are initially incorporated into the PM and converted to other lipids in the ER, an indirect indication of transport; for example, exogenous cholesterol is rapidly converted to cholesteryl ester, which happens in the ER. The rate of conversion is too fast to be entirely explained by vesicular transport and is not blocked by genetic or chemical treatments that block vesicular trafficking, indicating that transport is largely non-vesicular. This study uses phospholipids with 2 acyl chains, which has only been done in a few previous studies and not nearly as carefully as here. There have also been several papers on the transport of newly synthesized lipids from the to the PM and other organelles that suggest lipid can be moved by non-vesicular pathways that have a high capacity to move lipids. Therefore, the finding that exogenous phospholipids with two acyl chains also move by non-vesicular pathways to internal membranes is not terribly surprising. However, previous studies have not estimated what fraction of exogenous lipid (or endogenous lipid) moves by vesicular and non-vesicular mechanisms, which is a notable advance provided here.

There are some concerns about the transport assays that should be addressed. One is the rate of conversion of the exogeneously added lipids to other species and how this affects the calculated transport rate. For each time point in the transport assays (for example in Fig 2c), the percent of the added lipid that has been converted to other species should be shown. At the later time points, is a significant fraction of the signal in an organelle from lipid species other than the one added to the cells? This data and the transport data should be given for all the lipids shown in Fig 1b.

Answer:

We agree that the interplay between transport and metabolism is an important point that needs to be clarified. While the requested data was already part of the original manuscript (Fig. 5b, Supplementary Figure 8 in the revised version), we now realize that our explanation was not clear. Panels a and b of revised main text figure 5 shows the percentage of initially supplied lipid species as a fraction of all bifunctional lipids in the lipidome. Since other bifunctional lipids can only be generated by metabolism, we reasoned that this is the best available measure for overall metabolism of bifunctional lipids. Due to reviewer #2's comment, we realized that the

y-axis label in figure 3a should read: “initial species as % of all bifunctional lipids” rather than “% of bifunctional lipids”. We have changed the axis label accordingly.

Supplementary figure 8 (revised manuscript) shows the datasets for the lipid fraction of the initially supplied species as well as the % of all bifunctional lipids of the global lipidome. We use these percentages to weigh the lipid transport data used as input for the modelling for models 2a,b and 3a,b to account for lipid metabolism (models 2a,b) and overall loss of bifunctional lipid probes (models 3a,b). The rate constants for models 1a,b-3a,b are given in panels d and f of main text figure 2, and the corresponding Extended Data (ED 7) and Supplementary Tables 1-9. Overall, we find that accounting for metabolism does not change the outcome with regard to transport rates significantly. This is reasonable as there is a clear time-scale difference between lipid transport (minutes) and metabolism (hours). In a nutshell, the steady-state lipid distribution is dominated by lipid transport rather than metabolism.

We have expanded the description of the models and how metabolism is incorporated in the main text (Lines 175-186) and added references to the metabolic datasets in the modelling section (Lines 184-185). We furthermore include a portion of the data previously shown in Supplementary Figure 1 (now Supplementary Figure 8) in Figure 5, shown below.

Figure 5 Lipid metabolism is approximately one order of magnitude slower than lipid transport. a.

Bifunctional lipid retention and turnover. The fraction of initially supplied species as percentage of all bifunctional lipids and the fraction of bifunctional lipids of the total lipidome are exemplarily shown for PC(18:1/Y), PE(18:1/Y) and SM(Y). **b.** Fraction of initially supplied lipid probe as % of the bifunctional lipidome as a proxy for the speed of lipid metabolism. Solid lines indicate mono-exponential fits. SM(Y) data was not fitted as very little interconversion was observed, instead a linear interpolation is shown. Error bars: Mean and SD of $n = 3$ independent experiments. **c.** Comparison of determined mono-exponential rate constants for the metabolism of individual lipid species. Error bars: Mean and SE of $n = 3$ independent experiments **d.** Comparison of transport and metabolic rate constants shows that lipid transport is at least one order of magnitude faster. **e.** Lipid transport and metabolism rate constants

are highly correlated for PC species despite a clear timescale separation. Error bars metabolic rate constants: SE, transport rate constants: SD, calculated from 100 MC model runs.

Another concern about the transport assay are the results with sphingomyelin (SM). Since SM (and other complex sphingolipids) are synthesized on the extracytoplasmic leaflet of the Golgi and do not flip, it is widely thought they are not available to lipid transport proteins operating in the cytoplasm. There are also no lipid transport proteins known to bind SM. Therefore, the transport of intact SM should be almost entirely vesicular, which does not seem to be the case here. The ability of SM to be moved by non-vesicular pathways should be assessed by blocking vesicular transport, for example by depleting cells of NSF. If it turns out that SM transport is not blocked by inhibiting vesicular transport, the authors should explain why previous studies are incorrect.

Answer:

1. Clarification and textual changes

The reviewer's point that SM is synthesized on the exoplasmic leaflet of the Golgi membrane is correct and we do not challenge the notion that *anterograde* transport of newly synthesized Sphingomyelin is mostly vesicular. We describe anterograde transport from the ER to the PM in our main models 1a-3a with a summary rate that encompasses both vesicular and non-vesicular trafficking. This was indicated in the reaction scheme of Figure 2 in the original version, and also described in detail in the section "Kinetic models and model performance" in the supporting information, but we now realize that this is a critical piece of information and explicitly describe it in the main text (lines 176-182).

We do find that *retrograde* transport of SM from the plasma membrane is mostly non-vesicular. Retrograde transport of lipids is much less studied than anterograde lipid transport, as it cannot readily be monitored using metabolic labelling approaches and a systematic mechanistic investigation into retrograde lipid transport using near native lipid probes or isotope-labelled lipids has not been carried out to the best of our knowledge.

Retrograde non-vesicular transport of SM requires a) that lipid translocases (flippases or scramblases) exist that accept SM as a substrate, which has been convincingly shown by the Holthuis lab for the PM and lysosomes:

<https://www.nature.com/articles/s41467-022-29481-4>

and b) that LTPs exist that can transport SM. This is likely true for bridge-like LTPs, but there is also a SM-specific cup-shaped LTP, GLTPD1, as demonstrated by the Gavin lab:

<https://www.biorxiv.org/content/10.1101/2023.12.21.572821v1.full.pdf>

2. New experimental data (new Figure 3, new Supplementary Figures 3 and 5)

We agree with the reviewer that additional experimental evidence with regard to SM, but also other lipid species would further support to our central finding that non-vesicular lipid transport primarily governs the maintenance of organelle lipid compositions. We thus pharmacologically inhibited vesicular trafficking using two different compounds, Brefeldin A, which blocks COPI coat formation at Golgi membrane and Wortmannin, which broadly affects endosome formation and trafficking. We followed transferrin internalization to monitor protein cargo uptake together with lipid internalization and used ER and endo-lysosomal markers to assess the distribution of lipid signal between the trafficking compartment and the ER. We monitored lipid and transferrin localization at the 4 min timepoint, which is early in the process and should capture perturbations of retrograde transport of both lipid and protein cargoes. We furthermore monitored lipid and transferrin localization at the 30 min timepoint, which is near steady state for most lipids and should therefore also provide information with regard to the anterograde transport. To obtain a comprehensive dataset, we chose to monitor four lipid species, SM(Y), PE(18:1/Y), PC(Y:16:0), PC(Y/20:4), which cover multiple lipid classes and exhibit very different transport dynamics. We find the following:

- At 4 min, transferrin uptake is markedly reduced in cells treated with 30 μ M Brefeldin A or 3 μ M Wortmannin, a clear indication of a slower endocytic rate (new Supplementary figure 4, shown below).
- On the other hand, at 4 min, lipid distribution is indistinguishable from control cells for all investigated lipid species, exemplary shown for PE(18:1/Y) in the new Figure 3, shown below and in new Supplementary figures 5-8 for all lipids.
- The relative distribution of lipid material between the endo-lysosomal system and the ER is unaffected by the pharmacological interventions

These findings indicate that a lower endocytic rate has only a marginal effect on retrograde lipid transport and fully support our finding that retrograde lipid transport is primarily non-vesicular.

- At 30 min, transferrin is trapped in hyperfused ER-Golgi compartments in Brefeldin A treated cells and in enlarged perinuclear endosomes in Wortmannin-treated cells, confirming that the intended effects of the vesicular transport system were indeed achieved.
- Overall lipid distribution and the ratio of lipid signal between ER and the endo-lysosomal compartment does not differ significantly at 30 min from control cells for all lipids in Brefeldin A treated cells, indicating that ER-Golgi vesicular transport does not play a major role in anterograde lipid transport.
- This is also true for PE(18:1/Y), PC(Y:16:0), PC(Y/20:4) in Wortmannin treated cells, but SM(Y) accumulates in endosomes at 30 min, indicating that functional endosomes are indeed required for anterograde transport of SM.

Taken together, these findings indicate a prevalence of non-vesicular mechanisms for both retrograde and anterograde lipid transport for glycerophospholipids. For SM, retrograde transport appears to be mostly non-vesicular, which is also supported by the knockout of the GLTPD1 lipid transfer protein, which reduced retrograde transport of SM, specifically. Anterograde transport of SM, however appears to have a strong vesicular component, not only for de novo synthesized SM, but also for exogenously supplied material.

We have added a section to the main text describing these findings, which are presented in new Figure 3 shown below and new Supplementary figures 3 and 5. See lines 236-255 in the main text.

Figure 3 Pharmacological and genetic perturbations confirm non-vesicular transport as the primary retrograde lipid transport route. **a.** Inhibition of vesicular trafficking between the Golgi and the ER (Brefeldin A) and of endosome formation and trafficking (Wortmannin). **b.** Transferrin uptake and localization and lipid localization in control cells and drug treated cells at 4 min and 30 min after lipid loading. Scale bar: 10 μ m. Transferrin images are shown at identical settings, lipid images are brightness-contrast adjusted to allow for comparing lipid distributions at different timepoints. **c.** Quantification of lipid distribution between the ER and endo-lysosomes (mean and SD of 8-10 fields of view) after treatment DMSO (gray), Brefeldin A (pink), Wortmannin (light blue). **d.** Schematic representation of GLTPD1-mediated lipid transfer and kinetic model used to assess effects of GLTPD1 KO on non-vesicular lipid transport. **e.** Comparison of SM(Y) time-course experiments in GLTPD1 KO and U2OS WT cells. Scale bar: 10 μ m. **f.** Rate constants for retrograde and anterograde plasma membrane-ER lipid transport for PC(18:1/Y), PE(18:1/Y) and SM(Y). Kinetics were constructed from 6 independent time points (mean and SD) containing 5 field of views each with 5-10 cells. Mean and SD were calculated from 100 MC model runs. Pairwise effect size tests, Cohen's d values: >0.01 very small (VS), >0.20 small (S), >0.50 medium (M), >0.80 large (L), >1.20 very large (VL), >2.00 huge (H).

2. The finding that knock down of TMEM30A slows transport is interesting, but there are no controls to show that the effect is direct. It remains possible that TMEM30A affects membrane properties or the function of proteins other than lipid transport proteins. TMEM30A depletion could also affect vesicular transport. Other flippases should also be investigated.

Answer:

We now provide multiple lines of evidence that KD of TMEM30A has a direct effect on selective lipid transport. First, by comparing the primary flippase substrate PE(18:1/Y) to PC species, we demonstrate that transport of PE is decelerated to a significantly higher degree than is found for PC. This substrate-specific effect speaks against a general alteration of bulk membrane properties. These results are included in the new revised Figure 4 (shown below) and the corresponding Supplementary Figure 6. Second, we furthermore show that the steady state distribution of aminophospholipids at the plasma membrane is indeed altered by detecting phosphatidylserine exposure on the outer plasma membrane leaflet in KD cells, but not in WT cells, confirming that the KD cells do in fact exhibit an aminophospholipid flipping defect (New Supplementary Figure 4, shown below). Finally, regarding a possible effect of TMEM30A KD on vesicular transport, we pharmacologically inhibited vesicular transport and found no effect on the fast bulk transport of PE and PC (New figure 3, shown above in the answer to point 1).

Regarding the point of other flippases to be tested, we specifically chose TMEM30A as it is a shared subunit of a group of 11 mostly aminophospholipid flippases, and the KD affects the entire group of interacting flippases. By targeting an entire group of flippases we aimed to avoid having to do one-by one KOs. We agree that a full analysis of all flippases is an exciting project, but it is beyond the scope of the current manuscript.

Supplementary Figure 4 TMEM30A KD results in pronounced localization of PS on the outer plasma membrane leaflet. Upper panels: HCT116 WT cells treated with the small molecule plasma membrane marker Cell mask red and the PS sensor Lact-C2-EGFP. No recruitment of Lact-C2-EGFP from the extracellular medium to the outer plasma membrane leaflet is visible. Lower panels: HCT TMEM30A KD cells treated with the small molecule plasma membrane marker Cell mask red and the PS sensor Lact-C2-EGFP. Recruitment of Lact-C2-EGFP from the extracellular medium to the outer plasma membrane leaflet is indicative of a flipping defect. Scale bar: 10 μ m.

Figure 4 Genetic perturbation experiments confirm involvement of flippases in species-specific directional lipid transport. **a.** Schematic representation of lipid trans-bilayer movement (lipid flipping) and non-vesicular lipid transport by lipid transfer proteins. **b.** Kinetic model for exchange of lipids between the PM and the ER. **c.** Comparison of time-course experiments for PE(18:1/Y) show that lipid internalization dynamics are slower in HCT116 TMEM30A KD cells compared to HCT116 wild type cells. Coloured arrows indicate lipid localization in different membranes types (green: PM, yellow: ER). Scale bars: 10 μ m. Images are brightness-contrast adjusted to facilitate comparing intracellular lipid localization. **d.** Quantification of PE(18:1/Y) internalization kinetics and model fits. **e.** Quantification of PC(Y/16:0) internalization kinetics and model fits. **f.** Quantification of PC(Y/20:4) internalization kinetics and model fits. **d-f.** Kinetics were constructed from 5 independent time points (mean and SD) containing 3-15 field of views each with 5-10 cells. **g.** Rate constants and quasi-equilibrium constants for retrograde and anterograde plasma membrane-ER lipid transport for PC(Y/16:0), PC(Y/20:4) and PE(18:1/Y). Mean and SD were calculated from 100 MC model runs. Pairwise effect size tests, Cohen's d values: >0.01 very small (VS), >0.20 small (S), >0.50 medium (M), >0.80 large (L), >1.20 very large (VL), >2.00 Huge (H).

3. It is already well known that the saturation level of fatty acids affects their rates of incorporation of into neutral lipid. It is not clear what the findings here add to what is already known or how this is a significant advance.

Answer:

We do not claim a new finding on the influence of fatty acid saturation level on incorporation into neutral lipids which is indeed well-described in the literature. We find that it matters for cholesterol ester incorporation whether the fatty acid – irrespective of saturation degree – was previously attached to the sn1 or the sn2 position of a phosphatidylcholine lipid. This is an important finding, as it shows that the origin of chemically identical Acyl-CoA molecules matters for metabolic outcome. This cannot be explained by regular chemical kinetics but has to be a consequence of either highly compartmentalized metabolism on a sub-organelle scale or direct substrate handover mechanism between enzymes. Therefore, we consider it a substantial advance and also a beautiful illustration of the power of our approach.

4. More needs to be done to verify that biophysical properties of the lipids shown in Fig. 1b. The biophysical properties of membrane composed entirely of the PC or PA species shown in

Fig 1b should be compared to membranes composed of the identical lipids lacking the diazirine and alkyne groups. For the PC species, their ability to be transported in vitro by a PC lipid transport protein like Sec14 should be assessed and compared PC species with the diazirine and alkyne groups.

Answer:

We agree that the biophysical characterization of the bifunctional lipids is very important. We have now characterized all probes used in this study using FRET partitioning assays and compared them to their native counterparts lacking diazirine and alkyne moieties. We have furthermore conducted Laurdan temperature scans in vesicles containing bifunctional PC and PE probes. The combined results indicate that the biophysical property changes introduced by the diazirine and alkyne units are rather small and comparable to or slightly smaller than the effect of one additional double bond in the acyl chain, i.e. the bifunctional acyl chain is most similar to a 16:1 chain. These data are now shown in Supplementary Figure 1, shown below.

In addition, we performed control experiments to determine if metabolic conversion of bifunctional lipids is changed compared to endogenous lipids. We compared the metabolic fate of oleic acid and palmitic acid containing bifunctional PC species with a native, isotope labelled species, PC(18:1/16:0[¹³C]) delivered in the same fashion to the outer leaflet of the plasma membrane. Our results show that this species is metabolized similarly to the bifunctional PC species PC(Y/16:0) and PC(16:0/Y), whereas the oleic acid containing species PC(Y/18:1) and PC(18:1/Y) are metabolized faster. The level of retention of the original PC(18:1/16:0[¹³C]) as % of all labelled lipids is also similar to the palmitate containing bifunctional lipid species. This implies that the metabolism of bifunctional lipid species most closely resembles native lipids that contain a monounsaturated fatty acid chain in the place of the bifunctional fatty acid, which matches the biophysical characterization in vitro.

Since the biophysical properties of bifunctional lipids are comparable to endogenous lipids in vitro and the metabolic conversion rates of bifunctional lipids are comparable to their endogenous C13 labelled counterpart in cells, we are confident that the bifunctional fatty acid is a minimal perturbation that still allows us derive species specific conclusions. Performing additional transport assays would require to purify and characterize lipid transport proteins in vitro which is beyond the scope of this study.

Supplementary Figure 1 Biophysical characterization of bifunctional lipid probes by comparison to native counterparts (Supplement to ED figure 1a-c). **a.** Formation of ganglioside nanodomains leading to faster deexcitation of Bodipy-FL-GM₁ donors via FRET is unaffected by replacing the closest structurally related native lipid with the bifunctional variant for all probes. **b.** Laurdan temperature scans indicate that biophysical implications of introducing a bifunctional fatty acid are small.

Extended Data Figure 5 (Data on metabolism of isotope-labelled species versus bifunctional lipid species shown). Ultra-high resolution (UHR) shotgun lipidomics of bifunctional lipid probes. **e.** Comparison of bifunctional lipid metabolism with the native, isotope labelled, monounsaturated PC species PC(18:1/16:0^[13C]). **f.** Development of the PC species distribution of PC(18:1/16:0^[13C]) and palmitate-containing bifunctional PC species shows similar persistence of the original species in the labelled lipidome and similar product species forming. Shown are Mean and 95% CI of 3 biological repeats containing 3 technical replicates each

Referee #3 (Remarks to the Author):

In this manuscript, André Nadler's group has adapted innovative methods to study lipid transport between membranes in prototypical human cells. The method is based on the chemical synthesis of bifunctional lipids that contain both a photoactivatable and a clickable chemical label. The method is not new, but the authors have adapted it to systematic approaches and integrated pulsed-phase fluorescence imaging, ultra-high resolution mass spectrometry and mathematical modeling. The approach is well thought-out and elegant. The authors have created a library of 9 bifunctional lipid probes covering four different lipid classes: phosphatidylcholine (PC), phosphatidic acid (PA), phosphatidylethanolamine (PE) and sphingomyelin (SM). For the PC class, they also included 6 species with different fatty acids. The strength of this approach lies in the fact that it is based on pulse-chase experiments, enabling the authors to measure (and compare) the transport kinetics of different lipid classes and species. Such a quantitative analysis of lipid transport in a cell had, to my knowledge, never been carried out on this scale. This led to several interesting findings: FA composition is important and affects lipid transport kinetics, and retrograde lipid transport from the PM to the ER is faster and more selective than vesicular transport. The authors then address mechanistic questions concerning the directionality of non-vesicular transport (often against lipid gradients) and follow the hypothesis that active reversal of the trans layer between membrane sheets catalyzed by flippases and floppases may contribute. Overall, this work represents a tour de force and the first attempt to understand the general principles of non-vesicular lipid transport in a quantitative way. The study of lipids and their transport remains poorly understood due to the difficulty of analyzing and visualizing them in the cell. This work is a first attempt to answer this question and may lead/inspire many follow-up studies. Below are a number of points that I feel need to be addressed.

Answer:

We thank reviewer #3 for the very positive assessment of our work, both with regard to the methodological novelty and the biological findings

General points:

1) The reasons for selecting lipids for chemical engineering (and the scope of this work) should be explained. Are certain classes of lipids (i.e. those that are lacking: PS, PI, PG, cardiolipins) ceramides) not suitable for chemical synthesis, or have they been omitted because they were less relevant in the context of plasma membrane lipid supply, or because they are toxic (in high abundance and/or when found in the plasma membrane)? Why have the various FAs only been studied in the context of PC (and not other GPL/sphingolipids)? This information that could be useful for follow-up studies.

Answer:

Probe set choice:

The reasons for the selected probe set are the following: PE and PC are the main phosphoglycerolipid classes in U2OS cells, representing ~33 and ~20% of the total lipidome (See also ED 1d). Since we wanted to draw conclusions on bulk lipid transport in cells, we chose to cover the most abundant lipid classes PE and PC. The choice to use the PC lipid class for the assessment of species-specific transport was made for a number of reasons:

1. Synthetic convenience, as routes to bifunctional PCs are shorter than for PEs
2. PC is the main membrane forming lipid for most organelles
3. PCs are generally the most abundant phospholipids

In addition to PC and PE probes we included SM as the most abundant sphingolipid and PA as the central metabolic intermediate in glycerophospholipid metabolism. Taken together, we reasoned that this set of probes was of an optimal size and diversity to allow us to answer the most pressing questions with regard to lipid flux (transport and metabolism) in cells, while keeping the dataset size (42 TB for the core lipid transport pulse-chase experiments) within a range where automated data analysis on a cluster would take no more than a few hours and

actual manual imaging experiments including method development were manageable during a typical PostDoc time window of a few years. We have added a brief section on probe selection to the corresponding results section in the main text. See lines 83-92.

Species-specific lipid transport for other lipid classes:

We have (not yet) done the analysis of species-specific lipid transport for other lipid classes, as this would have made the (manual) pulse-chase imaging experiments and related data analysis prohibitively time-consuming and expensive, also to keep this manuscript to a manageable size. In a follow-up project, we are currently optimizing a high-throughput strategy, which will hopefully greatly reduce imaging time and data analysis as well. We feel that making new probes for other lipid classes and performing the corresponding lipid imaging experiments is beyond the scope of the current paper, but we are of course looking into this aspect in follow-up studies. The choice of PC as the model lipid class for studying species-specific lipid transport is now covered in the probe selection paragraph.

Synthetic accessibility:

There are no fundamental reasons why bifunctional lipid probes for other lipid classes cannot be made, e.g., membrane permeable version for most PIPs have been generated by the Schultz laboratory which requires quite sophisticated synthetic routes. Bifunctional lipid syntheses are usually challenging and time-consuming but feasible, making it primarily a resource allocation question. We are currently actively expanding the set of available probes, in close discussions with a number of new collaborators. Building a comprehensive set of bifunctional lipid probes is a task that will occupy chemists in the lab for at least the next decade, but we think it is a very worthwhile endeavor for lipid biology at large.

Toxicity:

So far, we have not yet encountered toxicity of any of the probes at the levels used for lipid imaging experiments. This may indeed be different for PS, or bona-fide mitochondrial lipids delivered to the plasma membrane. We will certainly test it in the relevant cases (PS, CL) once we have the probes in hand, but would refrain from speculation about the effects of yet-to-be-made probes on cell viability. We hope that reviewer #3 agrees with us in this regard.

2) As a control, the authors measured the possible effects of bifunctional lipid probes on general membrane properties in an in vitro biochemical assay, which is an important control - they claim to find no effect. This requires some explanation. Why is it that membranes containing PC(20:4) have properties similar to those of PC(16:0) or PC(18:1)?

Answer:

In the original submission we actually only provided data of membranes containing a single bifunctional lipid species, PC(18:1/Y) at 5% total lipid content and compared with membranes that contained 5% POPC as a close structural native homolog. The 5% level was chosen as this is approximately 2-5-fold higher than the maximal bifunctional lipid content in the cellular lipidome at early time points during the pulse chase experiments. We did not see any difference between the chosen probe and the native counterpart and extrapolated that the effect would be similar for other lipids. We agree with reviewer #3 that this initial characterization should be expanded. We now provide FRET assays in model membrane systems that compare all utilized bifunctional probes to their closest native structural counterparts, again at 5% lipid content. The resulting data show that the influence of the diazine and alkyne moieties on lipid biophysical properties is indeed small or negligible for all probes when compared to their native counterparts. These data also show that structurally dissimilar lipids (e.g. SM(Y) and PC(Y/20:4)), differentially affect membrane properties as reviewer #3 correctly pointed out.

We furthermore conducted Laurdan temperature scan experiments in pure bifunctional lipid membrane models to compare bifunctional PE and PC lipid probes with a set of structurally related native lipid species to estimate the influence that the presence of the diazine and

alkyne moieties have on membrane order. Again, we find that the effect is rather small, the maximally observed effect is comparable to the presence of one additional double bond in the lipid structure. These data are now included in new Supplementary Figure 1 (shown below) and discussed in the corresponding section in the SI and briefly in the main text (lines 95-99).

Supplementary Figure 1 Biophysical characterization of bifunctional lipid probes by comparison to native counterparts (Supplement to ED figure 1a-c). **a.** Formation of ganglioside nanodomains leading to faster deexcitation of Bodipy-FL-GM₁ donors via FRET is unaffected by replacing the closest structurally related native lipid with the bifunctional variant for all probes. **b.** Laurdan temperature scans indicate that biophysical implications of introducing a bifunctional fatty acid are small.

3) They use the same (liposome-based) delivery system for all probes, but can they really rule out that liposomes containing different bifunctional lipids lead to similar bifunctional lipid uptake mechanisms? As bifunctional lipids can be cross-linked to proteins, pull-down analyses followed by proteomics (after cross-linking) (at early time points) would help answer this question, at least for a few important cases (i.e. identified proteins/machines involved in bifunctional lipid uptake/manipulation). This could also provide evidence and further validation of their transport by non-vesicular transport pathways (i.e. LTP).

Answer:

1. The utilized delivery protocol has the advantage that individual lipid molecules are exchanged via α -methyl-cyclodextrin between the outer leaflets of the donor liposomes and the plasma membrane. If liposomes are supplied without cyclodextrin, no lipid signal is detected (first datapoint in panel f of ED Figure 2), indicating that lipids are not incorporated into the plasma membrane by fusion of vesicles or taken up into the cell by endocytosis to a

meaningful extent. Lipid loading by α -methyl-cyclodextrin has been mechanistically characterized in-depth before, see for example here:

<https://www.pnas.org/doi/abs/10.1073/pnas.1610705113>

Given that individual lipids molecules are incorporated into the plasma membrane, the fact that donor liposomes are not strictly required and are in any case removed quickly after lipid incorporation into the outer plasma membrane leaflet, most of the lipid transport occurs after donor liposomes have been removed and lipid transport of individual lipids is differentially affected by genetic perturbations (new Figure 3 below) we are of the opinion that any influence of the initial presence of liposomes on lipid transport kinetics can be ruled out.

We furthermore think that the (fast) loading rates initial loading rates do not affect the intracellular transport, as all lipid probes are incorporated to a similar level (between 1 and 3% of all lipids). In addition, there is no correlation between incorporation efficiency and transport rates. The biggest differences in incorporation levels are found between PE (3 %) and arachidonate containing PC species (1-1.5%), which are both transported quickly intracellularly, whereas the palmitate-containing PC species are incorporated to an intermediate amount (~2%) and transported slowly.

2. For the suggestion to use proteomics, please see our answer to point 4) below as this is crucially linked with the nature of lipid protein interactions.

4) The work actually focuses on a specific pool of bifunctional lipids, i.e. those that are cross-linked to interacting proteins, i.e. those embedded in biomolecular interactions. I understand that the crosslinking step is unavoidable, as the permeabilization of the cell (with detergent if I understand correctly) required for click chemistry will result in the loss of all non-crosslinked lipids, including bifunctional probes that do not interact with proteins. I think this should be explained and described, could it be that it includes a bias? When they normalize for total bifunctional lipids, it's actually for total bifunctional lipids associated with proteins (isn't it?) - could this include artifacts? Could it be that some of the observations described later stem indirectly from this ability (or inability) to interact with proteins (and not so much about transport)? More generally, what about "free bifunctional lipids" (i.e. not associated with proteins)? Do they exist? Are we learning anything from them? Is it acceptable to ignore them? This notion could be addressed as it could represent an advantage of their approach.

Answer:

Reviewer #3 points to a critical point for our work and lipid biology in points 3 and 4: The nature of lipid protein interactions. Lipid protein-interactions come in two main flavors: Specific, high-affinity interactions with long residence times (e.g. structural lipids in integral membrane protein complexes) and transient interactions with binding energies in the range of a few $k_B T$. The lipid residence times of the latter type are estimated to be in the nanosecond range, mainly based on MD simulations and EPR data. Even specific interactions have exchange rates in the microsecond regime. This indicates the presence of three main lipid pools in biological membranes: Tightly bound lipids with long residence times (usually microseconds, in exceptional cases s-min), which constitute a small percentage of all lipids, lipids transiently interacting with proteins in a mostly nonspecific manner and "free" lipids that are not physically interacting with proteins. Assuming a membrane surface coverage of 25% transmembrane helices (typical for e.g. the plasma membrane), the latter two pools are of roughly similar size. Importantly, lipids in the latter two pools, which constitute the vast majority of all membrane lipids, exchange in the nanosecond time regime. A good discussion of these categories, including the lifetime numbers and relative interaction frequencies mentioned here can be found in this excellent, recent review by Ilya Levental and Ed Lyman:

<https://www.nature.com/articles/s41580-022-00524-4>

This has significant implications for both our lipid imaging approach and crosslinking-based identification of lipid-interacting proteins. A crosslinking pulse of 1-10 s means that all individual lipids except those in very long-lived interactions exchange between pools many times during the irradiation period and therefore have rather similar crosslinking probabilities

to the average integral membrane protein. For imaging lipids, the rapid exchange between pools is helpful, as it minimizes the “dark pool” of lipids that is not detected due to the difference in timescales between residence times (nanoseconds) and irradiation (seconds). This means that the obtained imaging signal is in fact representative of lipid localization at any given time and as reviewer #3 correctly assumes, this is a major advantage of the technique, when monitoring inter-organelle lipid distributions. For lateral lipid segregation in the same membrane (not a topic in this manuscript), the length of the crosslinking pulse constitutes the minimal lifetime a domain must have for it to be detected, as more transient structures will be averaged out.

We agree with reviewer #3 that this notion should be discussed in the manuscript and have added a corresponding section to the discussion (Lines 509-514).

For identifying protein interaction partners by photo-crosslinking and subsequent proteomics, the prevalence of non-specific lipid-protein interactions is a major issue, as the large majority of productive “true” crosslinks are due to transient, nonspecific interactions, resulting in datasets that are mostly comprised of biological “noise” (Also discussed in the above-cited review by Levental and Lyman).

This is in our opinion the single most important reason why the results of many mass-spectrometric screens for identifying specific lipid-protein interactions (including studies by us) have been comparatively meager. There currently is no straightforward strategy of using approach for “on-demand” identification of lipid interacting proteins by photo-crosslinking. Using current screening strategies, we would have to be very lucky to identify more than one or two LTPs at most. We hope that reviewer #3 can agree with us that doing this type of resource-intensive experiment (with requires high amounts of bifunctional lipid probe material) with limited success probability is beyond the scope of the current manuscript, which should retain its focus on imaging and lipid transport, in particular since we have added quite a few genetic and pharmacological perturbation experiments that strengthen our initial mechanistic conclusions.

Full disclosure: We are working on strategies to circumvent the selectivity problem in a separate project in the laboratory, similar in scope to the current story, however this is some time away from publication.

For a few important cases, they could perform cell fractionation and biochemically measure bifunctional lipids in the different organellar fractions (without cross-linking). This would also represent (for a few selected cases) an orthogonal verification/validation of their cross-linking/click chemistry approaches measuring the accumulation of bifunctional lipids in the different organelles.

Answer:

Agreed, this would represent an orthogonal validation. We would, however, like to point out, that determination of organelle lipid compositions by organelle fractionation is notoriously error-prone and this was a significant part of the motivation to develop an alternative strategy. We chose to measure the amount a saturated (PC(Y/16:0)) and a polyunsaturated (PC(Y/20:4)) phosphatidylcholine species as they exhibit very different transport kinetics and measure bifunctional PC content as a fraction of total PC, shown in new Supplementary Figure 15, below. We used a sucrose density gradient fractionation as this allowed us to enrich different fractions from the same sample. We find that PC(Y/16:0) levels as a fraction of total PC are highest in the plasma membrane at 4 min, slightly lower at 30 min, whereas the respective level the ER/Golgi membranes rises slightly, but remains lower than in the PM fraction. This trend is in line with the imaging data. Similarly, PC(Y/20:4) is much more evenly distributed already both at the 4 min and the 30 min timepoint, again in agreement with the imaging data. The observed lower PC(Y/20:4) levels at 30 min are consistent with our previous observation of faster metabolism and overall loss of bifunctional lipid material for this species.

NOTE: The obtained percentage of total PC content within the organelle fractions are not directly comparable with the imaging data, which are percentages of total lipid signal, but general trends can of course be compared.

Supplementary Figure 7 Organelle fractionation of cells loaded with bifunctional lipids support lipid imaging data. a. Schematic description of the collection of mammalian cell culture samples loaded with bifunctional lipid probes (species indicated at far left) and processed for lipidomic analysis of cellular fractions corresponding to crude organelle isolates. A total of 48 samples corresponding to 4 fractions (WCL, PM, Other, Mito) in two separate timepoints (4 minutes, 30 minutes) and 3 lipid conditions were quantified. Homogenized cell lysate (in 8% sucrose) was added to a stepped gradient of 80, 45, 30 % sucrose to separate organelle components by density

via ultracentrifugation. Mass spectrometric detection and quantification was performed using both t-SIM and full-scan MS (FTMS) runs, as described in the methods of this manuscript. **b.** Coomassie total protein stain and corresponding Western Blot of the same gel to assess the purity of isolated crude fractions. Densitometry analysis of the Western blot is included below, showing relative protein levels (Western) as a fraction of total protein (Coomassie), scaled from minimum to maximum signal for the respective PM and mitochondria markers so that they can be plotted on the same scale. As expected for biochemical fractionation experiments due to the prevalence of membrane contact sites, there is carryover of the lighter endomembranes into the heavier fractions, whereas the PM fraction is depleted of mitochondrial marker proteins. **c.** Left panels: Exemplary images of PC(Y/16:0) localization at 4 and 30 min timepoints after lipid loading. Right panels: Lipidomic quantification of the bifunctional lipid abundances in crude organelle fractions at 4- and 30-minutes post-lipid addition. PC(Y/16:0) as % of the total PC content is highest in the PM and lower in internal organelles, which is consistent with the imaging data. Scale bar: 10 μ m **d.** Left panels: Exemplary images of PC(Y/20:4) localization at 4 and 30 min timepoints after lipid loading. Right panels: Lipidomic quantification of the bifunctional lipid abundances in crude organelle fractions at 4- and 30-minutes post-lipid addition. PC(Y/20:4) as % of the total PC content is relatively similar in all organelle fractions at both time points, which is consistent with the imaging data. Lower total levels of PC(Y/20:4) at the 30 min timepoint are indicative of faster metabolism and overall loss of bifunctional lipids, which is consistent with bulk lipidomics data. Scale bar: 10 μ m **e., f.** Data is plotted as the proportion of bifunctional PC lipid species to the average abundance of native PC lipids per fraction, which are normalized to an internal standard for lipidomics analysis. Each point is a separate sample replicate (N = 2 biological x 2 technical replicates for all samples except PC(Y/20:4) at 30 minutes). Error bars are standard deviation, and bars show the mean.

Points on bifunctional lipids metabolism:

The authors have carried out an important analysis of the metabolic fate of bifunctional lipids supplied to the cell at the plasma membrane. This section is currently mainly to be found in the supplementary information. In my opinion, this section is very important and interesting for several reasons. It describes the limitations of the approach (possible effect of chemical markers on lipid metabolism and transport). Such limitations are inevitable and not necessarily problematic as long as we understand them precisely. From a more biological point of view, this may also highlight metabolic biases, i.e. lipids from PM do not have the same metabolic fate - which is interesting and also alludes to transport mechanisms. However, this section remains limited to additional information, and I think these aspects would merit presentation in the main text.

5) They mention that the SM probe was metabolically stable. Does this mean that it is not a good substrate for sphingomyelinase, or could another possibility be that its transport is somehow affected? More generally, a description of the metabolic fate of the different probes including a comparison to what is known/expected and a discussion concerning possible shortcomings would greatly help. More generally what results in the Supplementary Material 1 is important and a few Figure panels describing the most salient properties/behaviors of the different probes would be extremely useful for the community and other groups who may want to apply the methods in the future. May be interesting to also analyze the secretome for SM-derived bifunctional probes – authors may still have the conditioned culture media.

Answer:

1. We agree with reviewer #3 that “metabolically stable” was probably not the best term to use to describe the observations with regard to the SM probe in the lipid metabolism annex. SM(Y) constitutes above 90% of the bifunctional lipidome at all timepoints, but the total content of bifunctional lipids in the overall lipidome drops sharply over time after SM(Y) loading. This can have multiple explanations, including secretion of bifunctional SM(Y) but it could also simply be that the amide bond cleavage required to generate free bifunctional fatty acid is the rate limiting step during SM(Y) degradation and products of SM(Y) metabolism never accumulate in the cell due to that reason. We unfortunately did not keep the conditioned medium of the bifunctional lipid from the lipid metabolism experiments and repeating this experiment would be rather time and probe consuming, given that lipid secretome quantifications are much more error-prone than from cellular data.

We have adapted the text to more clearly reflect the different possible options for the observed SM(Y) levels (Line 349-351).

2. We agree with reviewer #3 that a more thorough description of the metabolism of bifunctional probes is warranted. We have moved parts of this section and new data that precisely address the question of metabolic bias / influence of the diazirine/alkyne modification to the revised Figures 5 (previous Fig. 4) (panels a) and ED 5 (panels e and f). Briefly, we now show measures of probe metabolism (initial probe as % of all bifunctional lipids) and of probe retention (bifunctional lipids as % of total lipidome) and move a portion of the corresponding description of the data to the main text (Line 352-372)

We furthermore now include a comparison of the metabolic fate of oleic acid and palmitic acid containing bifunctional PC species with a native, isotope labelled species, PC(18:1/16:0^[13C]) delivered in the same fashion to the outer leaflet of the plasma membrane. Our results show that this species is metabolized similarly to the bifunctional PC species PC(Y/16:0) and PC(16:0/Y), whereas the oleic acid containing species PC(Y/18:1) and PC(18:1/Y) are metabolized faster. The level of retention of the original PC(18:1/16:0^[13C]) as % of all labelled lipids is also similar to the palmitate containing bifunctional lipid species. This leads to the following conclusions:

- The metabolism of bifunctional lipid species most closely resembles native lipids that contain a monounsaturated fatty acid chain in the place of the bifunctional fatty acid, which matches the biophysical characterization in vitro.
- The over-representation of the initially supplied species in the labelled (not overall) lipidome at later timepoints is not an effect that is due to the presence of the diazirine/alkyne modification, but indicative of the metabolic fate of lipids initially residing in the outer leaflet of the plasma membrane of tissue cultured cells.

A section covering the metabolic fate of the bifunctional lipid probes has been added to the main text (Lines 372-381). Revised Figures 5 and ED 5 are shown below.

Figure 5 Lipid metabolism is approximately one order of magnitude slower than lipid transport. a. Bifunctional lipid retention and turnover. The fraction of initially supplied species as percentage of all bifunctional

lipids and the fraction of bifunctional lipids of the total lipidome are exemplarily shown for PC(18:1/Y), PE(18:1/Y) and SM(Y). **b.** Fraction of initially supplied lipid probe as % of the bifunctional lipidome as a proxy for the speed of lipid metabolism. Solid lines indicate mono-exponential fits. SM(Y) data was not fitted as very little interconversion was observed, instead a linear interpolation is shown. **a-b.** Error bars: Mean and 95% CI of 3 biological repeats containing 3 technical replicates each. **c.** Comparison of determined mono-exponential rate constants for the metabolism of individual lipid species. Error bars: Mean and SE of 3 biological repeats containing 3 technical replicates each **d.** Comparison of transport and metabolic rate constants shows that lipid transport is at least one order of magnitude faster. **e.** Lipid transport and metabolism rate constants are highly correlated for PC species despite a clear timescale separation. Error bars metabolic rate constants: SE, transport rate constants: SD, calculated from 100 MC model runs.

Extended Data Figure 5. Ultra-high resolution (UHR) shotgun lipidomics of bifunctional lipid probes. a. Shotgun UHR mass spectrometry resolves lipid peaks spaced by a few mDa and matches bifunctional precursors and their metabolites in multiple lipid classes. Blue line: Section of the spectrum acquired at the conventional ($R_s \sim 120,000$) resolution on Q Exactive mass spectrometer; orange line: Same spectrum section acquired at $R_s \sim 1M$ resolution using optional Booster X2 data processing system and extended (2 sec) transients. Vertical lines are peak centroids. **b.** Mol% profile acquired at two time points (see inset for color coding) of 23 lipid classes (light bars), of which 7 classes comprise lipids with bifunctional lipid moieties (dark bars) produced from PC Y/20:4. **c.** PC profile covering 22 species with 5 species containing the bifunctional fatty acid. PCs bearing a bifunctional fatty acid (16:1) are annotated as endogenous lipids having the same number of carbons and double bonds in both FA moieties, albeit having different (+28.0061 Da) masses. **d.** Bifunctional fatty acids from the source PC(Y/20:4) are incorporated into different lipid classes e.g. the cellular TAG pool consisting of 33 species with 14 species bearing the bifunctional fatty acid. The molar abundance of PC species containing the bifunctional fatty acid other than PC(Y/20:4) (**c**) and TAG (**d**) species increases with time, while the abundance of the starting PC(Y/20:4) decreases. Molar% profiles of native lipid classes (**b**), but also the species profile within PC and TAG classes (**c, d**) are not

perturbed, indicating that the supplemented bifunctional lipids act as true tracer compounds and do not change the overall lipidome compositions. e. Comparison of bifunctional lipid metabolism with the native, isotope labelled, monounsaturated PC species PC(18:1/16:0[¹³C]). f. Development of the PC species distribution of PC(18:1/16:0[¹³C]) and palmitate-containing bifunctional PC species shows similar persistence of the original species in the labelled lipidome and similar product species forming. Shown are Mean and 95% CI of 3 biological repeats containing 3 technical replicates each.

6) The authors also allude to the fact that bifunctional lipids remain surprisingly well represented in the total lipidome (even for short-lived lipids such as PA). As the authors point out, this is surprising.

Answer:

Clarification: This is the case for the “bifunctional” or labelled lipidome, not the overall lipidome. The combined % of all bifunctional lipids of the total lipidome does indeed drop over time, as expected due to ongoing metabolism, secretion and dilution due to cell growth (see Supplementary Figure 8). We did however, initially expect that the bifunctional lipidome would ultimately resemble the native lipidome, where no single species constitutes more than 10% of all lipids.

They propose that this is due to the formation of lipid pools less likely to be metabolized. The authors suggest many hypotheses, but seem to rule out the possibility that this is due to chemical engineering of lipids (i.e. bifunctional lipids have a different metabolic fate/rate) and/or to less susceptibility to transport (e.g. linked to the flippase mechanism they describe later- bifunctional lipids are supplied on liposomes and on both sides of the leaflet). This point should be discussed. Here again, a description of the metabolic fate of the different probes in the context of a comparison with what we expect would greatly help to understand the possible limitations of the analysis. For example, do we expect bifunctional PC not to be metabolized into other glycerophospholipids? If not, can we learn from these discrepancies (origin of PC (PM instead of ER), effect of chemical labels, etc.). This part is important because it delimits the scope and limits.

Answer:

As outlined above in our answer to point 5), isotope-labelled PC species and the corresponding bifunctional PC species delivered to the PM are metabolized similarly. In general, we argue that the presence of diazirine and alkyne groups do not affect lipid metabolism to a large degree as metabolic labelling with a very similar (C15 vs C16 used here) free bifunctional fatty acid led to rapid incorporation into cellular glycerolipids (including PS, PI, PG and CL) with a distribution that reflected the overall lipid distribution (data mainly shown in the SI of the Haberkant et al. paper):

<https://onlinelibrary.wiley.com/doi/full/10.1002/anie.201210178>

We interpret our data such that not all possible interconversions between lipid classes of the cellular lipidome contribute to cellular metabolism, at least in the studied cell line which is cultured in regular nutrient-rich cell culture medium. In particular, our results suggest that once a fatty acid has been incorporated into a PC lipid, certain metabolic paths are preferred, in particular incorporation into neutral lipids, to some extent into PE and fatty acid cycling within the cellular pool of PC species. Other routes are suppressed – there is for instance almost no incorporation into PS.

With regard to the mechanisms, as discussed above, we do not think that lipid loading (the presence of liposomes within the first four minutes of the experiment) is a decisive factor, given that metabolism of bifunctional lipids occurs over many hours. We do think that metabolic bias is the most likely explanation

We have added a section to the main text (lines 341-374) to put our findings with regard to lipid metabolism into context and also added a section on the scope of our new method to the discussion (lines 511-529).

Part on lipid transport:

7) One interesting finding is that specific retrograde transport takes place via non-vesicular pathways. However, this part is currently based mainly on kinetic models; a validation step would make it more convincing. What if they inhibit vesicular transport, etc.? Later, they claim that polyunsaturated PC species were transported up to 10 times faster by the non-vesicular route than saturated PC species. Can they be sure that saturated species were not transported by the vesicular route? It seems that the model in figure 2e does not consider non-vesicular transport from endosomes to the ER. In general, the author should better define the criteria used by their model to predict vesicular versus non-vesicular transport, and validate their model experimentally.

Answer:

These are important points, which were also brought up by the other reviewers. Please see our answers to the individual questions below:

1. Experimental Validation of modelling results

We used three different approaches to experimentally validate our modelling results.

(1) We inhibited vesicular trafficking using two different compounds (Brefeldin A, which blocks ER-Golgi transport and Wortmannin, which blocks endosome maturation and trafficking). In a nutshell, we find that inhibition of vesicular trafficking has little impact on lipid transport except for anterograde transport of SM(Y) (New Figure 3 in the main text, shown below).

(2) We find that KO of a SM-specific lipid transfer protein results in slower SM transport, but does not affect other lipids much (New Figure 3 in the main text, shown below).

(3) Finally, perturbation of lipid asymmetry via the TMEM30A flippase KD has species-specific effects on lipid transport, as the primary substrate PE(18:1/Y) is affected to a much stronger degree than both saturated and unsaturated PC species (Revised Figure 4, shown below). These results are all in line with the modelling results and thus provide significant additional experimental support for our main conclusion that non-vesicular lipid transport is the primary means of specific delivery of individual lipids to organelle membranes.

Figure 3 Pharmacological and genetic perturbations confirm non-vesicular transport as the primary retrograde lipid transport route. **a.** Inhibition of vesicular trafficking between the Golgi and the ER (Brefeldin A) and of endosome formation and trafficking (Wortmannin). **b.** Transferrin uptake and localization and lipid localization in control cells and drug treated cells at 4 min and 30 min after lipid loading. Scale bar: 10 μ m. Transferrin images are shown at identical settings, lipid images are brightness-contrast adjusted to allow for comparing lipid distributions at different timepoints. **c.** Quantification of lipid distribution between the ER and endo-lysosomes (mean and SD of 8-10 fields of view) after treatment DMSO (gray), Brefeldin A (pink), Wortmannin (light blue). **d.** Schematic representation of GLTPD1-mediated lipid transfer and kinetic model used to assess effects of GLTPD1 KO on non-vesicular lipid transport. **e.** Comparison of SM(Y) time-course experiments in GLTPD1 KO and U2OS WT cells. Scale bar: 10 μ m. **f.** Rate constants for retrograde and anterograde plasma membrane-ER lipid transport for PC(18:1/Y), PE(18:1/Y) and SM(Y). Kinetics were constructed from 6 independent time points (mean and SD) containing 5 field of views each with 5-10 cells. Mean and SD were calculated from 100 MC model runs. Pairwise effect size tests, Cohen's d values: >0.01 very small (VS), >0.20 small (S), >0.50 medium (M), >0.80 large (L), >1.20 very large (VL), >2.00 huge (H).

Figure 4 Genetic perturbation experiments confirm involvement of flippases in species-specific directional lipid transport. **a.** Schematic representation of lipid trans-bilayer movement (lipid flipping) and non-vesicular lipid transport by lipid transfer proteins. **b.** Kinetic model for exchange of lipids between the PM and the ER. **c.** Comparison of time-course experiments for PE(18:1/Y) show that lipid internalization dynamics are slower in HCT116 TMEM30A KD cells compared to HCT116 wild type cells. Coloured arrows indicate lipid localization in different membranes types (green: PM, yellow: ER). Scale bars: 10 μ m. Images are brightness-contrast adjusted to facilitate comparing intracellular lipid localization. **d.** Quantification of PE(18:1/Y) internalization kinetics and model fits. **e.** Quantification of PC(Y/16:0) internalization kinetics and model fits. **f.** Quantification of PC(Y/20:4) internalization kinetics and model fits. **d-f.** Kinetics were constructed from 5 independent time points (mean and SD) containing 3-15 field of views each with 5-10 cells. **g.** Rate constants and quasi-equilibrium constants for retrograde and anterograde plasma membrane-ER lipid transport for PC(Y/16:0), PC(Y/20:4) and PE(18:1/Y). Mean and SD were calculated from 100 MC model runs. Pairwise effect size tests, Cohen's d values: >0.01 very small (VS), >0.20 small (S), >0.50 medium (M), >0.80 large (L), >1.20 very large (VL), >2.00 Huge (H).

2. Model structure and differentiation of vesicular and non-vesicular lipid transport

The two *a priori* assumptions of the model are that retrograde lipid transport along the endocytosis pathway PM \rightarrow endosomes \rightarrow Golgi \rightarrow ER is primarily vesicular and direct exchange of lipids between the PM and the ER is non-vesicular. This is in line with all known evidence as there is no known bulk vesicular transport between ER and PM. The model is designed to distinguish between these two routes. Thus, we do get estimates of lipid amount transported by vesicular and non-vesicular pathways for each lipid. We now included a more precise description of the model design choices in the main text (lines 189-201). Independent of the kinetic modelling the quantification of lipid amount in each organelle directly shows that the % of bifunctional lipids in ER rises much faster and to a higher level compared to the amount in endosomes. Therefore, the endosome trafficking pathway cannot account for the fast bulk lipid transport into the ER.

3. Non-vesicular lipid transport between endosomes and the ER

Regarding non-vesicular transport from endosomes to the ER: We actually tried to implement that route into the kinetic model, however this led to over-parametrization and we did not find unique solutions for individual rates anymore. We would need an entirely new set of transport time course experiments with the endosome as a starting point to determine this rate. We

decided on the current models, as they could be fully parametrized and explain the data well, essentially an Ockham's razor argument.

We tried to circumvent this issue by knocking out one of the key bridge-like lipid-transfer proteins residing at Endosome-ER contacts, VPS13C, and obtain an estimate of the reduction in transport between the ER and endosomes using a minimal kinetic model. However, we did not find a big effect, which could either mean that on a global, cell-wide scale this pathway does not contribute in a major way to lipid flux, or simply, that there is redundancy for this particular gene. Since these results were rather inconclusive, we decided to not include them in the manuscript to keep the manuscript size somewhat manageable. The data are shown below in reviewer Figure 1

Reviewer Figure 1. VPS13C KO effects on lipid transport are small and inconclusive. **a.** Schematic representation of lipid transport by bridge-like LTPs. **b.** Kinetic model used to assess transport between the plasma membrane, the ER and Endosomes. **c.** Model fits of lipid transport for PC(18:1/Y), PE(18:1/Y) and SM(Y) in U2OS WT and VPS13C KO cells. Rate constants describing PM-ER lipid transport for PC(18:1/Y), PE(18:1/Y) and SM(Y) in U2OS WT and VPS13C KO cells. Mean and SD were calculated from 100 MC model runs. Pairwise effect size tests, Cohen's d values: >0.01 very small (VS), >0.20 small (S), >0.50 medium (M), >0.80 large (L), >1.20 very large (VL), >2.00 Huge (H). **e.** Quasi-equilibrium constants describing PM-ER lipid transport for PC(18:1/Y), PE(18:1/Y) and SM(Y) in U2OS WT and VPS13C KO cells.

8) The authors then explore an interesting hypothesis that trans-layer transport may play a role in non-vesicular lipid transport. However, many other mechanisms could explain the directionality of lipid transport (often against concentration gradients), such as metabolism (i.e. once transported, the lipid is rapidly metabolized into products that are not transported by the same LTP), which should probably also be mentioned. Their metabolic data (i.e. the metabolic fate of bifunctional lipids) could be re-examined in the light of this (alternative) hypothesis.

Answer:

Reviewer #3 is correct that there are quite a few conceivable mechanisms for directional lipid transport, for instance by localized metabolism. We would argue that our bulk lipidomic data suggest that metabolism of the lipid probes themselves is likely not a viable explanation as there is a) a clear time scale difference between the speed of probe transport and probe metabolism and b) the model variants specifically accounting for bulk metabolic turnover return essentially the same rate constants for lipid transport processes as the “transport only” variant (see Fig 2). We can of course not rule out coupling to other, faster metabolic processes such as the PI(4)P/PI cycle phosphorylation-dephosphorylation cycle. However, these would have to be localized pools of lipids on organelle membranes which we cannot follow in the bulk lipidomics data. In our opinion, directionality of lipid transport is probably not explained by one overriding mechanism and the contribution of individual mechanisms probably differs between lipids. Our results show that lipid asymmetry plays a role and coupling to lipid metabolism has already been shown to be important. We have clarified the corresponding sections in the manuscript (Lines 279-282).

9) Validation should be provided to show that knockout of the common PM flippase subunit does indeed lead to a flipping defect (and that this function is not redundant, etc.) - i.e. show that PE/PS is exposed at the cell surface (in non-permeabilized cells using specific probes).

Answer:

We now show that the steady state distribution of aminophospholipids at the plasma membrane is indeed altered by detecting phosphatidylserine exposure on the outer plasma membrane leaflet in KD cells, but not in WT cells, confirming that the KD cells do in fact exhibit an aminophospholipid flipping defect (New Supplementary Figure 4, shown below).

Supplementary Figure 4 TMEM30A KD results in pronounced localization of PS on the outer plasma membrane leaflet. Upper panels: HCT116 WT cells treated with the small molecule plasma membrane marker Cell mask red and the PS sensor Lact-C2-EGFP. No recruitment of Lact-C2-EGFP from the extracellular medium to the outer plasma membrane leaflet is visible. Lower panels: HCT TMEM30A KD cells treated with the small molecule plasma membrane marker Cell mask red and the PS sensor Lact-C2-EGFP. Recruitment of Lact-C2-EGFP from the extracellular medium to the outer plasma membrane leaflet is indicative of a flipping defect. Scale bar: 10 μ m.

Furthermore, as these experiments are carried out in the presence of excess extracellular PE present on liposomes, i.e. presented on both leaflets of artificial membranes, what's the idea/model? PE is loaded into the cell on both sheets anyway. So, could it be that what is observed here is a consequence of the way lipids are delivered to the cell, but is not at play under more physiological conditions when endogenous PE comes from the inner leaflet of the PM or the ER?

Answer:

Lipids are loaded into the outer leaflet of the PM in an alpha-methyl-cyclodextrin mediated exchange reactions as individual molecules, see also our detailed answer to point 3. Direct fusion of the liposomes with the PM can be ruled out, as we do not observe any incorporation of bifunctional lipids into the cell in the absence of cyclodextrin and dextran in the cell medium does not enter the cells during loading (see ED Figure 2). Furthermore, as discussed above, liposomes are removed after the loading pulse, so we do not think they perturb the lipid transport processes.

More generally, every experiment aimed at determining actual reaction or transport rate constants has to be a kinetic relaxation experiment. For this the investigated system has to be rapidly (faster than the investigated reactions) removed from equilibrium or steady state through a perturbation after which the return to the steady state / equilibrium is monitored. As long as the underlying reaction network is not changed, the specific nature of the perturbation does not matter, as the rate constants are intrinsic to the reaction network. We make sure of that by using a tracer approach that does not affect the overall endogenous lipid composition of the cell, which constitutes the majority of lipids (see ED Figure 5).

Therefore, the lipid loading procedure enabled us to follow the redistribution of lipids from the outer leaflet of the PM and assess the influence of lipid transbilayer movement on inter-organelle lipid transport, which would not have been possible if an unperturbed distribution on both PM leaflets would have been the starting points.

The drawback of the initial PM localization is of course that information about transport processes further downstream from the PM is limited (here for instance PM-Mito exchange or anterograde lipid transport). For this, other starting points would be required. See also our answer with regard to trifunctional lipids below.

Minor points:

- In paragraph "" lines 70-78, the authors briefly describe alternative approaches to studying lipids. This should also include tri-functional lipids, i.e. combining photocaging with cross-linking/click chemistry labels, as they also represent interesting alternatives for such analyses allowing targeting the release of the lipid from different organelle (not only the PM) in the future. They should also be mentioned.

Answer:

We do in fact cite the original PNAS paper on trifunctional lipids (Höglinger et al., PNAS 2017), but did not want to place it to prominently as one of us (AN) is the chemist who made the first trifunctional lipid probes for that paper and we were a bit reluctant to highlight our own work too much. Reviewer #3 is correct that organelle-specific trifunctional lipids provide one – and possibly the most flexible – possibility for different lipid “starting points”. There even has been a recent example of one such probe from Doris Höglinger’s lab. The drawback (again, speaking from experience) is that the synthetic effort required to generate a full panel of organelle-specific trifunctional lipids is pretty daunting. We have included a brief section in the discussion where we discuss alternative lipid delivery strategies including trifunctional lipids (lines 549-556).

- Figure 1 Panel e. The color code for the different locations is difficult to “decode” as the ER and lipids have very similar ranges of colors (at least on my screen). We cannot see cases of assigned to ER.

Answer:

We actually do not show an overlay of ER marker and lipid signal. The right half of the original panel 1e only shows assigned lipid signal, the left half the original images and the marker

channels. It is, however, clear how this can be confusing and we have now split up this dataset into two panels, which should improve clarity. The magenta/blue/yellow LUT was chosen as this is one of the most colorblind friendly LUT combinations for three-color overlays, and the orange hot LUT for the lipid signal was chosen as it has great dynamic range for visual perception without resorting to too many colors (e.g. fire LUT). We have tested quite a few options during the initial manuscript preparation and liked this combination the most.

- Line 171-173; the authors state that "...transport rate constants for vesicular transport via endosomes with non-vesicular transport to the ER revealed that non-vesicular trafficking was up to an order of magnitude faster for all lipids compared to vesicular transport." This seems to me to be a circular argument, as both vesicular and non-vesicular transport were initially modeled from transport kinetics. This needs clarification and/or rewording. Perhaps with a graph describing how the model has been constructed and/or what are the key parameters that define vesicular and non-vesicular transport?

Answer:

This is a good catch. We now state more clearly that the basic assumptions that our models are based on, see our answer to point 7 above.

- Line 223-224 the authors claim "While some lipid transfer proteins can move cholesterol, PA and PS against concentration gradients by PI(4)P₂ co-transport^{8,31}, it is unclear which process provides the energy for directional transport of other lipids, in particular via bridge-like lipid transfer proteins" is not completely exact. In those cases, we know that the directionality of PS or cholesterol transport is driven to the counter transport of PI(4)P back to the donor membrane and its dephosphorylation into PI in this donor membrane. Energy/ATP hydrolysis is provided by rounds of PI phosphorylation dephosphorylation and PI(4)P. The authors should clearly mention this alternative hypothesis. i.e. that directionality is driven by lipid metabolism.

Answer:

This is actually what we meant to say. For cholesterol, PA and PS, directional transport is driven by PI(4)P₂ co-transport and of course the energy comes ultimately from a PI-phosphorylation/dephosphorylation cycle and thus localized metabolism. We rephrased the section to make this clearer (see lines 279-382).

- Line 225: BTW it is not PI(4)P₂ but PI(4)P

Answer:

Corrected.

Referee #1 (Remarks to the Author):

The authors have successfully addressed my previous concerns, and the new experiments contribute sufficient novel insights into the biology of lipid transport and metabolism for me to recommend publication of this work, which is now significantly improved. This is an amazing tour-de-force in term of methodological developments that I expect will contribute to important further future discoveries in the field of lipid biology.

I only have a few minor points to improve the clarity of the manuscript:

Abstract

- "Here we measured transport and metabolism"  I think should be replaced by "Here we measured retrograde transport and metabolism" as forward transport is not directly estimated in this work

- "active lipid flipping and passive non-vesicular transport" should be replaced by the more neutral and precise terms "energy-dependent" and "energy independent"

Figure 3c: the color code of the 3 treatments (grey, pink, cyan) could be also explained directly in the figure, rather than only in the legend.

Stefano Vanni

Author response:

We thank Stefano for his kind words and have implemented his suggestions in the final manuscript versions.

Referee #2 (Remarks to the Author):

My concerns have been carefully and thoughtfully addressed and I now favor publication of this study.

Author response:

We thank referee #2 for the positive assessment of our work

Referee #3 (Remarks to the Author):

The authors have taken into account all my points and comments. The revised manuscript has been considerably improved. This is an important work, the approach is powerful and elegant. The results show that it allows us to tackle aspects hitherto unexplored in lipid biology.

Author response:

We thank referee #3 for the positive assessment of our work